# 🧩 CubeBench: Diagnosing Interactive, Long-Horizon Spatial Reasoning under Partial Observations

**Huan-ang Gao**[1]* **Zikang Zhang**[1]* **Tianwei Luo**[1] **Kaisen Yang**[1] **Xinzhe Juan**[3] **Jiahao Qiu**[2]
**Tianxing Chen**[4] **Bingxiang He**[1] **Hao Zhao**[1] **Hao Zhou**[1] **Shilong Liu**[2]† **Mengdi Wang**[2]†

[1]THU  [2]Princeton  [3]SJTU & UMich  [4]HKU

{gha24,zhang-zk21}@mails.tsinghua.edu.cn
{sl8264,mengdiw}@princeton.edu

## Abstract

Large Language Model (LLM) agents, while proficient in the digital realm, face a significant gap in physical-world deployment due to the challenge of forming and maintaining a robust spatial mental model. We identify three core cognitive challenges hindering this transition: spatial reasoning, long-horizon state tracking via mental simulation, and active exploration under partial observation. To isolate and evaluate these faculties, we introduce **CubeBench**, a novel generative benchmark centered on the Rubik's Cube. CubeBench uses a three-tiered diagnostic framework that progressively assesses agent capabilities, from foundational state tracking with full symbolic information to active exploration with only partial visual data. Our experiments on leading LLMs reveal critical limitations, including a uniform 0.00% pass rate on all long-horizon tasks, exposing a fundamental failure in long-term planning. We also propose a diagnostic framework to isolate these cognitive bottlenecks by providing external solver tools. By analyzing the failure modes, we provide key insights to guide the development of more physically-grounded intelligent agents. **Webpage:** https://cubebench.c7w.tech/

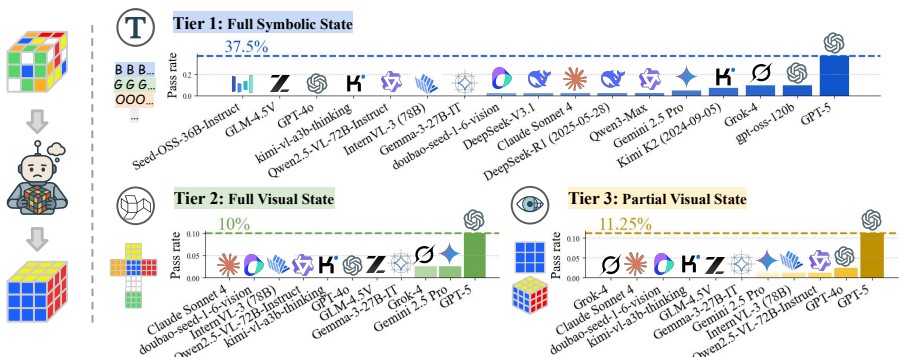

Figure 1: An overview of the performance of leading LLMs on the CubeBench benchmark, broken down by its three diagnostic tiers. **Tier 1 (Full Symbolic State)** tests foundational state tracking using complete symbolic information, where the best average pass rate is only 37.5%. **Tier 2 (Full Visual State)** challenges visual and spatial reasoning by requiring agents to interpret a 2D unfolded map, and **Tier 3 (Partial Visual State)** evaluates active exploration from partial views. Across all tiers, GPT-5 emerges as the top-performing model, though the results highlight a significant performance gap between symbolic and visual reasoning tasks.

---

*Equal contribution.
†Corresponding author.

# 1 INTRODUCTION

Agents powered by Large Language Models (LLMs) have demonstrated remarkable potential within the digital realm (Gao et al., 2025; Fang et al., 2025). Their proficiency in using tools to navigate websites or write code heralds the dawn of general-purpose AI assistants (Luo et al., 2025; Ma et al., 2025). However, a far grander ambition is to deploy these agents into the physical world. This vision confronts a significant gap: an agent's success on one-dimensional, symbolic tasks does not readily translate to effective decision-making in three-dimensional, dynamic environments. The physical world demands more than language comprehension; it requires the ability to form and maintain a robust *spatial mental model* (Johnson-Laird, 1980; 1983).

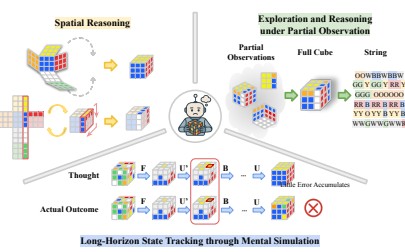

This gap manifests as critical deficiencies in the core cognitive abilities of current agents, which we show in Fig. 2. We identify three such challenges. The first is ***Spatial Reasoning***: physical tasks are inherently three-dimensional, requiring an agent to comprehend an object's geometry, the relative positions of its components, and the precise consequences of actions in 3D space. The second is ***Long-Horizon State Tracking through Mental Simulation***. Unlike digital tasks where state is often externally visible, physical interaction requires an agent to internally maintain and update its world model over long action sequence planning, where even minor errors can accumulate and lead to catastrophic failure. Finally, and most crucially, is the ability for ***Exploration and Reasoning under Partial Observation***. The real world rarely provides complete information, so an agent must actively explore its environment to construct a complete mental model from limited views.

Figure 2: Visualization of the three core cognitive challenges required for spatial reasoning.

To rigorously measure and advance these core capabilities, isolated from the complexities of physical perception, we introduce **CubeBench**, a novel, generative benchmark centered on the Rubik's Cube. We posit that the cube serves as an ideal laboratory; its deterministic rules and vast state space allow us to conduct controlled experiments that isolate the three core cognitive faculties. To achieve this, CubeBench features a three-tiered diagnostic framework to progressively probe an agent's capabilities: Tier 1 tests foundational state tracking with complete symbolic information; Tier 2 challenges visual and spatial reasoning by requiring the creation of a 3D model from a 2D unfolded map; and Tier 3 evaluates active exploration using only partial visual information.

Our comprehensive evaluation on CubeBench reveals a staggering performance gap in current LLMs. The results are stark: across all models, **the pass rate on any long-horizon task is a uniform 0.00**, exposing a critical failure in long-term planning and state tracking. Even on short-horizon symbolic tasks, the top-performing LLM, GPT-5 [1] , achieves a success rate of just 0.75, merely matching the performance of a traditional Policy Gradient agent and highlighting the difficulty of even basic structured reasoning. Furthermore, our experiments with dense rewards show that while external feedback can provide a local guide on simpler problems, it is insufficient to overcome these core planning deficits. By equipping agents with solver tools, our diagnostic framework successfully pinpoints these failures, isolating long-horizon planning as a primary bottleneck and the inability to reason from partial observations as a more fundamental challenge.

In summary, the primary contributions of this paper are:

- We identify and formalize three core cognitive challenges that impede the deployment of LLM agents into the physical world: spatial reasoning, long-horizon state tracking, and exploration under partial observation.

- We propose CubeBench, a novel, generative benchmark for the controlled evaluation of these cognitive challenges, decoupled from the complexities of visual perception.

- Through extensive experiments on leading LLMs, we reveal their current limitations in forming and utilizing spatial mental models, offering key insights for future development.

---

[1] Accessed via OpenRouter (ID: `openai/gpt-5`) with unspecified reasoning effort.

Table 1: A comparison of agentic benchmarks. Besides the three core cognitive challenges, we also evaluate key task characteristics: **Verifiable Outcome Reward**, which assesses if the environment operates on fixed, predictable principles rather than subjective or stochastic outcomes such as LLM-as-a-judge; **Non-static environment**, which measures if the *state* of the environment changes with different agent actions; and whether the task is **Humanly Challenging**, requiring deliberate exploration for acquiring problem-solving skills beyond simple perception or motor control.

| | Core Cognitive Challenges | | | Environmental & Task Properties | | |
|---|---|---|---|---|---|---|
| **Benchmark Type** | **3D Reasoning** | **Long-Hori. ST. Track** | **Partial Obs.** | **Verifiable Outcome Rwd.** | **Non-static Env.** | **Humanly Challenging** |
| Search | ✗ | ✓ | ✗ | ✓ | ✗ | ✓ |
| Code | ✗ | ✓ | ✓ | ✓ | ✓ | ✓ |
| GUI | ✗ | ✓ | ✗ | ✓ | ✓ | ✗ |
| Embodied Simulators | ✓ | ✓ | ✓ | ✓ | ✓ | ✗ |
| Gyms | ✗ | ✓ | ✓ | ✓ | ✓ | ✓ |
| ARC-AGI-3 | ✗ | ✓ | ✗ | ✓ | ✓ | ✓ |
| MINDCUBE | ✓ | ✗ | ✗ | ✓ | ✗ | ✗ |
| **CubeBench (Ours)** | ✓ | ✓ | ✓ | ✓ | ✓ | ✓ |

✓ for being explicitly designed to test the capability. ✓ for being partially tested in the benchmark. ✗ for benchmarks not primarily focusing on this capability.

- We demonstrate through intervention studies—specifically solver integration and learning from experience—that the identified limitations of base LLMs can be significantly mitigated, pointing toward promising avenues for building more capable agents.

## 2 RELATED WORKS

**Self-evolving Agents.** The paradigm of AI is shifting from static, pre-trained models to dynamic, *self-evolving agents* (Gao et al., 2025; Fang et al., 2025) capable of continual learning and adaptation from experience (Wang et al., 2024a;b; Luo et al., 2025; Zhang et al., 2025b; Hu et al., 2024a; 2025; Liang et al., 2024a; Ma et al., 2025). Unlike foundational agents with fixed capabilities, self-evolving agents can autonomously modify their own components—including memory (Zhang et al., 2024a; Zhou et al., 2024; Liang et al., 2024b; Xu et al., 2025b; Zhao et al., 2024a; Chhikara et al., 2025; Guan et al., 2024; Yu et al., 2025), tools (Qiu et al., 2025; Haque et al., 2025; Zheng et al., 2025; Zhao et al., 2024b; Qu et al., 2025; Wang et al., 2025), and architecture (Zhuang et al., 2025; Zhang et al., 2025c; Sapkota et al., 2025) — in response to environmental interaction. As these agents evolve to tackle the physical world, a fundamental shift in their evaluation is required—moving beyond traditional static assessments to benchmarks that can rigorously measure the acquisition and application of *spatial intelligence*.

**Benchmarks for Self-evolving Agents.** Existing benchmarks (Chan et al., 2024; Chen et al., 2024b; Wei et al., 2025; Levy et al., 2024; Wu et al., 2025; Mialon et al., 2023; Liu et al., 2023; Chen et al., 2025a; Zhu et al., 2025; Hu et al., 2024b), however, are not designed for these dynamics. As shown in Table 1, different categories of benchmarks test these cognitive skills to varying degrees, but none provides a focused, isolated evaluation. Digital environments for Search and GUI interaction (Xie et al., 2024; Zhang et al., 2025a; Levy et al., 2024; Wu et al., 2025; Zhou et al., 2023; Deng et al., 2023; Mialon et al., 2023; Wei et al., 2025; Phan et al., 2025), for instance, are primarily 2D and feature explicit states, thus not addressing 3D spatial reasoning. While Code and Gym environments (Hu et al., 2024b; Jimenez et al., 2023; Chan et al., 2024; Chen et al., 2025b; Aleithan et al., 2024; Yang et al., 2024; Xu et al., 2024; Su et al., 2025; Tassa et al., 2018; Yu et al., 2020; Rajeswaran et al., 2017) require long-horizon state tracking, they do not involve the complex 3D geometric understanding that is crucial for physical-world tasks. Embodied simulators (Gao et al., 2024; Yang et al., 2025b; Li et al., 2024; Savva et al., 2019b; Shridhar et al., 2021; Kolve et al., 2017) do engage all three faculties but inherently couple them with complex visual perception, making it difficult to isolate cognitive failures. While recent work like MindCube (Liu et al., 2024) evaluates reasoning on static 3D scenes, our work introduces the challenge of updating a spatial model through long-horizon, state-altering interaction.

In this work, we develop CubeBench, which is specifically designed to fill this gap by decoupling perception from reasoning. Its deterministic, rule-based nature makes it an ideal suite for studying an agent's evolution; when an agent fails, the cause can be precisely attributed to a failure in its internal spatial model or its long-horizon planning, as shown in Sec. 4.3. Furthermore, CubeBench's generative nature allows for the creation of a virtually infinite curriculum of tasks with fine-grained difficulty, enabling the rigorous evaluation of an agent's ability to learn and adapt over time—a cornerstone of assessing true self-evolution (Gao et al., 2025).

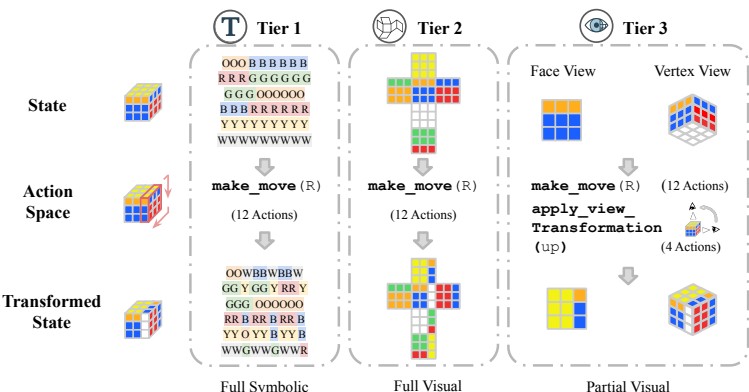

Figure 3: **Illustration on the three-tiered task of CubeBench.** Tier 1 (Full Symbolic State) provides the agent with complete state information in a string format, which makes the problem a fully observable MDP. Tier 2 (Full Visual State) presents the full state as a 2D unfolded map, which challenges the agent's visual thinking. Tier 3 (Partial Visual State) provides only a partial view of the cube (Face view or Vertex view), which requires the agent to explore the environment to gather the full state information.

## 3 THE CUBEBENCH BENCHMARK

### 3.1 TASK DEFINITION

We formalize the Rubik's Cube challenge as a Partially Observable Markov Decision Process (POMDP), providing a structured framework to analyze agent behavior. A POMDP is defined by a tuple $(S, A, T, R, \Omega, O)$, where $S$ is a set of states, $A$ is a set of actions, $T$ is the state transition function, $R$ is the reward function, $\Omega$ is a set of observations, and $O$ is the observation function. In this context, the agent's goal is to learn a policy $\pi(a|o)$ that selects an action $a \in A$ given an observation $o \in \Omega$ to maximize the expected cumulative reward. We now define each of these components within the CubeBench environment.

#### 3.1.1 STATE SPACE

The state space $S$ encompasses all possible configurations of the 3x3x3 Rubik's Cube. The internal state of the cube, $s \in S$, is deterministically represented by a data structure that tracks the color of the 54 individual facelets (stickers). This symbolic representation is unambiguous and allows for perfect state tracking within the simulation. The Rubik's Cube is a classic example of a system governed by the principles of group theory. Each move corresponds to a permutation of the cube's facelets, and the set of all possible move sequences forms a mathematical group. This deterministic, non-stochastic nature makes it an ideal environment for isolating an agent's reasoning and planning capabilities from the complexities of physical uncertainty. The state space is vast, containing over 43 quintillion ($4.3 \times 10^{19}$) unique configurations, yet it is finite and structured. This combination of immense scale and deterministic rules makes it a compelling microcosm for studying autonomous problem-solving on tasks that are too large for naive search but are perfectly predictable.

#### 3.1.2 OBSERVATION SPACE

The observation space $\Omega$ is defined by the observation function $O(s)$, which maps the true internal state $s$ to an observation $o$ that is presented to the agent. As shown in Fig. 3, CubeBench features a three-tiered observation space, where each tier presents the state information in a different modality, posing distinct perceptual challenges.

**Tier 1: Full Symbolic State.** The observation is a 54-character string that symbolically represents the complete state of the cube. Each character corresponds to the color of a single facelet (e.g., 'W' for White, 'R' for Red, 'B' for Blue, 'O' for Orange, 'G' for Green, 'Y' for Yellow). In this tier, the observation function provides the full state information as a 54-character structed string, making the problem a fully observable MDP.

**Tier 2: Full Visual State.** The observation is a single image depicting the cube's complete 2D unfolded map, which visually presents all 54 facelets in a planar layout. This tier specifically challenges an agent's visual reasoning capabilities, requiring it to mentally *fold* the 2D layout into a coherent 3D spatial model to understand the adjacency of faces that are not contiguous in the planar representation.

**Tier 3: Partial Visual State.** The observation is a single image of a partial view of the cube. This can be either an image of a single face (*face view*) or an image from a corner's perspective showing three adjacent faces (*vertex view*). In this tier, the observation function provides incomplete state information, thus formulating the task as a true POMDP.

### 3.1.3 ACTION SPACE

The action space $A$ consists of the set of discrete, deterministic commands an agent can execute to interact with the environment.

**State Transition Actions:** The primary action for rotating the cube, which implements the environment's transition function $T(s, a)$. A `make_move` command accepts one of 12 standard Singmaster notation inputs corresponding to a 90-degree rotation of a face: `F` (Front), `B` (Back), `L` (Left), `R` (Right), `U` (Up), `D` (Down), and their counter-clockwise prime versions (`F'`, `B'`, etc.).

**Observation-Altering Actions:** An action exclusive to Tier 3 that allows the agent to change its observational viewpoint (i.e., up, down, left, right) without altering the cube's underlying state $s$. This is the primary mechanism for exploration in the partially observable setting.

### 3.1.4 REWARD FUNCTION

The agent's objective is to reach the solved state. We define two types of reward signals within CubeBench to facilitate and evaluate this process.

**Sparse Terminal Reward:** The primary success metric is a sparse, binary reward. The agent receives a reward of $R = 1$ upon entering the terminal solved state (i.e., all stickers on six faces are matched), and $R = 0$ for all other state transitions. The agent's goal is to find a policy that maximizes the probability of achieving this terminal reward within the given constraints.

**Dense Progressive Reward:** To potentially guide the agent's search process, we also implement an optional *dense* reward mechanism. Unlike a state-value function, our dense rewards are calculated as the *change* in a given metric before and after a state transition action. Specifically, the reward $R_t$ for taking state transition action $a_t$ in state $s_t$ to reach state $s_{t+1}$ is defined as the difference in a metric function $\phi(s)$:

$$R_t = \phi(s_{t+1}) - \phi(s_t)$$

We implemented and tested three different metric functions ($\phi$) to explore how the conceptual granularity of the feedback affects agent performance. *(1) Sticker Metric ($\phi_{sticker}$):* This function quantifies the total number of individual facelets (stickers) that are in their correct home positions. The score $\phi_{sticker}(s)$ ranges from 9 for a highly scrambled cube to 54 for the solved state. This provides a fine-grained, low-level signal of progress. *(2) Face Metric ($\phi_{face}$):* This function counts the number of fully solved faces, where all 9 stickers on a face are correct. It provides a high-level, more conceptually grounded signal that is sparser than the sticker metric. *(3) Heuristic Metric ($\phi_{heuristic}$):* This function uses an algorithmic heuristic from a common solving method to estimate the distance to the goal state. It is designed to provide a more informed, albeit abstract, numerical signal, which we explain in detail in Sec. J. As a default setting, we also include a `no_reward` condition where $R_t = 0$ for all transitions.

## 3.2 TASK EVALUATION AND GENERATION

**Agent Interaction Protocol.** The agent's interaction with the environment follows the ReAct paradigm (Yao et al. (2022)), structured into a sequence of decision-making steps. As shown in Fig. 4, we define a single *step* as a complete `Thought-Code-Observation` block. Within each step, the agent first generates its reasoning (`Thought`), then writes and executes code to interact with the environment (`Code`), and finally receives the output of that code as feedback (`Observation`) for its next cycle. Each experimental run is subject to a maximum of 20 steps

and a timeout of 30 minutes to ensure fair comparison. Note that in each step, the agent could write code to make more than one move.

**Evaluation Metrics.** Our primary metrics are designed to measure both success and effort. The *Pass Rate* is the fraction of test cases successfully solved within the execution constraints, serving as the primary measure of an agent's capability. To quantify search effort, we use the *Number of* `make_move` *calls (#MM)* as a proxy. We report this metric aggregated in three ways: the average over all normally terminated runs, the average over only successful runs, and the maximum count observed across all normally terminated runs.

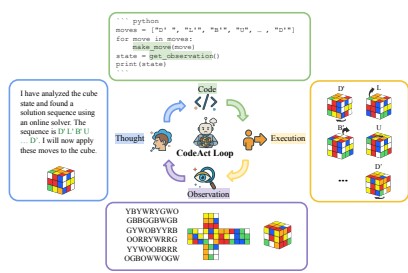

Figure 4: Illustration on the interaction protocol.

**Task Generation and Difficulty Scaling.** We define task difficulty based on the *optimal* number of moves required to solve a given cube configuration, a metric we refer to as the state's *depth*. A state's depth serves as a robust proxy for its complexity; solving high-depth states is infeasible through random exploration and necessitates a coherent strategy. To generate our test cases, we employ a provably optimal solver (see Appendix K). For a target depth $d$, we generate scrambled states and confirm their optimality by verifying that a solution of length $d$ exists, but no solution of length $d - 1$ can be found. This guarantees the true depth is precisely $d$. The detail of this process is described in the appendix. To analyze agent performance across varying complexities, we group these cases into two distinct categories: *Short-Horizon* tasks, comprising states with depths of 1, 2, 3, and 4, and *Long-Horizon* tasks, which include the more challenging depths of 8, 12, 16, and 20. The configuration of the generated test split is described in detail in Sec.G.

# 4 DIAGNOSING LLM AGENT CAPABILITIES ON CUBEBENCH

In this section, we introduce our systematic framework for evaluating Large Language Model (LLM) agents on the CubeBench benchmark. As shown in Fig. 5, the evaluation process is designed as a three-part diagnostic, structured around three central research questions that aim to progressively uncover the cognitive strengths and weaknesses of current agents:

> **Questions**
>
> - **Q1:** What are the baseline capabilities and limitations of current LLM agents when trying to solve the typical cube problem in an unaided setting?
> - **Q2:** Can the introduction of dense reward signals effectively guide an agent's context-based reasoning process and enhance its performance on these complex spatial tasks?
> - **Q3:** How can we design a diagnostic evaluation to isolate the impact of each core cognitive challenge, thereby identifying the primary bottlenecks for agent failure—is it high-level planning, state reconstruction from partial perception, or spatial reasoning?

## 4.1 EXPERIMENT 1: BASIC AGENT WITH NO AID

**Experimental Setup.** To answer our first research question, we establish the baseline capabilities of unaided LLMs. For this experiment, we utilize the ***Basic Agent*** configuration. This agent is provided with only the fundamental interaction tools: `make_move`, `get_observation`, and, for Tier 3 tasks, `apply_view_transformation`. It must solve the task from first principles, relying entirely on its internal faculties for planning and reasoning. We evaluated this agent across all four observation modalities (`Full Symbolic`, `Full Visual`, `Face view`, and `Partial Visual`) on both short- and long-horizon tasks. The results are presented in Table 2.

**Key Observations.** **(1)** All models exhibit a 0.00 pass rate on long-horizon tasks across all input modalities. **(2)** For short-horizon tasks, non-zero pass rates are achieved almost exclusively with the symbolic string input; performance on all visual inputs is near or at zero for most models. **(3)** A clear

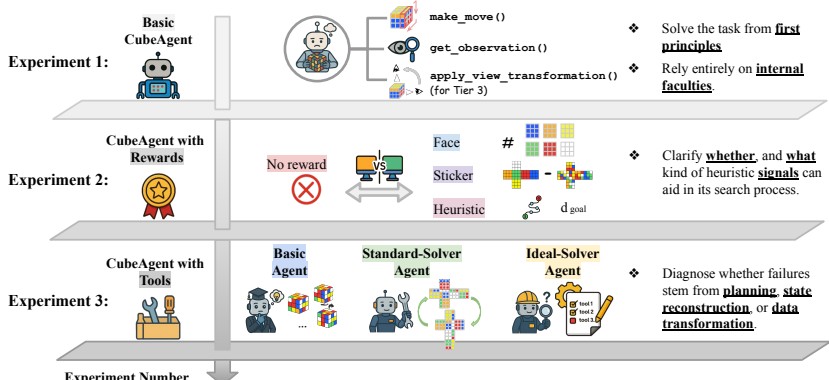

Figure 5: **Visualization of our three-part diagnostic framework for systematically evaluating LLM agents.** To answer **Q1**, we test a basic agent with only fundamental interaction tools to establish its baseline capabilities from first principles. For **Q2**, we augment the agent with various dense reward signals to determine if external feedback can effectively guide its search process. Finally, to address **Q3**, we deploy agents with different levels of tool support to diagnose whether failures originate from high-level planning, state reconstruction, or procedural data transformation.

Table 2: Baseline performance across modalities and horizons on CubeBench. Top row: metric groups (**Pass rate**, **#MM**, where #MM is the average number of make_move calls); second row: observation modalities; third row: task horizons (Short = S, depths 1–4; Long = L, depths 8, 12, 16, 20). **Tier 3** denotes the hardest split, evaluated under two projections: *Face View* and *Vertex View*. Blue shading denotes open-source models, and pink denotes proprietary models. For each metric column, we shade the top-3 entries (**red** = 1st, **orange** = 2nd, **yellow** = 3rd). We also train an MLP with policy gradient on the Full Symbolic setting; details are in Sec. H. "−" : Model does not support visual inputs.

| | Pass rate | | | | | | | | #MM | | | | | | | |
| | Full Symbolic | | Full Visual | | Face view | | Vertex view | | Full Symbolic | | Full Visual | | Face view | | Vertex view | |
| Model | S | L | S | L | S | L | S | L | S | L | S | L | S | L | S | L |
|---|---|---|---|---|---|---|---|---|---|---|---|---|---|---|---|---|
| GPT-5 | 0.75 | 0.00 | 0.20 | 0.00 | 0.40 | 0.00 | 0.05 | 0.00 | 85869.16 | 438193.56 | 387.42 | 376.47 | 161.65 | 189.00 | 8773.00 | 5574.45 |
| MLP (Policy Gradient) | 0.75 | 0.00 | – | – | – | – | – | – | 5.17 | 400.00 | – | – | – | – | – | – |
| gpt-oss-120b | 0.20 | 0.00 | – | – | – | – | – | – | 115585.30 | 197923.80 | – | – | – | – | – | – |
| Grok-4 | 0.20 | 0.00 | 0.05 | 0.00 | 0.00 | 0.00 | 0.00 | 0.00 | 3.75 | 9.45 | 3.35 | 60.00 | 3.45 | 3.25 | 42.00 | 9.75 |
| Kimi K2 (2024-09-05) | 0.15 | 0.00 | – | – | – | – | – | – | 47446.30 | 1297.06 | – | – | – | – | – | – |
| Gemini 2.5 Pro | 0.10 | 0.00 | 0.05 | 0.00 | 0.05 | 0.00 | 0.00 | 0.00 | 180.50 | 114.45 | 36.15 | 36.25 | 7.60 | 8.25 | 27.35 | 25.65 |
| DeepSeek-R1 (2025-05-28) | 0.05 | 0.00 | – | – | – | – | – | – | 28143.85 | 37819.20 | – | – | – | – | – | – |
| Claude Sonnet 4 | 0.05 | 0.00 | 0.00 | 0.00 | 0.00 | 0.00 | 0.00 | 0.00 | 69.10 | 126.75 | 27.55 | 23.20 | 6.70 | 25.10 | 19.30 | 65.35 |
| Qwen3-Max | 0.05 | 0.00 | – | – | – | – | – | – | 35.20 | 43.35 | – | – | – | – | – | – |
| DeepSeek-V3.1 | 0.05 | 0.00 | – | – | – | – | – | – | 33.80 | 20.85 | – | – | – | – | – | – |
| doubao-seed-1-6-vision | 0.05 | 0.00 | 0.00 | 0.00 | 0.00 | 0.00 | 0.00 | 0.00 | 10.70 | 11.25 | 12.75 | 12.35 | 12.75 | 10.65 | 14.65 | 13.85 |
| InternVL-3 (78B) | 0.00 | 0.00 | 0.00 | 0.00 | 0.05 | 0.00 | 0.00 | 0.00 | 56499.75 | 61.15 | 48.45 | 49.75 | 42.90 | 38.11 | 62.00 | 79.95 |
| Qwen2.5-VL-72B-Instruct | 0.00 | 0.00 | 0.00 | 0.00 | 0.05 | 0.00 | 0.00 | 0.00 | 47390.10 | 51351.79 | 45.25 | 36.10 | 13.30 | 15.55 | 80.75 | 30.05 |
| kimi-vl-a3b-thinking | 0.00 | 0.00 | 0.00 | 0.00 | 0.00 | 0.00 | 0.00 | 0.00 | 5276.13 | 0.18 | 0.00 | 0.11 | 0.00 | 0.00 | 1.06 | 0.05 |
| GPT-4o | 0.00 | 0.00 | 0.00 | 0.00 | 0.10 | 0.00 | 0.00 | 0.00 | 83.90 | 104.10 | 106.70 | 104.50 | 50.40 | 59.00 | 118.45 | 115.35 |
| GLM-4.5V | 0.00 | 0.00 | 0.00 | 0.00 | 0.00 | 0.00 | 0.00 | 0.00 | 36.85 | 55.55 | 30.75 | 39.70 | 51.70 | 58.21 | 35.50 | 101.70 |
| Gemma-3-27B-IT | 0.00 | 0.00 | 0.00 | 0.00 | 0.00 | 0.00 | 0.00 | 0.00 | 22.60 | 18.25 | 30.30 | 26.10 | 19.95 | 20.30 | 23.60 | 16.10 |
| Seed-OSS-36B-Instruct | 0.00 | 0.00 | – | – | – | – | – | – | 14.68 | 10.94 | – | – | – | – | – | – |

performance hierarchy is evident, with GPT-5's 0.75 pass rate on the symbolic task significantly exceeding all other models. A Policy Gradient agent matches GPT-5's performance, outperforming most LLMs in this setting. **(4)** On the Full Symbolic task, a subset of models engage in computationally intensive search, indicated by average #MM counts several orders of magnitude higher than other models.

**Insights.** **(1)** The universal failure on long-horizon tasks is direct evidence of a fundamental deficit in *Long-Horizon State Tracking through Mental Simulation*. A related case study is presented in Sec. L.2. **(2)** The sharp performance decline from symbolic to visual inputs indicates that *Visual Thinking* is a primary limiting factor for these agents. **(3)** While symbolic inputs enable search-based strategies, they are often computationally expensive. A notable phenomenon emerges in these tasks: agents exhibit a diversity of problem-solving strategies. Lower #MM values typically correspond to directly reasoning through the sequence of moves logically, whereas higher #MM values are indicative of search-based strategies. The choice of searching algorithm substantially impacts pass rates. More capable agents, such as GPT-5, tend to systematically search using algorithms like beam search and iterative deepening depth-first search (IDDFS) with skills like backtracking, as shown in

Table 3: Pass rates of different agent types across modalities and horizons on CubeBench. Metrics include **Pass rate** (higher is better). Modalities: `Full Symbolic`, `Full Visual`, `Face View`, `Vertex View`. Tier 3 denotes the hardest split and is evaluated under two projections: `Face View` and `Vertex View`. Horizons: Short (S) and Long (L).

| Model | Reward Type | Full Symbolic | | Full Visual | | Face view | | Vertex view | |
|---|---|---|---|---|---|---|---|---|---|
| | | S | L | S | L | S | L | S | L |
| GPT-5 | no reward | 0.75 | 0.00 | 0.20 | 0.00 | 0.40 | 0.00 | 0.05 | 0.00 |
| | face | 0.85 | 0.00 | 0.55 | 0.00 | 0.50 | 0.00 | 0.40 | 0.00 |
| | sticker | 0.65 | 0.00 | 0.55 | 0.00 | 0.55 | 0.00 | 0.50 | 0.00 |
| | heuristic | 0.50 | 0.00 | 0.45 | 0.00 | 0.65 | 0.00 | 0.30 | 0.00 |
| Gemini 2.5 Pro | no reward | 0.10 | 0.00 | 0.05 | 0.00 | 0.05 | 0.00 | 0.00 | 0.00 |
| | face | 0.00 | 0.00 | 0.00 | 0.00 | 0.00 | 0.00 | 0.00 | 0.00 |
| | sticker | 0.10 | 0.00 | 0.00 | 0.00 | 0.05 | 0.00 | 0.00 | 0.00 |
| | heuristic | 0.05 | 0.00 | 0.00 | 0.00 | 0.10 | 0.00 | 0.00 | 0.00 |
| Claude Sonnet 4 | no reward | 0.05 | 0.00 | 0.00 | 0.00 | 0.00 | 0.00 | 0.00 | 0.00 |
| | face | 0.10 | 0.00 | 0.10 | 0.00 | 0.05 | 0.00 | 0.00 | 0.00 |
| | sticker | 0.25 | 0.00 | 0.15 | 0.00 | 0.00 | 0.00 | 0.05 | 0.00 |
| | heuristic | 0.20 | 0.00 | 0.05 | 0.00 | 0.05 | 0.00 | 0.10 | 0.00 |

Sec. L.1 and L.2. In contrast, less capable agents often devolve into largely unguided enumeration (shown in L.3). However, even models such as GPT-5 struggle to perform effective pruning; their capabilities remain insufficient to curb the rapid growth in computational complexity, leading to failures on long-horizon tasks (shown in L.4).

## 4.2 EXPERIMENT 2: CUBEAGENT WITH REWARDS

**Experimental Setup.** Our second experiment was designed to measure the impact of different dense reward mechanisms on agent performance. We used the ***Basic Agent*** agent configuration as the testbed. Its performance was evaluated under four distinct conditions: a baseline with no progressive feedback (`no reward`), and three conditions providing different dense reward signals (`face`, `sticker`, and `heuristic`), which are introduced in Sec. 3.1.4 and used as the return value for the `make_move` function. This direct comparison aims to clarify whether, and what kind of dense rewardss can aid the agent. The results are presented in Table 3.

**Key Observations. (1)** On short-horizon tasks, dense rewards generally lead to an increase in pass rates. **(2)** The pass rate on all long-horizon tasks remains at 0.00, regardless of the presence or type of dense reward. **(3)** The impact of rewards is inconsistent; in some cases, such as for GPT-5 on the `Full Symbolic` task with `heuristic` or `sticker` rewards, performance is lower than the no-reward baseline. **(4)** The ability to leverage rewards varies notably across models.

**Insights. (1)** Dense rewards can guide an agent's search on short-horizon tasks by providing a local heuristic guide. **(2)** The failure of rewards on long-horizon tasks indicates that local feedback cannot compensate for a fundamental deficit in long-horizon state tracking. **(3)** On visual inputs, agents may leverage reward signals through symbolic reasoning, bypassing genuine visual reasoning, as shown in Sec. L.5. **(4)** For more capable agents like GPT-5, an external reward can potentially conflict with their emergent internal strategies, leading to suboptimal performance. For less capable agents that may lack a strong internal strategy, any form of guidance from a dense reward is often helpful, as seen with Claude Sonnet 4. The case studies are presented in Sec. L.6 and L.7 respectively.

## 4.3 EXPERIMENT 3: CUBEAGENT WITH SOLVER TOOLS

**Experimental Setup.** To precisely identify the primary bottlenecks in agent performance, our final experiment removes the burden of long-horizon planning by equipping agents with an optimal solver. We introduce two distinct configurations to isolate different cognitive challenges: the ***Standard-Solver Agent*** and the ***Ideal-Solver Agent***. The ***Standard-Solver Agent*** is given a solver that requires a specific, strict symbolic input format. To succeed, this agent must first accurately perceive the cube's state, then perform the crucial step of **translating** that perception into the required format. *This translation process is non-trivial, as it requires spatial understanding to reconcile potential differences between the environment's state representation (e.g., one type of 2D unfolded map) and the solver's expected input (e.g., a different face order or vertex numbering scheme).* Fi-

Table 4: Comparison of pass rates for *Basic*, *Standard-Solver*, and *Ideal-Solver* agent configurations. Modalities: `Full Symbolic`, `Full Visual`, `Face View`, `Vertex View`. Tier 3 denotes the hardest split and is evaluated under two projections: `Face View` and `Vertex View`. Horizons: Short (S) and Long (L).

| Model | Agent Type | Full Symbolic | | Full Visual | | Face view | | Vertex view | |
|---|---|---|---|---|---|---|---|---|---|
| | | **S** | **L** | **S** | **L** | **S** | **L** | **S** | **L** |
| **GPT-5** | Basic | 0.75 | 0.00 | 0.20 | 0.00 | 0.40 | 0.00 | 0.05 | 0.00 |
| | Standard-Solver | 0.95 | 0.95 | 0.65 | 0.70 | 1.00 | 0.95 | 0.00 | 0.00 |
| | Ideal-Solver | 1.00 | 1.00 | 0.95 | 0.80 | 0.85 | 1.00 | 0.00 | 0.00 |
| **Gemini 2.5 Pro** | Basic | 0.10 | 0.00 | 0.05 | 0.00 | 0.05 | 0.00 | 0.00 | 0.00 |
| | Standard-Solver | 0.70 | 0.65 | 0.25 | 0.00 | 0.20 | 0.00 | 0.00 | 0.00 |
| | Ideal-Solver | 1.00 | 1.00 | 0.25 | 0.00 | 0.00 | 0.00 | 0.00 | 0.00 |
| **Claude Sonnet 4** | Basic | 0.05 | 0.00 | 0.00 | 0.00 | 0.00 | 0.00 | 0.00 | 0.00 |
| | Standard-Solver | 0.35 | 0.85 | 0.00 | 0.00 | 0.00 | 0.00 | 0.00 | 0.00 |
| | Ideal-Solver | 1.00 | 1.00 | 0.00 | 0.00 | 0.00 | 0.00 | 0.00 | 0.00 |

nally, the agent must execute the solver's plan. This setup tests the agent's ability to handle state reconstruction, spatial transformation, and procedural tool use.

In contrast, the ***Ideal-Solver Agent*** is provided with a more advanced tool that automates the translation step. This agent can directly pass its perceived state to the solver, thus bypassing the data formatting challenge. By comparing the performance of these two agents, we can isolate whether failures stem from reconstructing a state from perception or from the challenge of spatial understanding during translation. The results are presented in Table 4.

**The Diagnosing Framework.** Our evaluation is structured as a progressive, three-step diagnostic process designed to systematically isolate and assess the core cognitive faculties of an agent. ***(1) Diagnosing Long-Horizon State Tracking and Planning.*** We first diagnose long-horizon tracking by comparing the *Basic Agent* with the *Standard-Solver Agent* on long-horizon symbolic tasks. The Basic Agent relies on internal reasoning, while the Standard-Solver outsources the planning challenge to an optimal tool. Their performance gap reveals the agent's intrinsic planning capability. ***(2) Diagnosing Spatial Reasoning and Procedural Tool Use.*** Next, we diagnose the spatial reasoning required for tool use by comparing the *Standard-Solver Agent* to the *Ideal-Solver Agent*. Since both agents offload planning, the performance gap isolates the challenge of spatial thinking, which translates perceptual input into a usable format for the tool. ***(3) Diagnosing Active Exploration under Partial Observation.*** Finally, to isolate exploration, we evaluate the *Ideal-Solver Agent* in a partial observation setting. The ideal tool removes both planning and translation challenges, leaving only the task of reconstructing a complete world model from fragmented information. Success here depends entirely on the agent's ability to actively explore its environment.

**Key Observations.** **(1)** The addition of the tools generally leads to marked performance gains compared to the basic agent. **(2)** There is still a performance gap between the Standard-Solver and Ideal-Solver agents. **(3)** On `Full Visual` and `Face view` tasks, only GPT-5 maintains strong performance, while all models fail universally on the `Vertex view` task.

**Insights. (1)** High-level, multi-step planning, or **Long-Horizon State Tracking**, is a primary deficit that can be successfully offloaded to external solvers. **(2)** The procedural challenge of using tools is non-trivial, rendering **Spatial Reasoning** an important challenge to resolve for further development. **(3)** An unanticipated but noteworthy finding is the emergence of tool-learning strategies in the Standard-Solver Agent. In some instances, we observed a remarkable capability for autonomous tool-learning, where agents learn to master the tool through trial-and-error experimentation for this spatial conversation, as shown in Sec. L.8. **(4)** A significant performance gap exists between the `Face view` and `Vertex view` tasks. The reason is that the orderly, grid-like structure of the `Face view` allows agents to succeed by recasting the task as an algorithmic parsing problem. This indicates that models will attempt to bypass direct spatial reasoning in favor of a parsing-based approach whenever possible, and their performance suffers when the input's complexity, as in the `Vertex view`, makes this bypass strategy infeasible. The corresponding case study is presented in Sec. L.9.

## 5 CONCLUSION

In this work, we introduced CubeBench, a diagnostic benchmark designed to probe the cognitive faculties required for spatial reasoning. Our comprehensive experiments demonstrate a critical failure in current leading models, which uniformly achieve a zero pass rate on all long-horizon tasks and struggle to bridge the gap from visual perception to symbolic understanding. Our diagnostic framework successfully isolated these bottlenecks, confirming fundamental deficits in Spatial Reasoning, Long-Horizon State Tracking through Mental Simulation, and Exploration and Reasoning under Partial Observation. Our findings underscore the need for future research to focus on developing more robust spatial mental models and grounding agents in the principles of three-dimensional interaction to unlock their potential in the physical world.

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

## A  LLM Usage Statement

*Model.* The LLMs employed in our study are GPT-5 and Gemini 2.5 Pro.

*Scope.* We used large language models (LLMs) only as general-purpose assistants for language polishing, typo checking, and minor code boilerplate generation. LLMs did not contribute to research ideation or produce novel scientific claims, proofs, or results.

*Human oversight and verification.* All text and code produced with LLM assistance were reviewed, corrected, and verified by the authors. Experimental results were reproduced independently of any LLM outputs.

*Data governance.* We did not share proprietary or sensitive data with third-party services beyond materials already included in the anonymous submission artifacts.

*Attribution.* LLMs are not authors and bear no responsibility for the content; full responsibility lies with the paper's authors.

## B  What Exactly Does CubeBench Measure?

CubeBench is not intended to be a broad coverage benchmark that competes with large embodied suites or classical Rubik's Cube solvers. Instead, it serves as a *minimal, verifiable, factorized* diagnostic environment. By stripping away perception noise, multi-object dynamics, affordances, and actuation, it allows us to focus on three core cognitive abilities that repeatedly emerge as bottlenecks for LLM/MLLM agents in more complex settings: (i) 3D spatial reasoning, (ii) long-horizon non-commutative planning, and (iii) belief-state construction under partial observability.

Our three-tier design decouples these abilities by progressively increasing the burden placed on the agent's internal world model. Tier 1 (*Full Symbolic*) exposes a complete 54-character state, so all perception and grounding are provided by the environment; this setting primarily stresses non-commutative long-horizon planning and state tracking. Tier 2 (*Full Visual*) replaces the symbolic string with a 2D unfolded image, forcing the agent to construct its own symbolic representation from pixels: segmenting the cube, clustering colors, assigning stickers to faces, and mapping the 2D layout to a consistent 3D frame before any planning can begin. Tier 3 (*Partial Visual*) further restricts each observation to a single face or corner view, plus view-change actions, so the agent must actively explore, aggregate partial views over time, and maintain a coherent latent world state while its own actions continuously perturb the cube.

On top of these observation tiers, the different solver configurations act as *controls* that selectively remove specific difficulties. A *Basic Agent* must handle perception, internal state tracking, planning, and formatting. In the *Standard-Solver* setting, we outsource optimal planning to a Kociemba-based solver, so performance is driven mainly by visual $\rightarrow$ symbolic translation and correct schema formatting. The *Ideal-Solver* setting goes one step further by also hiding the solver's input schema, leaving only the requirement to output a correct symbolic cube state; any gap between Standard- and Ideal-Solver performance therefore isolates visual grounding and spatial mapping errors rather than planning or string-format issues.

This factorized design explains "what" CubeBench measures in contrast to existing planning and embodied benchmarks. Classical planning suites such as PlanBench (Valmeekam et al., 2023), SPIN-Bench (Yao et al., 2025), and ARC/ARC-AGI (Foundation, 2025) probe rich algorithmic and combinatorial structures, but they do not directly target the specific triad of 3D spatial reasoning, non-commutative dynamics, and partial-observation belief-state tracking in a single, fully verifiable physical system. Conversely, embodied 3D suites such as ALFRED (Shridhar et al., 2020), Habitat(Savva et al., 2019a; Szot et al., 2021; Puig et al., 2023), BEHAVIOR-100 (Srivastava et al., 2021), LogiCity (Li et al., 2025), EmbodiedBench (Yang et al., 2025b), and EAI (Li et al., 2024) place agents in visually rich, multi-object worlds with realistic physics and affordances, but necessarily entangle perception, control, and high-level reasoning. As several of these works themselves emphasize, final success rates in such environments make it difficult to pinpoint which cognitive ability has failed.

CubeBench occupies an orthogonal niche in this landscape. It uses a single rigid object with deterministic kinematics to provide a low-noise, automatically verifiable testbed where failures can

be crisply attributed: incorrect 3D world modeling, short effective planning horizons, or unstable belief-state tracking under partial observation. The negative results we obtain—universal collapse at depth 8, severe degradation from symbols to images, and near-zero success under partial views—mirror the failure modes reported in broader embodied benchmarks, but in a setting where they can be disentangled and systematically ablated. In this sense, CubeBench is best viewed as a *first diagnostic stop* before running expensive embodied evaluations: if an agent already fails to maintain a consistent 3D mental model and long-horizon plan in this simplified domain, it is unlikely to succeed in more complex physical worlds.

## C  REVISED BASELINE PERFORMANCE USING MOVE RATIOS

To account for the variation in task difficulty, we introduce the Number of Move Ratio (#MR) as a normalized measure of search efficiency. Optimal path lengths increase non-linearly with scramble depth, therefore we define #MR as the ratio of the agent's move count to the optimal solution length:

$$\text{\#MR (number of move ratio)} = \frac{\text{\#MM (number of } \texttt{make\_moves})}{depth \text{ (number of optimal moves)}}$$

Table 5: Baseline performance across modalities and horizons on CubeBench. Top row: metric groups (**Pass rate**, **#MR**, where #MR is the average number of move ratios; second row: observation modalities; third row: task horizons (Short = S, depths 1–4; Long = L, depths 8, 12, 16, 20). **Tier 3** denotes the hardest split, evaluated under two projections: *Face View* and *Vertex View*. Blue shading denotes open-source models, and pink denotes proprietary models. For each metric column, we shade the top-3 entries (**red** = 1st, **orange** = 2nd, **yellow** = 3rd). We also train an MLP with policy gradient on the Full Symbolic setting; details are in Sec. H. "−" : Model does not support visual inputs.

| | Pass rate | | | | | | | | #MR | | | | | | | |
|---|---|---|---|---|---|---|---|---|---|---|---|---|---|---|---|---|
| | Full Symbolic | | Full Visual | | Face view | | Vertex view | | Full Symbolic | | Full Visual | | Face view | | Vertex view | |
| Model | S | L | S | L | S | L | S | L | S | L | S | L | S | L | S | L |
| GPT-5 | 0.75 | 0.00 | 0.20 | 0.00 | 0.40 | 0.00 | 0.05 | 0.00 | 27124.66 | 33982.78 | 149.29 | 29.22 | 51.25 | 17.69 | 8378.55 | 672.57 |
| MLP (Policy Gradient) | 0.75 | 0.00 | – | – | – | – | – | – | 5.17 | 400.00 | – | – | – | – | – | – |
| gpt-oss-120b | 0.20 | 0.00 | – | – | – | – | – | – | 47173.21 | 14219.57 | – | – | – | – | – | – |
| Grok-4 | 0.20 | 0.00 | 0.05 | 0.00 | 0.00 | 0.00 | 0.00 | 0.00 | 1.84 | 0.97 | 1.76 | 5.28 | 1.81 | 0.25 | 18.36 | 0.84 |
| Kimi K2 (2024-09-05) | 0.15 | 0.00 | – | – | – | – | – | – | 23705.38 | 118.25 | – | – | – | – | – | – |
| Gemini 2.5 Pro | 0.10 | 0.00 | 0.05 | 0.00 | 0.05 | 0.00 | 0.00 | 0.00 | 59.01 | 12.55 | 19.30 | 3.13 | 4.05 | 0.55 | 13.28 | 1.85 |
| DeepSeek-R1 (2025-05-28) | 0.05 | 0.00 | – | – | – | – | – | – | 14074.89 | 4724.51 | – | – | – | – | – | – |
| Claude Sonnet 4 | 0.05 | 0.00 | 0.00 | 0.00 | 0.00 | 0.00 | 0.00 | 0.00 | 28.59 | 9.05 | 14.81 | 1.93 | 2.67 | 2.43 | 9.40 | 5.75 |
| Qwen3-Max | 0.05 | 0.00 | – | – | – | – | – | – | 15.92 | 3.48 | – | – | – | – | – | – |
| DeepSeek-V3.1 | 0.05 | 0.00 | – | – | – | – | – | – | 21.89 | 1.66 | – | – | – | – | – | – |
| doubao-seed-1-6-vision | 0.05 | 0.00 | 0.00 | 0.00 | 0.00 | 0.00 | 0.00 | 0.00 | 4.77 | 0.94 | 6.79 | 1.00 | 6.26 | 0.84 | 7.47 | 1.10 |
| InternVL-3 (78B) | 0.00 | 0.00 | 0.00 | 0.00 | 0.05 | 0.00 | 0.00 | 0.00 | 56466.74 | 4.85 | 23.27 | 3.95 | 18.16 | 2.61 | 29.14 | 6.84 |
| Qwen2.5-VL-72B-Instruct | 0.00 | 0.00 | 0.00 | 0.00 | 0.05 | 0.00 | 0.00 | 0.00 | 43712.71 | 2575.65 | 18.26 | 2.54 | 5.97 | 1.26 | 27.28 | 2.44 |
| kimi-vl-a3b-thinking | 0.00 | 0.00 | 0.00 | 0.00 | 0.00 | 0.00 | 0.00 | 0.00 | 2623.57 | 0.01 | 0.00 | 0.01 | 0.00 | 0.00 | 0.36 | 0.00 |
| GPT-4o | 0.00 | 0.00 | 0.00 | 0.00 | 0.10 | 0.00 | 0.00 | 0.00 | 48.10 | 8.01 | 52.67 | 8.48 | 24.85 | 4.55 | 55.41 | 9.41 |
| GLM-4.5V | 0.00 | 0.00 | 0.00 | 0.00 | 0.00 | 0.00 | 0.00 | 0.00 | 16.61 | 5.44 | 15.48 | 3.35 | 28.49 | 5.17 | 21.08 | 8.33 |
| Gemma-3-27B-IT | 0.00 | 0.00 | 0.00 | 0.00 | 0.00 | 0.00 | 0.00 | 0.00 | 12.29 | 1.56 | 15.49 | 2.09 | 9.38 | 1.64 | 11.49 | 1.21 |
| Seed-OSS-36B-Instruct | 0.00 | 0.00 | – | – | – | – | – | – | 9.22 | 0.93 | – | – | – | – | – | – |

## D  THE NO-CODE EXPERIMENT

**Experimental Setup.** In our primary evaluation (the *Code* setting, which corresponds to the ***Basic Agent*** configuration in Experiment 1), agents are permitted to write and execute Python code. While writing code to perform search algorithms is a valid problem-solving strategy in the agentic era, we acknowledge that distinguishing intrinsic spatial reasoning from programmatic search is crucial. To isolate the model's internal state tracking and planning capabilities from code-based search, we introduce a *No-Code* evaluation mode. In this mode, the agent must output a pre-planned sequence (e.g., `"moves":['D'', 'B', 'B']`) without the ability to manage control flows during tool use. This mode is more consistent with common tool-use interfaces such as the OpenAI API. We conducted an ablation study comparing the *No-Code* and *Code* settings on the Short-Horizon `Full Symbolic` tasks. The results are presented in Table 6.

**Key Observations & Insights.** (1) Models employing intensive search strategies (High #MR) suffered catastrophic declines. Both GPT-5 (0.75 to 0.25) and gpt-oss-120b (0.20 to 0.00) saw their performance evaporate as their search volume collapsed (e.g., GPT-5 #MR: ∼27k to ∼72; gpt-oss-120b #MR: ∼47k to ∼14). This confirms that for these agents, the code-based search compensates

for limited internal planning. (2) Models with initially low search volume (Low #MR) displayed mixed outcomes rather than a uniform drop. Grok-4 improved (0.20 to 0.30) in the No-Code setting, while many remained poor.

Table 6: Comparison of agent performance in *No-Code* vs. *Code* settings on Short-Horizon `Full Symbolic` tasks. #MR is averaged over testcases.

| | No Code | | Code | |
| Model | Pass Rate | #MR | Pass Rate | #MR |
|---|---|---|---|---|
| GPT-5 | 0.25 | 72.79 | **0.75** | 27124.66 |
| gpt-oss-120b | 0.00 | 14.20 | 0.20 | 47173.21 |
| Grok-4 | **0.30** | 2.02 | 0.20 | 1.84 |
| Kimi-K2-0905 | 0.00 | 55.15 | 0.15 | 23705.38 |
| Gemini 2.5 Pro | 0.15 | 18.68 | 0.10 | 59.01 |
| Claude Sonnet 4 | 0.00 | 68.03 | 0.05 | 28.59 |
| DeepSeek-V3.1 | 0.00 | 17.11 | 0.05 | 21.89 |
| DeepSeek-R1-0528 | 0.00 | 12.28 | 0.05 | 14074.89 |
| Qwen3-Max | 0.00 | 18.11 | 0.05 | 15.92 |
| GPT-4o | 0.05 | 61.53 | 0.00 | 48.10 |
| GLM-4.5v | 0.00 | 4.31 | 0.00 | 16.61 |

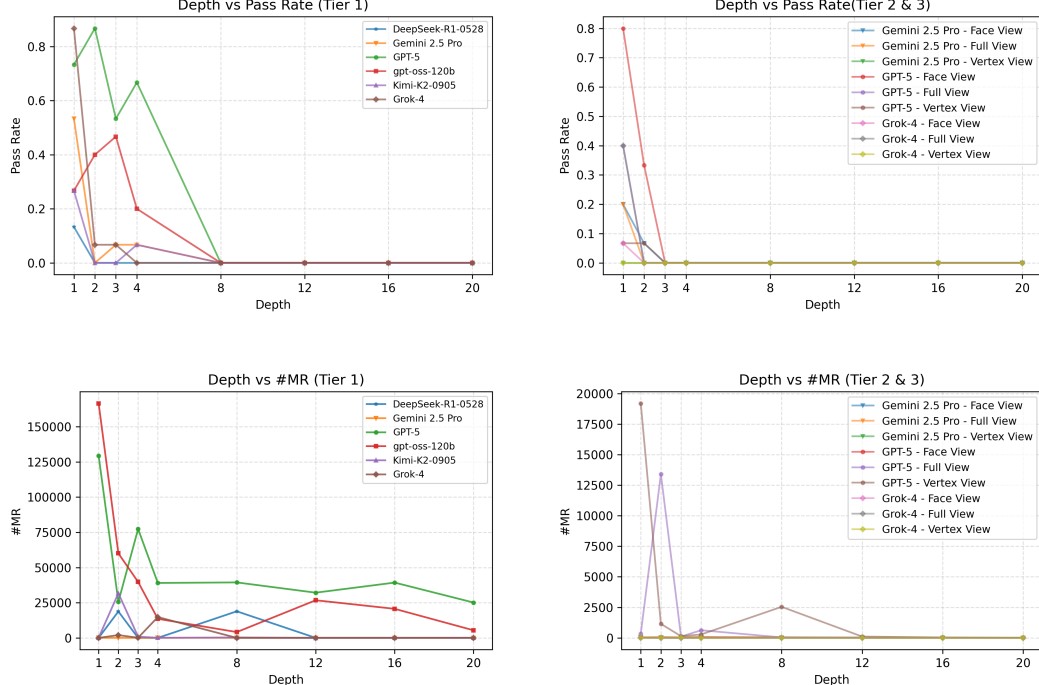

Figure 6: **Pass rates and average number of move ratios (#MR) across varying depths.** Top row: Pass Rates for Tier 1 (Left) and Tier 2/3 (Right). Bottom row: Corresponding #MR analysis.

# E DEPTH-WISE METRICS

**Experimental Setting.** We adopt the experimental protocol from Experiment 1, extending the evaluation to three independent trials per test case to ensure robustness. For this analysis, we mainly focus exclusively on the top-performing models in Experiment 1. The comparative results are presented in Figure 6.

**Key Observations & Insights.**

(1) *Depth-vs-Performance.* In the Tier 1 setting, top-tier models capable of systematic search (e.g., GPT-5) maintain high accuracy across Depths 1–4, , with pass rates decaying gracefully. Conversely, less capable models (e.g., Grok-4) succeed at Depth 1 via direct policy but degrade immediately at Depth 2. In Tier 2/3 settings, the performance cliff arrives earlier. While agents can manage Depth 1 tasks, we observe a drop starting at just Depth 2. A universal limit appears between Depth 4 and 8. Pass rates drop to 0.00 at Depth 8 across all settings, confirming that current planning capabilities fail to track state over extended horizons.

(2) *Depth-vs-#MR.* We interpret the #MR metric not as a measure of "effort" scaling linearly with difficulty, but as an indicator of cognitive mode. A low move count typically signals reliance on direct intuition or trivial heuristics, while a high move count signals the activation of explicit search. The switch is often a discrete jump rather than a gradual increase. A close examination of execution traces reveals how agents dynamically adapt within a single trial. Typically, search-based agents begin with low-cost heuristics, switching to high-volume search when those initial attempts fail. In rare instances (e.g., Appendix L.2), we observe a further strategic shift: abandoning search entirely to implement structured human algorithms, such as the Beginner's Method (see Appendix J), as a fallback.

## F   RELATED WORKS (CONT.)

**Spatial Cognition.** The ability to reason about three-dimensional space (Li et al., 2023; Pi et al., 2024; Alayrac et al., 2022), a cornerstone of intelligence, relies on an internal ***spatial mental model*** (Johnson-Laird, 1980; 1983) to infer unseen properties and predict the consequences of actions. (Yang et al., 2025a; Zhang et al., 2024c) This concept is rooted in cognitive science and has been a long-standing goal, with specialized systems like SLAM in robotics (Aulinas et al., 2008) and NeRFs (Mildenhall et al., 2021) in computer vision designed to construct explicit 3D representations. However, the intrinsic ability of Large Language Models (LLMs) (Bai et al., 2025; Hurst et al., 2024; Chen et al., 2024a), which excel at sequential data, to form and manipulate such spatial models remains a critical open question. Recent works have begun to probe this (Xu et al., 2025a; Zhang et al., 2024b; Chen et al., 2025c; Qi et al., 2025); for instance, MindCube (Liu et al., 2024) evaluates an agent's ability to reason about a *static* 3D scene by reasoning about its complete layout from a few partial viewpoints. While also leveraging partial observations, CubeBench introduces the distinct challenge of long-horizon interaction with a ***dynamic*** cube. Our work therefore shifts the focus from reasoning about static perspectives to the more complex challenge of ***updating*** a mental model through direct, state-altering interaction with the environment, emphasizing mental simulation and long-horizon state tracking.

## G   CONFIGURATION OF TEST SPLIT

Our evaluation set is constructed across eight distinct levels of task difficulty. These levels are determined by the state's "depth", which is the optimal number of moves required to solve a given cube configuration. For each of the eight difficulty levels, we sample five unique initial states. Each state is then tested across four different environment settings derived from the three-tiered framework: Full Symbolic (Tier 1), Full Visual (Tier 2), and the two Partial Visual modalities (Face View and Vertex View) from Tier 3. This methodology yields a total of 160 unique evaluation configurations (8 difficulties × 5 states × 4 settings) for each Large Language Model (LLM) agent. To provide an estimate of the resources required, a single, complete evaluation run of these 160 configurations on GPT-5 consumes a total of 59.3 million tokens (50.2 million input tokens and 9.1 million output tokens), resulting in a total cost of approximately $153, based on the pricing of $1.25 per million input tokens and $10 per million output tokens.

## H   INTRODUCTION TO THE POLICY GRADIENT BASELINE

As a baseline for comparison in the `Full Symbolic` setting, we implement a classic reinforcement learning agent based on the Policy Gradient (PG) method. Policy Gradient algorithms directly optimize the parameters of a policy by estimating the gradient of the expected return. The core idea is to adjust the policy's parameters to increase the probability of taking actions that lead to higher

cumulative rewards. Our implementation uses the REINFORCE algorithm, a foundational Monte Carlo policy gradient method.

The objective of the REINFORCE algorithm is to maximize the expected total discounted reward, $J(\theta)$, by updating the policy parameters $\theta$ in the direction of the gradient $\nabla_\theta J(\theta)$. The policy gradient theorem provides an estimate for this gradient:

$$\nabla_\theta J(\theta) = \mathbb{E}_{\tau \sim \pi_\theta} \left[ \sum_{t=0}^{T} \nabla_\theta \log \pi_\theta(a_t|s_t) G_t \right]$$

where $\pi_\theta(a_t|s_t)$ is the policy (the probability of taking action $a_t$ in state $s_t$), and $G_t = \sum_{k=t}^{T} \gamma^{k-t} r_k$ is the total discounted return from time step $t$ onward.

Our specific implementation utilizes a Multi-Layer Perceptron (MLP) to represent the policy network. The training is conducted using a curriculum learning strategy, where the agent is progressively trained on more difficult tasks by increasing the scramble length of the cube. The detailed configuration is as follows:

- **Algorithm:** REINFORCE
- **Policy Network:** A five-layer MLP with 256 neurons per layer and ReLU activation functions.
- **Optimizer:** Adam with a learning rate of $5 \times 10^{-4}$.
- **Discount Factor ($\gamma$):** 0.99.
- **Training Environment:** We use 64 parallel vectorized environments for efficient data collection.
- **Update Rule:** The policy is updated after collecting a rollout of 512 steps from each parallel environment, using a batch size of 512 episodes for the gradient update.
- **Curriculum:** Training uses the curriculum learning for better convergence. We train two separate models, one for short-horizon tasks and one for long-horizon. For short-horizon tasks, the model is trained sequentially on scramble depths of 1, 2, 3, and 4, with the number of training timesteps scaled quadratically for each level (e.g., from 40,000 for depth 1 up to 320,000 for depth 4). For long-horizon tasks, a separate model is initialized from the converged short-horizon agent and then continues training on the more challenging depths from 5 to 20 with 320K steps each level.
- **Max Number of Make-moves During Evaluation:** For short-horizon tasks, the maximum number of making moves is set to 16. For long-horizon ones, they are set to 400.

## I  Tools for Agents

### I.1  Fundamental Interaction Tools

**make_move**: This tool executes a single face rotation. It accepts one of 12 possible inputs corresponding to standard Singmaster notation: F (Front), B (Back), L (Left), R (Right), U (Up), and D (Down) for clockwise turns. A prime symbol (′) denotes a counter-clockwise rotation (e.g., F′). Each call to this tool deterministically alters the cube's internal state. Since the agent can generate custom scripts in the Code block, this function can be called multiple times (e.g., in a loop) within a single step to execute a sequence of moves.

**get_observation**: This tool retrieves the current observation of the cube. The format of the returned data is contingent upon the experimental tier:

- In **Tier 1 (Full Symbolic State)**, it returns a 54-character string that symbolically represents the complete state of the cube.
- In **Tier 2 (Full Visual State)** and **Tier 3 (Partial Visual State)**, it returns an image. Depending on the specific task configuration, this can be a complete 2D unfolded map of the cube, an image of a single face (*face view*), or an image from a corner's perspective (*vertex view*).

**apply_view_transformation**: Exclusive to the **Tier 3 (Partial Visual State)** setting, this tool allows the agent to alter its viewpoint (e.g., up, down, left, right). This capability is essential for actively exploring the cube to reconstruct its full state from a series of limited views.

## I.2 AUXILIARY SOLVER TOOLS

**StandardSolverTool** and **IdealSolverTool**: These tools provide the agent with access to a solver based on Kociemba's two-phase algorithm (see Appendix K). They differ in a crucial aspect related to data formatting. The underlying solver requires a specific input format that is distinct from the state representation provided by the get_observation tool.

- The StandardSolverTool requires the agent to perform the necessary format conversion itself, thus testing its ability to transform data into a usable representation.

- The IdealSolverTool features a built-in converter. It allows the agent to input the state in the environment's native format and receive a solution directly, thereby bypassing the format conversion challenge.

## J HEURISTIC ALGORITHM FOR SOLVING THE RUBIK'S CUBE: LAYER-BY-LAYER APPROACH

This subsection formalizes the heuristic algorithm for solving the 3×3 Rubik's Cube using the layer-by-layer (LBL) method. We drew on the method from *solvethecube* website [2] and adopted its illustrations. The cube-solving process is divided into seven major steps, each focusing on solving specific parts of the cube. Each step involves operations or algorithms that manipulate specific pieces, progressively solving the puzzle. As introduced in our discussion on reward functions (Sec. 3.1.4), this LBL structure forms the basis of our Heuristic Metric ($\phi_{heuristic}$). To calculate a score for any given cube state $s$, we evaluate it against the seven steps of the LBL method. The state's score, $\phi_{heuristic}(s)$, is defined as the highest step number (from 0 for a scrambled cube to 7 for a solved one) that the configuration has successfully completed.

**Overall Strategy**   The solving process begins with the initial scrambled state $S_0$ and progresses through seven steps until the solved state $S^*$ is achieved. Each transformation corresponds to a specific phase of the solution:

$$S_0 \xrightarrow{f_1} S_1 \xrightarrow{f_2} S_2 \xrightarrow{f_3} \cdots \xrightarrow{f_7} S^*$$

where:

- $f_1$: Forming the bottom cross.
- $f_2$: Positioning the bottom corners.
- $f_3$: Solving the second layer edges.
- $f_4$: Creating the top cross.
- $f_5$: Permuting the top edges.
- $f_6$: Positioning the top corners.
- $f_7$: Orienting the top corners.

Each step is represented by an algorithm or set of moves that solves a specific portion of the cube without disrupting previously solved sections. See Fig. 7 for an intuitive schematic.

**Step 1: Forming the Bottom Cross**   The first task is to form a cross on the bottom layer by positioning the four edge pieces such that their colors match both the bottom face center and the adjacent side centers. This step can be formalized as:

---

[2]URL:https://solvethecube.com/

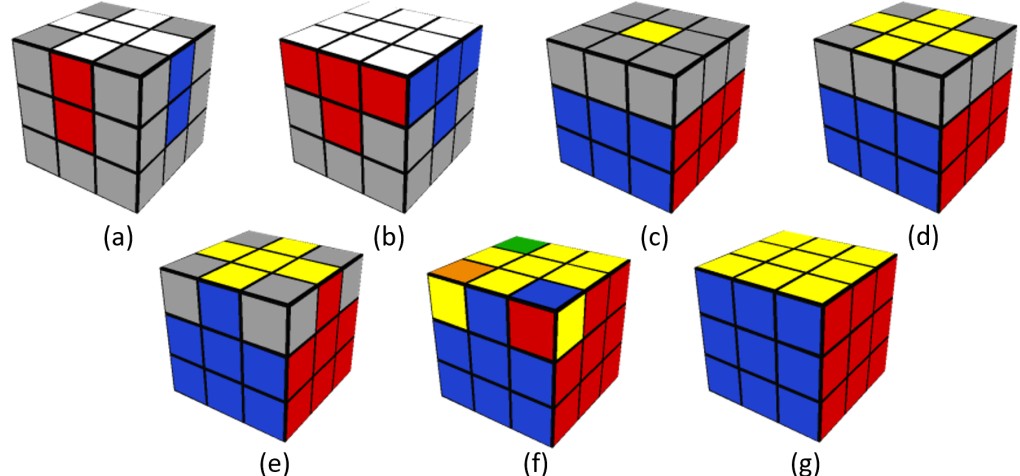

Figure 7: The diagram from solvethecube website illustrates a heuristic method for solving a Rubik's Cube: (a) Forming the bottom cross, (b) Positioning the bottom corners, (c) Solving the second layer edges, (d) Creating the top cross, (e) Permuting the top edges, (f) Positioning the top corners, and (g) Orienting the top corners.

$$f_{\text{cross}} : C_{\text{bottom}} \rightarrow C^*_{\text{bottom}}$$

where $C_{\text{bottom}} = \{E_1, E_2, E_3, E_4\}$ represents the four edge pieces to be positioned, and $C^*_{\text{bottom}}$ represents the state where the bottom cross is correctly formed. The process involves identifying the edge pieces and applying algorithms to move them into place.

**Step 2: Positioning the Bottom Corners**  After the bottom cross is formed, the next objective is to position the bottom corner pieces. Let $C_{\text{bottom-corner}} = \{C_1, C_2, C_3, C_4\}$ represent the four corner pieces to be positioned in the bottom layer. This step is formalized as:

$$f_{\text{corners}} : C_{\text{bottom-corner}} \rightarrow C^*_{\text{bottom-corner}}$$

where $C^*_{\text{bottom-corner}}$ represents the correctly positioned bottom corners. The operation involves applying specific algorithms to move each corner piece into its correct location without disturbing the already solved bottom cross.

**Step 3: Solving the Second Layer Edges**  The next step is to solve the edges of the second layer. Let $C_{\text{second-layer-edge}} = \{E_5, E_6, E_7, E_8\}$ represent the four edge pieces that need to be positioned in the middle layer. The operation can be formalized as:

$$f_{\text{second-layer}} : C_{\text{second-layer-edge}} \rightarrow C^*_{\text{second-layer-edge}}$$

where $C^*_{\text{second-layer-edge}}$ represents the state where the second layer edges are correctly positioned. The goal is to move the edge pieces from the top layer to the second layer, maintaining the solved bottom layer.

**Step 4: Creating the Top Cross**  After solving the second layer, the next objective is to create a cross on the top layer. Let $C_{\text{top-edge}} = \{E_9, E_{10}, E_{11}, E_{12}\}$ represent the four edge pieces that need to be positioned on the top layer. This step can be formalized as:

$$f_{\text{top-cross}} : C_{\text{top-edge}} \rightarrow C^*_{\text{top-edge}}$$

where $C^*_{\text{top-edge}}$ represents the state where the top layer edges are correctly aligned. The transformation for each edge piece is represented by:

$$E_i \xrightarrow{\Delta_{E_i}} E_i^* \quad \text{for} \quad i \in \{9, 10, 11, 12\}$$

The goal is to position the edge pieces correctly on the top layer without disturbing the solved portions of the bottom and second layers.

**Step 5: Permuting the Top Edges**   Once the top cross is formed, the next objective is to permute the top layer edge pieces into their correct positions. This operation can be formalized as:

$$f_{\text{permute-edges}} : C^*_{\text{top-edge}} \to C^{**}_{\text{top-edge}}$$

where $C^{**}_{\text{top-edge}}$ represents the state where the top layer edges are correctly permuted. The goal is to apply specific algorithms that permute the top edges into their correct positions.

**Step 6: Positioning the Top Corners**   After the top edges are permuted, the next step is to position the top layer corners. Let $C_{\text{top-corner}} = \{C_{13}, C_{14}, C_{15}, C_{16}\}$ represent the four top corner pieces. This operation can be formalized as:

$$f_{\text{position-corners}} : C_{\text{top-corner}} \to C^*_{\text{top-corner}}$$

where $C^*_{\text{top-corner}}$ represents the state where the top corners are positioned correctly. The goal is to apply specific algorithms to move the top layer corner pieces into their correct positions.

**Step 7: Orienting the Top Corners**   The final step is to orient the top corners, ensuring that the top face becomes uniform in color. This operation can be formalized as:

$$f_{\text{orient-corners}} : C^*_{\text{top-corner}} \to S^*$$

where $S^*$ represents the solved state of the Rubik's Cube. The transformation for each corner piece is represented by:

$$C_i \xrightarrow{\Delta_{C_i}} C_i^* \quad \text{for} \quad i \in \{13, 14, 15, 16\}$$

The goal is to apply specific algorithms to orient the top corners without disturbing the already solved portions of the cube.

**Mathematical Summary of the Layer-by-Layer Approach**   The Rubik's Cube solution can be mathematically summarized as a series of state transformations:

$$S_0 \xrightarrow{f_{\text{cross}}} S_1 \xrightarrow{f_{\text{corners}}} S_2 \xrightarrow{f_{\text{second-layer}}} S_3 \xrightarrow{f_{\text{top-cross}}} S_4 \xrightarrow{f_{\text{permute-edges}}} S_5 \xrightarrow{f_{\text{position-corners}}} S_6 \xrightarrow{f_{\text{orient-corners}}} S^*$$

Each transformation $f_{\text{cross}}, f_{\text{corners}}, \ldots, f_{\text{orient-corners}}$ corresponds to a specific set of moves that transform the cube toward the solved state. By applying these transformations in sequence, the cube is solved layer by layer.

## K   CUBE SOLVERS (TWO-PHASE AND OPTIMAL)

This appendix introduces the two solvers used in our benchmark: the *Two-Phase Solver* and the *Optimal Solver*. The two-phase solver is used as the basic component for `StandardSolverTool` and `IdealSolverTool`, while the optimal solver is used for generating the testcases. In the following , we first outline the mathematical background for the two-phase method, and then describe the I/O formats and the roles each solver plays in our experiments.

### K.1 INPUT AND OUTPUT FORMATS FOR CUBE STATE

To interact with solvers, the cube state must be expressed in a precise *facelet string* representation: a fixed ordering of all 54 stickers. Figure 8 illustrates the two indexing conventions used in our system: (1) the **Initial Format** used by our environment, and (2) the **Solver Format** required by the Kociemba two-phase solver.

**Initial Format.** The environment internally stores the cube as a 2D unfolded cross (see left of Fig. 8). Each sticker is labeled by its face and index, e.g. `U1`–`U9` for the Up face, `R1`–`R9` for the Right face, etc. Within each face, indices increase row by row from top-left to bottom-right:

$$U1, U2, U3, \ U4, U5, U6, \ U7, U8, U9,$$

and similarly for $R, F, D, L, B$.

The environment's *concatenation order* follows the visual "cross net" layout: first the Up face, then Left–Front–Right in a row, followed by Down, and finally the Back face. Explicitly, the 54-character string is constructed as

$$(F1, \ldots, F9, \ B1, \ldots, B9, \ L1, \ldots, L9, \ R1, \ldots, R9, \ U1, \ldots, U9, \ D1, \ldots, D9).$$

**Solver Format.** The two-phase solver requires the cube state as a 54-character string, concatenated in the strict order

$$(U1, \ldots, U9, \ R1, \ldots, R9, \ F1, \ldots, F9, \ D1, \ldots, D9, \ L1, \ldots, L9, \ B1, \ldots, B9).$$

Each character encodes the color on the corresponding facelet. For example, the string `UBL...` means: - position `U1` has the U-color, - position `U2` has the B-color, - position `U3` has the L-color, and so on. This flattened sequence is the standard *facelet string* convention used in Kociemba's solver.

**Conversion.** Figure 8 shows how the Initial Format is mapped into the Solver Format. This conversion step is crucial: any misalignment (e.g., rotated faces or incorrect indexing) produces invalid solver inputs and prevents the plan from being executed correctly. The Standard-Solver Agent must handle the format conversion by itself, whereas the Ideal-Solver Agent has the conversion built in `IdealSolverTool` and requires no further effort.

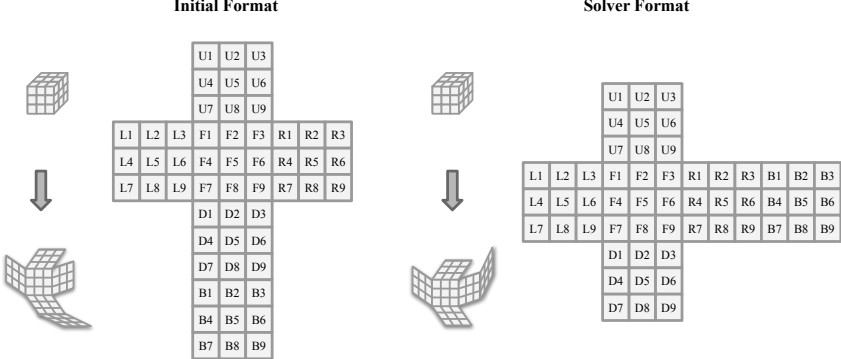

Figure 8: **Input/Output formats of cube state.** Left: Initial Format used in our environment. Right: Solver Format required by the two-phase solver. Both correspond to the same cube state but differ in indexing layout and concatenation order.

### K.2 TWO-PHASE SOLVER

**Mathematical Background.** Let $G_0$ denote the full Rubik's Cube group generated by face turns in the half-turn metric (HTM). Using standard Singmaster letters, a convenient presentation is

$$G_0 = \langle U, D, R, L, F, B \rangle$$
$$= \langle U, U', U^2,\ D, D', D^2,\ R, R', R^2, L, L', L^2,\ F, F', F^2,\ B, B', B^2 \rangle.$$

That is, quarter turns, their inverses, and half turns generate the same group.

The Two-Phase method (Kociemba; wik) reduces an arbitrary state to a structured subgroup and then completes the solve inside that subgroup. Let $H$ (often also written as $G_1$) be the Phase-1 target subgroup,

$$H = \big\{ g \in G_0 \,\big|\, \text{all edges oriented, all corners oriented,}$$
$$\text{and the four UD-slice edges lie in the UD slice} \big\}.$$

Equivalently, $H$ is exactly the set of states reachable using the restricted Phase-2 move set

$$\langle U, U', U^2,\ D, D', D^2,\ R^2,\ L^2,\ F^2,\ B^2 \rangle,$$

since these moves preserve the above invariants. Finally,

$$\{e\} = G_2 \subset H \subset G_0,$$

where $e$ is the identity (the solved cube). The overall decomposition is

$$\text{Phase 1: } G_0 \to H \quad \text{(reduce to } H\text{)} \qquad \text{Phase 2: } H \to \{e\} \quad \text{(solve within } H\text{)}.$$

In practice, both phases use depth-iterative search guided by large pruning (distance) tables; Phase 1 explores cosets of $H$ in $G_0$, and Phase 2 searches within $H$ down to $e$.

**Algorithmic sketch.** Phase 1 finds a short maneuver (typically $\leq 12$ HTM moves) that sends the current state into $H$. Phase 2 continues from $H$ using only $U^{\pm 1}, U^2, D^{\pm 1}, D^2, R^2, L^2, F^2, B^2$ to reach the identity. Due to strong heuristics and symmetry reductions, solutions are obtained quickly and are usually $\leq 20$ moves, though not formally guaranteed to be optimal (Kociemba).

**Typical properties.** The method is fast, yields short, clean plans, and is well-suited as a callable planning tool for agents.

### K.3 Optimal Solver (IDA* with Complete Pruning)

**Algorithmic sketch.** The Optimal Solver performs Iterative Deepening A* (IDA*) search (Korf, 1985; 1997), guided by admissible heuristics derived from large pattern/pruning databases. IDA* combines the space-efficiency of depth-first search with the optimality guarantees of A*. The search depth limit is increased incrementally; once a solution is found at depth $d$, minimality is certified since all shorter paths have been exhausted.

---

**Algorithm 1** IDA* with Pattern Database Heuristics

---

1: **function** IDA*($s_0$)
2:     $bound \leftarrow h(s_0)$
3:     **while** true **do**
4:        $t \leftarrow$ SEARCH($s_0$, 0, $bound$)
5:        **if** $t = \infty$ **then return** failure
6:        **else if** $t =$ solution **then return** solution path
7:        **else**
8:           $bound \leftarrow t$
9: **function** SEARCH($s$, $g$, $bound$)
10:     $f \leftarrow g + h(s)$
11:     **if** $f > bound$ **then return** $f$
12:     **if** $s$ is goal **then return** solution
13:     $min \leftarrow \infty$
14:     **for** each successor $s'$ of $s$ **do**
15:        $t \leftarrow$ SEARCH($s'$, $g + 1$, $bound$)
16:        **if** $t =$ solution **then return** solution
17:        **if** $t < min$ **then**
18:           $min \leftarrow t$
    **return** $min$

---

**Heuristic.** The pruning tables (pattern databases) precompute exact solution lengths for subproblems such as subsets of edges or corners. During search, these values serve as admissible heuristics, drastically reducing the number of expanded states.

**Typical properties.** Under HTM, the $3 \times 3 \times 3$ cube can always be solved within 20 moves, with some positions requiring the exact 20 moves. (Rokicki et al., 2014; 2010). While run time can be seconds on easy instances, hard positions near the 20-move depth may require minutes to hours. However, the returned solution is provably optimal, making the solver suitable for dataset construction and difficulty certification.

Table 7: Two-Phase vs. Optimal Solvers.

| Solver | Optimality | Role |
|--------|-----------|------|
| Two-Phase | Near-optimal | Agent tool |
| Optimal | Provably optimal | Testcase construction |

## L    CASE STUDIES

- Sec. L.1: An agent solves the cube by applying a heuristic beam search.

- Sec. L.2: An agent tries first with IDDFS and then with the Beginner's Method.

- Sec. L.3: A less capable agent attempts a random walk (brute-force) and fails to make progress.

- Sec. L.4: An agent implements a meet-in-the-middle search, but its pruning strategies are insufficient to overcome the exponential state space.

- Sec. L.5: An agent bypasses visual reasoning, relying exclusively on a heuristic reward to guide its symbolic search.

- Sec. L.6: A capable agent is misled by a simplistic reward and only succeeds after abandoning it in favor of its own subgoal-based reasoning.

- Sec. L.7: A simple external reward successfully guides an agent's search, leading to incremental progress.

- Sec. L.8: An agent learns to use an external planner by systematically experimenting with input formats.

- Sec. L.9: An agent succeeds on a structured visual task by using a symbolic shortcut but fails on a distorted view that requires genuine spatial reasoning.

### L.1    BASIC AGENT – BEAM SEARCH

| Model | Agent Type | Input | Reward | Depth |
|-------|-----------|-------|--------|-------|
| GPT-5 | Basic | Full Symbolic | No reward | 8 |

In this case, GPT-5 applies a heuristic beam search strategy to solve the Rubik's Cube. Initially, the agent deduces the exact permutations of the six basic moves (F, B, U, D, R, L) using randomized scrambles and signature matching. With these move permutations, the agent can simulate cube state transitions internally, allowing it to conduct search without repeated environment queries.

The search itself is implemented as a beam-style search with heuristic pruning:

**Heuristic function:** The agent uses a simple facelet mismatch count to evaluate each state relative to the solved configuration.

**Beam width:** At each depth, candidate states are sorted by heuristic value, and only the top 1,500 candidates are retained (`frontier = candidates[:beam_width]`), limiting the branching factor.

**Pruning rules:** Immediate inverse moves are skipped (`if last and inverse[last] == mv: continue`), and already visited states are not revisited (`if new_state in visited: continue`).

During search, the frontier grows progressively with depth (e.g., $12 \rightarrow 114 \rightarrow 1{,}068$ candidates in the first three layers), demonstrating layered expansion with controlled breadth. Eventually, the agent identifies a four-move solution, applies it, and verifies that the cube reaches the solved state.

*Step 3*

**Thought.**

```
Thought: The previous attempt failed because the assumed face rotation orientations for U, D, R, L did
not match the environment. I will now robustly deduce the exact permutation for each basic move (F, B,
U, D, R, L) by signature matching: I'll run multiple randomized scrambles, snapshot the state before
and after applying the move, and then reconstruct the index permutation by matching the K-length color
signature for each index. This is independent of face orientations. With all six move permutations
learned, I'll perform a heuristic beam search offline to find a solution, then apply the move sequence
with make_move and verify. If the cube becomes solved, I'll call final_answer("").
```

**Code.**

```python
import random
import time

# Utilities
def inverse_move(mv):
    return mv[:-1] if mv.endswith("'") else mv + "'"

def apply_seq(seq):
    for m in seq:
        make_move(m)

def invert_seq(seq):
    return [inverse_move(m) for m in reversed(seq)]

def deduce_move_by_signature(move, trials=7, scramble_len=12):
    # Ensure we start from a known base and return to it after each trial and at the end.
    base0 = get_observation()
    pre_list = []
    post_list = []

    # Allowed moves for scrambling
    moves = ['F','B','U','D','R','L']
    all_moves = moves + [m+"'" for m in moves]

    for t in range(trials):
        # Make a scramble that avoids immediate inverses
        scramble = []
        last = None
        for k in range(scramble_len):
            while True:
                m = random.choice(all_moves)
                if last and inverse_move(last) == m:
                    continue
                scramble.append(m)
                last = m
                break
        # Apply scramble
        apply_seq(scramble)
        s_pre = get_observation()
        # Apply the move to learn
        make_move(move)
        s_post = get_observation()
        # Undo the move
        make_move(inverse_move(move))
        # Undo scramble
        apply_seq(invert_seq(scramble))
        # Verify we returned to base0
        if get_observation() != base0:
            print("Warning: state mismatch after undoing scramble during deduction of", move)
        pre_list.append(s_pre)
        post_list.append(s_post)
```

```python
    # Build signature vectors for each index
    pre_sigs = ["".join(pre[i] for pre in pre_list) for i in range(54)]
    post_sigs = ["".join(post[i] for post in post_list) for i in range(54)]

    # Map post index j to pre index i such that signatures match
    perm = [None]*54
    # Build dict from signature to indices for pre
    sig_to_indices = {}
    for i, sig in enumerate(pre_sigs):
        sig_to_indices.setdefault(sig, []).append(i)

    ambiguous = []
    missing = []
    for j, sig in enumerate(post_sigs):
        candidates = sig_to_indices.get(sig, [])
        if len(candidates) == 1:
            perm[j] = candidates[0]
        elif len(candidates) == 0:
            missing.append(j)
        else:
            ambiguous.append((j, candidates))

    if missing or ambiguous:
        # Try additional trials to disambiguate
        extra = 0
        max_extra = 6
        while (missing or ambiguous) and extra < max_extra:
            extra += 1
            # Do one extra trial
            scramble = []
            last = None
            for k in range(scramble_len):
                while True:
                    m = random.choice(all_moves)
                    if last and inverse_move(last) == m:
                        continue
                    scramble.append(m)
                    last = m
                    break
            apply_seq(scramble)
            s_pre = get_observation()
            make_move(move)
            s_post = get_observation()
            make_move(inverse_move(move))
            apply_seq(invert_seq(scramble))
            pre_list.append(s_pre)
            post_list.append(s_post)
            # Recompute signatures
            pre_sigs = ["".join(pre[i] for pre in pre_list) for i in range(54)]
            post_sigs = ["".join(post[i] for post in post_list) for i in range(54)]
            sig_to_indices = {}
            for i, sig in enumerate(pre_sigs):
                sig_to_indices.setdefault(sig, []).append(i)
            perm = [None]*54
            missing = []
            ambiguous = []
            for j, sig in enumerate(post_sigs):
                candidates = sig_to_indices.get(sig, [])
                if len(candidates) == 1:
                    perm[j] = candidates[0]
                elif len(candidates) == 0:
                    missing.append(j)
                else:
                    ambiguous.append((j, candidates))

        if missing or ambiguous:
            print(f"Could not uniquely deduce mapping for {move}. Missing={len(missing)}, "
            ambiguous={len(ambiguous)}")
            return None

    # Validate by applying to a fresh snapshot
    base = get_observation()
    make_move(move)
    s1 = get_observation()
    make_move(inverse_move(move))
    # Build simulated application
    sim = "".join(base[perm[i]] for i in range(54))
    if sim != s1:
        print(f"Validation failed for move {move}.")
        return None
    return perm

# Learn all base move permutations
```

```python
moves = ['F','B','U','D','R','L']
perm_map = {}
for mv in moves:
    print("Learning permutation for", mv)
    p = deduce_move_by_signature(mv, trials=7, scramble_len=10)
    if p is None:
        print("Failed to learn move:", mv)
        # Try a second attempt with more trials if failed
        p = deduce_move_by_signature(mv, trials=10, scramble_len=12)
    if p is None:
        print("Giving up on move:", mv)
        break
    perm_map[mv] = p

if len(perm_map) != 6:
    print("Failed to learn all permutations. Learned:", list(perm_map.keys()))
else:
    # Build prime move perms
    def invert_perm(perm):
        inv = [0]*54
        for i, src in enumerate(perm):
            inv[src] = i
        return inv
    for mv in moves:
        perm_map[mv+"'"] = invert_perm(perm_map[mv])

    def sim_apply_state(state, move):
        perm = perm_map[move]
        return "".join(state[perm[i]] for i in range(54))

    # Solved target
    solved = "R"*9 + "G"*9 + "B"*9 + "Y"*9 + "O"*9 + "W"*9

    def heuristic(state):
        # Simple facelet mismatch
        return sum(1 for i,ch in enumerate(state) if ch != solved[i])

    # Beam search
    all_moves = moves + [m+"'" for m in moves]
    inverse = {m: m+"'" for m in moves}
    inverse.update({m+"'": m for m in moves})

    start = get_observation()
    if start == solved:
        final_answer("")
    else:
        beam_width = 1500
        max_depth = 20

        frontier = [(start, [])]
        visited = set([start])
        best_h = heuristic(start)
        print("Initial heuristic:", best_h)

        solved_path = None

        for depth in range(1, max_depth+1):
            candidates = []
            for state, path in frontier:
                last = path[-1] if path else None
                for mv in all_moves:
                    if last and inverse[last] == mv:
                        continue
                    new_state = sim_apply_state(state, mv)
                    if new_state in visited:
                        continue
                    visited.add(new_state)
                    new_path = path + [mv]
                    if new_state == solved:
                        solved_path = new_path
                        break
                    h = heuristic(new_state)
                    candidates.append((h, new_state, new_path))
                if solved_path is not None:
                    break
            if solved_path is not None:
                print("Found solution of length", len(solved_path))
                break
            candidates.sort(key=lambda x: (x[0], len(x[2])))
            frontier = [(st, p) for (h, st, p) in candidates[:beam_width]]
            print(f"Depth {depth}: frontier={len(frontier)}, best_h_so_far={min([h for (h,_,_) in
            candidates], default=best_h)}")
            if not frontier:
```

```
            break

    if solved_path is None:
        print("Beam search did not find a solution.")
    else:
        print("Applying solution:", " ".join(solved_path))
        apply_seq(solved_path)
        final_state = get_observation()
        print("Final state:", final_state)
        if final_state == solved:
            final_answer("")
        else:
            print("Solution application did not reach solved state. Mismatches:",
            heuristic(final_state))
```

**Observation.**

```
Learning permutation for F
Learning permutation for B
Learning permutation for U
Learning permutation for D
Learning permutation for R
Learning permutation for L
Initial heuristic: 36
Depth 1: frontier=12, best_h_so_far=31
Depth 2: frontier=114, best_h_so_far=22
Depth 3: frontier=1068, best_h_so_far=12
Found solution of length 4
Applying solution: B R' F U
Final state: RRRRRRRRRRGGGGGGGGGGGGBBBBBBBBBBBBYYYYYYYYYYOOOOOOOOOOWWWWWWWWW
```

*Solved.*

## L.2 BASIC AGENT – IDDFS AND BEGINNER'S METHOD

| Model | Agent Type | Input | Reward | Depth |
|---|---|---|---|---|
| gpt-oss-120b | Basic | Full Symbolic | No reward | 8 |

In this case, the agent first performs a depth-limited iterative deepening search `dfs(depth, max_depth, last_face)` systematically explores move sequences, applying each candidate move with `apply(m)` and backtracking via `undo(m)`. Pruning strategies are directly reflected in the code: `if last_face is not None and face_of(m) == last_face: continue` prevents consecutive moves on the same face, and `is_inverse(a, b)` is used to avoid immediate inverse moves. Early goal checks (`if is_solved_state(get_observation())`) terminate branches once a solution is found. Iterative deepening is implemented by looping over depth limits (`for d in range(limit_depth+1)`).

Unfortunately, several searches failed. After a few failures, it attempted a different strategy.

In its thoughts (Steps 8-10), the agent correctly outlines the sub-goals of the beginner's method, from building the white cross to solving the final layer with OLL (Orientation of the Last Layer) and PLL (Permutation of the Last Layer) algorithms.

However, the agent's execution is undermined by a subtle but critical flaw in its internal world model—an incorrect mapping of the cube's indices to its symbolic representation. The agent itself explicitly identifies this error in a moment of self-correction in Step 13, noting that "earlier mappings had mistakes." Despite this high-level awareness, the cognitive load of implementing the entire long-horizon plan within a single, complex code block introduces new, unperceived errors. The final outcome is a failure, where the cube's state is no closer to being solved.

*Step 4*

**Thought.**

> Thought: The previous searches were too deep or inefficient. I will attempt a carefully pruned iterative deepening search up to depth 5 using a reduced move set that includes quarter and half turns, with early goal checks at every node and strict pruning (no repeating the same face, and no immediate inverses). This should keep the number of operations manageable. If a solution exists within 5 moves, it will be found and applied directly on the real cube; then I will call final_answer("").

**Code.**

```python
# Depth-limited IDDFS with pruning and early goal checks.
# Uses moves including half turns (implemented as two quarter turns).

from time import time

def is_solved_state(s):
    return (all(c == 'R' for c in s[0:9]) and
            all(c == 'G' for c in s[9:18]) and
            all(c == 'B' for c in s[18:27]) and
            all(c == 'Y' for c in s[27:36]) and
            all(c == 'O' for c in s[36:45]) and
            all(c == 'W' for c in s[45:54]))

start = get_observation()
if is_solved_state(start):
    final_answer("")
else:
    # Define move variants with half-turns
    basic_faces = ['F','B','L','R','U','D']
    moves = []
    for f in basic_faces:
        moves.append(f)
        moves.append(f+"'")
        moves.append(f+"2")
    # Helper to apply and undo moves in the environment
    def apply(m):
        if m.endswith("2"):
            face = m[0]
            make_move(face)
            make_move(face)
        else:
            make_move(m)
    def undo(m):
        if m.endswith("2"):
            face = m[0]
            make_move(face)
            make_move(face)
        else:
            # inverse of X is X', inverse of X' is X
            if m.endswith("'"):
                make_move(m[0])
            else:
                make_move(m+"'")
    def face_of(m):
        return m[0]
    def is_inverse(a, b):
        # Treat 2-turn as self-inverse
        if a[0] != b[0]:
            return False
        if a.endswith("2") and b.endswith("2"):
            return True
        if a.endswith("'") and (b == a[0]):
            return True
        if (a == a[0]) and b.endswith("'") and (b[0] == a[0]):
            return True
        return False

    max_nodes = 300000  # safeguard against excessive search
    node_count = 0
    found = False
    solution_path = []

    def dfs(depth, max_depth, last_face):
        nonlocal node_count, found, solution_path
        if found:
            return True
        node_count += 1
        if node_count > max_nodes:
            return False
```

```
        # Early goal check at every node
        if is_solved_state(get_observation()):
            found = True
            return True
        if depth == max_depth:
            return False
        for m in moves:
            # pruning: avoid repeating the same face consecutively
            if last_face is not None and face_of(m) == last_face:
                continue
            # pruning: avoid immediate inverse with previous move in path if exists
            prev = solution_path[-1] if solution_path else None   # note: we append on success; here
            prev is not applicable
            # Apply move
            apply(m)
            if dfs(depth+1, max_depth, face_of(m)):
                solution_path.append(m)
                return True
            # Backtrack
            undo(m)
        return False

    # Iterative deepening up to depth 5
    start_time = time()
    limit_depth = 5
    for d in range(limit_depth+1):
        if dfs(0, d, None):
            break
    # If found, the moves leading to solution were applied in-place during recursion.
    if found and is_solved_state(get_observation()):
        final_answer("")
    else:
        print(f"Search exhausted (nodes={node_count}). Could not solve within depth {limit_depth}.")
```

**Observation.**

```
<Killed by the Python interpreter>
Maybe there is an infinite loop somewhere in the code, or you're just asking too many calculations.
```

Step 5, 6 still failed. Changed strategy at Step 7.

**Thought.**

```
Thought: Our previous brute-force search approaches hit interpreter limits. I will now implement a
deterministic, step-by-step white-cross builder (beginner's method) that:
- Analyzes the current state string (observation-type is state_string).
- Locates each white edge (W with its partner color).
- Applies targeted moves to:
  - Lift the white edge to the U layer with white facing up using single face turns.
  - Rotate U (and D when needed) to align the edge above its correct face.
  - Insert with a double face turn (X2) to place it correctly in the D (white) cross while keeping
  previously placed edges intact.

I will implement robust mapping for edges and use only simple, safe moves (F/F'/F2, R/R'/R2, B/B'/B2,
L/L'/L2, U/U', D/D'). I will repeat for the four target white edges: WR (to FD), WY (to RD), WG (to
BD), WB (to LD). After building the white cross, I will check if the cube is already solved; if it is,
I will call final_answer(""). If not, we can proceed further in subsequent steps (but for now, try to
get the cross in place). Let's execute this first stage.
```

**Code.**

```
# Build the white cross step-by-step using targeted moves.
# Faces and centers per the problem:
#   F (front) = Red (R)
#   B (back)  = Green (G)
#   L (left)  = Blue (B)
#   R (right) = Yellow (Y)
```

```python
#   U (up)    = Orange (O)
#   D (down)  = White (W)

# Helper: Apply a move, double move, inverse move
def do(m):
    make_move(m)

def do2(face):
    make_move(face); make_move(face)

def inv(m):
    return m[:-1] if m.endswith("'") else (m + "'")

def is_solved_state(s):
    return (all(c == 'R' for c in s[0:9]) and
            all(c == 'G' for c in s[9:18]) and
            all(c == 'B' for c in s[18:27]) and
            all(c == 'Y' for c in s[27:36]) and
            all(c == 'O' for c in s[36:45]) and
            all(c == 'W' for c in s[45:54]))

# Centers indices
def centers_of(state):
    return {
        'F': state[4],    # should be 'R'
        'B': state[13],   # should be 'G'
        'L': state[22],   # should be 'B'
        'R': state[31],   # should be 'Y'
        'U': state[40],   # should be 'O'
        'D': state[49],   # should be 'W'
    }

# Edge map: name -> ((face1, idx1), (face2, idx2))
edge_defs = {
    'UF': (('U', 42), ('F', 1)),
    'UR': (('U', 41), ('R', 28)),
    'UB': (('U', 37), ('B', 10)),
    'UL': (('U', 39), ('L', 19)),
    'FR': (('F', 5),  ('R', 30)),
    'FL': (('F', 3),  ('L', 23)),
    'FD': (('F', 7),  ('D', 46)),
    'RB': (('R', 32), ('B', 12)),
    'RD': (('R', 34), ('D', 50)),
    'LB': (('L', 21), ('B', 14)),
    'LD': (('L', 25), ('D', 48)),
    'BD': (('B', 16), ('D', 52)),
}

# For each face's "UX" and "XD" reference
UX_of = {'F': 'UF', 'R': 'UR', 'B': 'UB', 'L': 'UL'}
XD_of = {'F': 'FD', 'R': 'RD', 'B': 'BD', 'L': 'LD'}

# Rotations of U that cycle UF->UR->UB->UL->UF and likewise the edge names
U_cycle = ['UF', 'UR', 'UB', 'UL']
# Rotations of D that cycle FD->LD->BD->RD->FD under D' (counterclockwise) or FD->RD->BD->LD->FD under
D
D_cycle_clockwise = ['FD', 'RD', 'BD', 'LD']

# Helper to read current state char at a given absolute index
def read(state, idx):
    return state[idx]

def get_edge_colors(state, edge_name):
    (f1, i1), (f2, i2) = edge_defs[edge_name]
    return (state[i1], state[i2])

def locate_white_edge_partner(state, partner_color):
    # Returns (edge_name, where_white) where where_white in {0,1} indexes which of the two stickers is
    white
    for name, ((f1, i1), (f2, i2)) in edge_defs.items():
        c1, c2 = state[i1], state[i2]
        if (c1 == 'W' and c2 == partner_color):
            return name, 0
        if (c2 == 'W' and c1 == partner_color):
            return name, 1
    return None, None

# Rotate U k times (k in {0..3})
def U_times(k):
    for _ in range(k % 4):
        do('U')

# Rotate D k times clockwise
```

```python
def D_times(k):
    for _ in range(k % 4):
        do('D')

# Align a U-layer edge to be at a specific UX position by rotating U
def align_U_edge_to(state, edge_name_current, target_UX):
    # edge_name_current must be one of U-layer edges: 'UF','UR','UB','UL'
    # compute offset
    cur_idx = U_cycle.index(edge_name_current)
    tgt_idx = U_cycle.index(target_UX)
    k = (tgt_idx - cur_idx) % 4
    U_times(k)

# Align a D-layer edge to be at a specific XD position by rotating D
def align_D_edge_to(state, edge_name_current, target_XD):
    cur_idx = D_cycle_clockwise.index(edge_name_current)
    tgt_idx = D_cycle_clockwise.index(target_XD)
    k = (tgt_idx - cur_idx) % 4
    D_times(k)

# Given an edge in the middle layer, lift it to U with white sticker on U using a single move.
def lift_middle_white_to_U(state, edge_name, white_on_face):
    # white_on_face is 'F','R','B','L' representing which face holds the white sticker for this edge.
    # For each middle edge we choose the face turn that moves it to the U layer with white on U.
    if edge_name == 'FR':
        if white_on_face == 'F':
            do("F'")
        elif white_on_face == 'R':
            do("R")
    elif edge_name == 'FL':
        if white_on_face == 'F':
            do("F")
        elif white_on_face == 'L':
            do("L'")
    elif edge_name == 'RB':
        if white_on_face == 'R':
            do("R'")
        elif white_on_face == 'B':
            do("B")
    elif edge_name == 'LB':
        if white_on_face == 'L':
            do("L")
        elif white_on_face == 'B':
            do("B'")
    else:
        # Not a middle edge
        pass

# If in U-layer but white on side, drop it to D with white on D using a single face turn.
def drop_U_side_white_to_D(state, edge_name, white_on_face):
    # edge_name one of UF,UR,UB,UL; white_on_face in {'F','R','B','L'}
    if edge_name == 'UF' and white_on_face == 'F':
        do("F")
    elif edge_name == 'UR' and white_on_face == 'R':
        do("R")
    elif edge_name == 'UB' and white_on_face == 'B':
        do("B")
    elif edge_name == 'UL' and white_on_face == 'L':
        do("L")

# Eject from D when white on side: bring under that face then do that face' to bring to middle/U
def eject_D_side_white(state, edge_name, white_on_face):
    # edge_name in {'FD','RD','BD','LD'}, white_on_face in {'F','R','B','L'}
    # First rotate D to bring it under the face that has the white sticker
    target = white_on_face + 'D'
    align_D_edge_to(state, edge_name, target)
    # Now do the inverse face to move the edge up into the middle layer where we can lift it in next
    steps
    do(white_on_face + "'")

# After white edge is on U face with white up at some U*, align above its target face and insert with
X2
def insert_from_U_white_up(state, target_face):
    # target_face in {'F','R','B','L'}
    # Bring the white edge to UX position by U-turns, then do target_face2
    # To know current UX position, find WR edge in U layer:
    # But we call this only after aligning with previous helpers; safe to rotate U until the partner
    color aligns with target center
    # We'll explicitly rotate U until the edge is at UX.
    # Determine which UX currently holds the (W, partner) edge
    desired_edge_name = UX_of[target_face]
    # Find the current U-edge location of the piece with colors {W, centers[target_face]}
    state2 = get_observation()
```

```python
    centers = centers_of(state2)
    partner = centers[target_face]
    # Find among U-edges where this WR is
    u_positions = ['UF','UR','UB','UL']
    current_u = None
    where_white = None
    for en in u_positions:
        (f1, i1), (f2, i2) = edge_defs[en]
        c1, c2 = state2[i1], state2[i2]
        if (c1 == 'W' and c2 == partner):
            current_u, where_white = en, f1  # white on f1
            break
        if (c2 == 'W' and c1 == partner):
            current_u, where_white = en, f2
            break
    if current_u is None:
        return
    # Rotate U to bring it to UX_of[target_face]
    align_U_edge_to(state2, current_u, desired_edge_name)
    # Perform the double turn to insert
    do2(target_face)

# Main white cross builder
def build_white_cross():
    # Process targets in order: F (R), R (Y), B (G), L (B)
    order = ['F','R','B','L']
    max_outer_iters = 80
    for face in order:
        iters = 0
        while iters < 40:
            iters += 1
            st = get_observation()
            ctr = centers_of(st)
            partner = ctr[face]  # desired partner color for this edge
            # Check if it's already solved at XD with correct orientation
            xd = XD_of[face]
            (f1, i1), (f2, i2) = edge_defs[xd]
            c1, c2 = st[i1], st[i2]
            # Determine orientation at XD: white on D position?
            if (c1 == partner and c2 == 'W') or (c2 == partner and c1 == 'W'):
                # Additionally ensure white is on D (i.e., the index corresponding to D in this XD)
                # For XD edge, the pair is (face, idx_face), (D, idx_down). The D index may be either
                # i1 or i2.
                # Identify which is D:
                down_idx = i1 if f1 == 'D' else (i2 if f2 == 'D' else None)
                if down_idx is not None and st[down_idx] == 'W':
                    # Solved for this edge
                    break

            # Otherwise, locate this white-partner edge
            edge_name, pos = locate_white_edge_partner(st, partner)
            if edge_name is None:
                # Should not happen
                break

            # Determine which faces the two stickers are on and their colors
            (ef1, ei1), (ef2, ei2) = edge_defs[edge_name]
            col1, col2 = st[ei1], st[ei2]

            # Case handling
            if edge_name in ['UF','UR','UB','UL']:
                # U layer
                # Identify where white is: on U or on side
                white_on = None
                if st[ei1] == 'W':
                    white_on = ef1
                elif st[ei2] == 'W':
                    white_on = ef2
                if white_on == 'U':
                    # Rotate U to UX and insert with face2
                    insert_from_U_white_up(st, face)
                else:
                    # White on side face in U layer: drop to D with single move on that face
                    drop_U_side_white_to_D(st, edge_name, white_on)
            elif edge_name in ['FR','FL','RB','LB']:
                # Middle layer: lift to U with white on U
                white_on = None
                if st[ei1] == 'W':
                    white_on = ef1
                elif st[ei2] == 'W':
                    white_on = ef2
                lift_middle_white_to_U(st, edge_name, white_on)
            elif edge_name in ['FD','RD','BD','LD']:
```

```python
                # D layer
                # Determine if white is on D or on side
                white_on = None
                if st[ei1] == 'W':
                    white_on = ef1
                elif st[ei2] == 'W':
                    white_on = ef2
                if white_on == 'D':
                    # Rotate D to bring under target face, insert with face2
                    current_xd = edge_name
                    align_D_edge_to(st, current_xd, XD_of[face])
                    do2(face)
                else:
                    # White on side at D layer: eject up to middle/U
                    eject_D_side_white(st, edge_name, white_on)
            else:
                # Should not happen
                pass
        # Next target
    # End for faces

# First, if solved already, finish
st0 = get_observation()
if is_solved_state(st0):
    final_answer("")
else:
    # Build white cross
    build_white_cross()
    # Check if solved by chance, else continue later
    st1 = get_observation()
    if is_solved_state(st1):
        final_answer("")
    else:
        # Print progress: show current state to proceed next steps in subsequent iteration
        print(get_observation())
```

**Observation.**

```
OGGYRRRBBORWRGBWGGYWBYBGYOBWOYOYGOYGROOBOYGWBWRRBWWRWY
```

*Step 8*

**Thought.**

```
I will continue solving using the beginner's method. I already built some progress toward the white
cross. Next, I will insert the four white corners into the D (white) layer without disturbing the
cross, using standard triggers:
- For DFR (from UFR): R U R' U'
- For DRB (from URB): B U B' U'
- For DBL (from UBL): L' U' L U
- For DLF (from ULF): F' U' F U

I will:
- Define corner indices.
- Detect each target white corner by its color set {W, center(face1), center(face2)}.
- If found in D but unsolved: rotate D to bring it to the correct slot and eject with the corresponding
trigger until it pops to U.
- If found in U: rotate U to place above its target slot, then apply the corresponding trigger
repeatedly until solved.
- Repeat for all four corners, then print the current state. If solved, I will call final_answer("").
```

*Step 9*

**Thought.**

```
  I will proceed with the next stage of the beginner's method: solve the middle layer edges (second
  layer) without disturbing the completed white layer. I will:
- Identify middle layer target edges (FR, FL, RB, LB).
```

```
- Insert correct U-layer edges that do not contain U or D colors (i.e., not containing O or W).
- If no such U edges exist, eject an incorrectly placed middle edge to the U layer, then insert it
correctly.
I will then print the updated cube state (or call final_answer("") if by chance the cube becomes
solved).
```

*Step 10*

**Thought.**

```
The cube now has the first two layers solved. I will complete the last layer using beginner's method:
- OLL: orient U (Orange) edges to a cross using the FRUR'U'F' sequence with correct U rotations; then
orient corners using Sune/anti-Sune until all U stickers are Orange.
- PLL: first permute last-layer corners using U R U' L' U R' U' L; then permute last-layer edges using
Ua/Ub permutations. I will implement helper checks and safe application with inverses so I can revert
if a guess doesn't solve. I'll check for solved after each stage and finish with final_answer("") if
solved.
```

*Step 13*

**Thought.**

```
I will correct the cube indexing for edges and corners (earlier mappings had mistakes), then run a full
beginner-method pipeline: build the white cross, insert white corners, solve middle-layer edges, and
finish with OLL and PLL. I will expand any \2" turns into two quarter turns so the environment accepts
them. I will check for solved after each phase, and if solved at any point, I will immediately call
final_answer("").
```

**Code.**

```python
# Corrected, end-to-end beginner method solver with accurate index mappings and safe "2" expansion.

# ========== Helpers and Mappings ==========

def do(m):
    # Expand "2" moves into two quarter turns
    if m.endswith("2"):
        f = m[0]
        make_move(f)
        make_move(f)
    else:
        make_move(m)

def do2(face):
    make_move(face)
    make_move(face)

def is_solved_state(s):
    return (all(c == 'R' for c in s[0:9]) and
            all(c == 'G' for c in s[9:18]) and
            all(c == 'B' for c in s[18:27]) and
            all(c == 'Y' for c in s[27:36]) and
            all(c == 'O' for c in s[36:45]) and
            all(c == 'W' for c in s[45:54]))

def centers_of(state):
    return {
        'F': state[4],   # R
        'B': state[13],  # G
        'L': state[22],  # B
        'R': state[31],  # Y
        'U': state[40],  # O
        'D': state[49],  # W
    }

# Correct edge indices based on the provided net
edge_defs = {
```

```python
    'UF': (('U', 43), ('F', 1)),
    'UR': (('U', 41), ('R', 28)),
    'UB': (('U', 37), ('B', 10)),
    'UL': (('U', 39), ('L', 19)),

    'FR': (('F', 5),  ('R', 30)),
    'FL': (('F', 3),  ('L', 23)),
    'RB': (('R', 32), ('B', 12)),
    'LB': (('L', 21), ('B', 14)),

    'FD': (('F', 7),  ('D', 46)),
    'RD': (('R', 34), ('D', 50)),
    'BD': (('B', 16), ('D', 52)),
    'LD': (('L', 25), ('D', 48)),
}

# Correct corner indices
corners = {
    'UFR': [('U',44),('F',2),('R',27)],
    'URB': [('U',38),('R',29),('B',9)],
    'UBL': [('U',36),('B',11),('L',18)],
    'ULF': [('U',42),('L',20),('F',0)],
    'DFR': [('D',47),('F',8),('R',33)],
    'DRB': [('D',45),('R',35),('B',15)],
    'DBL': [('D',51),('B',17),('L',24)],
    'DLF': [('D',53),('L',26),('F',6)],
}

# Cycles for U and D layer edges
U_cycle = ['UF', 'UR', 'UB', 'UL']
D_cycle_cw = ['FD', 'RD', 'BD', 'LD']  # Under D (clockwise)

UX_of = {'F': 'UF', 'R': 'UR', 'B': 'UB', 'L': 'UL'}
XD_of = {'F': 'FD', 'R': 'RD', 'B': 'BD', 'L': 'LD'}

def U_times(k):
    for _ in range(k % 4): do('U')

def D_times(k):
    for _ in range(k % 4): do('D')

def align_U_edge_to(state, cur_edge, target_edge):
    ci = U_cycle.index(cur_edge)
    ti = U_cycle.index(target_edge)
    k = (ti - ci) % 4
    U_times(k)

def align_D_edge_to(state, cur_edge, target_edge):
    ci = D_cycle_cw.index(cur_edge)
    ti = D_cycle_cw.index(target_edge)
    k = (ti - ci) % 4
    D_times(k)

def get_edge_colors(state, edge_name):
    (f1, i1), (f2, i2) = edge_defs[edge_name]
    return (state[i1], state[i2])

def locate_white_edge_partner(state, partner_color):
    for name, ((f1,i1),(f2,i2)) in edge_defs.items():
        c1, c2 = state[i1], state[i2]
        if c1 == 'W' and c2 == partner_color:
            return name, 0
        if c2 == 'W' and c1 == partner_color:
            return name, 1
    return None, None

# ========== White Cross ==========

def lift_middle_white_to_U(state, edge_name, white_on_face):
    # Move middle edge to U with white on U
    if edge_name == 'FR':
        if white_on_face == 'F': do("F'")
        elif white_on_face == 'R': do("R")
    elif edge_name == 'FL':
        if white_on_face == 'F': do("F")
        elif white_on_face == 'L': do("L'")
    elif edge_name == 'RB':
        if white_on_face == 'R': do("R'")
        elif white_on_face == 'B': do("B")
    elif edge_name == 'LB':
        if white_on_face == 'L': do("L")
        elif white_on_face == 'B': do("B'")
```

```python
def drop_U_side_white_to_D(state, edge_name, white_on_face):
    if edge_name == 'UF' and white_on_face == 'F': do("F")
    elif edge_name == 'UR' and white_on_face == 'R': do("R")
    elif edge_name == 'UB' and white_on_face == 'B': do("B")
    elif edge_name == 'UL' and white_on_face == 'L': do("L")

def eject_D_side_white(state, edge_name, white_on_face):
    target = white_on_face + 'D'
    align_D_edge_to(state, edge_name, target)
    do(white_on_face + "'")

def insert_from_U_white_up(state, target_face):
    desired_u = UX_of[target_face]
    st2 = get_observation()
    centers = centers_of(st2)
    partner = centers[target_face]
    current_u = None
    for en in U_cycle:
        (f1,i1),(f2,i2) = edge_defs[en]
        c1, c2 = st2[i1], st2[i2]
        if (c1 == 'W' and c2 == partner) or (c2 == 'W' and c1 == partner):
            current_u = en
            break
    if current_u is None:
        return
    align_U_edge_to(st2, current_u, desired_u)
    do2(target_face)

def build_white_cross():
    order = ['F','R','B','L']
    for face in order:
        for _ in range(60):
            st = get_observation()
            ctr = centers_of(st)
            partner = ctr[face]
            # Check if solved at XD (white on D index for that edge)
            xd = XD_of[face]
            (f1,i1),(f2,i2) = edge_defs[xd]
            d_idx = i1 if f1 == 'D' else (i2 if f2 == 'D' else None)
            if d_idx is not None:
                if ((st[i1] == partner and st[i2] == 'W') or (st[i2] == partner and st[i1] == 'W')) and \
                    st[d_idx] == 'W':
                    break  # Edge in place and oriented
            # Locate target edge
            en, pos = locate_white_edge_partner(st, partner)
            if en is None:
                break
            (ef1,ei1),(ef2,ei2) = edge_defs[en]
            # Determine where white is
            white_on = ef1 if st[ei1] == 'W' else (ef2 if st[ei2] == 'W' else None)
            if en in U_cycle:
                if white_on == 'U':
                    insert_from_U_white_up(st, face)
                else:
                    drop_U_side_white_to_D(st, en, white_on)
            elif en in ['FR','FL','RB','LB']:
                lift_middle_white_to_U(st, en, white_on)
            elif en in D_cycle_cw:
                if white_on == 'D':
                    align_D_edge_to(st, en, XD_of[face])
                    do2(face)
                else:
                    eject_D_side_white(st, en, white_on)

# ========== White Corners ==========

def get_corner_colors(state, name):
    return tuple(state[idx] for _, idx in corners[name])

def corner_color_set(state, name):
    return set(get_corner_colors(state, name))

top_cycle = ['UFR','URB','UBL','ULF']
down_cycle = ['DFR','DRB','DBL','DLF']

def U_to_top_corner(current_top, target_top):
    ci = top_cycle.index(current_top)
    ti = top_cycle.index(target_top)
    U_times((ti - ci) % 4)

def D_to_down_corner(current_down, target_down):
    ci = down_cycle.index(current_down)
    ti = down_cycle.index(target_down)
```

```python
        D_times((ti - ci) % 4)

def is_corner_solved(state, name):
    # White on D face and other two colors match centers
    faces = [f for f,_ in corners[name]]
    idxs = [i for _,i in corners[name]]
    d_pos = None
    for i,f in enumerate(faces):
        if f == 'D':
            d_pos = i
            break
    if d_pos is None:
        return False
    if state[idxs[d_pos]] != 'W':
        return False
    ctr = centers_of(state)
    for i,f in enumerate(faces):
        if f == 'D': continue
        if state[idxs[i]] != ctr[f]:
            return False
    return True

triggers = {
    'DFR': {'top':'UFR', 'do': lambda: (do('R'), do('U'), do("R'"), do("U'"))},
    'DRB': {'top':'URB', 'do': lambda: (do('B'), do('U'), do("B'"), do("U'"))},
    'DBL': {'top':'UBL', 'do': lambda: (do("L'"), do("U'"), do('L'), do('U'))},
    'DLF': {'top':'ULF', 'do': lambda: (do("F'"), do("U'"), do('F'), do('U'))},
}

def find_corner_by_colors(state, target_set):
    for name in corners:
        if corner_color_set(state, name) == target_set:
            return name
    return None

def eject_down_corner(slot_name):
    trig = triggers[slot_name]['do']
    for _ in range(3):
        trig()

def insert_corner(slot_name, target_set):
    top_name = triggers[slot_name]['top']
    trig = triggers[slot_name]['do']
    for _ in range(40):
        st = get_observation()
        if is_corner_solved(st, slot_name):
            return True
        pos = find_corner_by_colors(st, target_set)
        if pos is None:
            return False
        if pos in down_cycle:
            D_to_down_corner(pos, slot_name)
            eject_down_corner(slot_name)
        elif pos in top_cycle:
            U_to_top_corner(pos, top_name)
            trig()
        else:
            trig()
    return False

def solve_white_corners():
    st = get_observation()
    ctr = centers_of(st)
    targets = [
        ('DFR', {'W', ctr['F'], ctr['R']}),
        ('DRB', {'W', ctr['R'], ctr['B']}),
        ('DBL', {'W', ctr['B'], ctr['L']}),
        ('DLF', {'W', ctr['L'], ctr['F']}),
    ]
    for slot, tset in targets:
        for _ in range(60):
            st = get_observation()
            if is_corner_solved(st, slot):
                break
            insert_corner(slot, tset)

# ========== Middle Layer ==========

def right_of(face):
    return {'F':'R','R':'B','B':'L','L':'F'}[face]

def left_of(face):
    return {'F':'L','L':'B','B':'R','R':'F'}[face]
```

```python
def is_middle_edge_solved(state, slot):
    (f1, i1), (f2, i2) = edge_defs[slot]
    c1, c2 = state[i1], state[i2]
    ctr = centers_of(state)
    return c1 == ctr[f1] and c2 == ctr[f2]

def any_middle_unsolved(state):
    return not (is_middle_edge_solved(state, 'FR') and
                is_middle_edge_solved(state, 'RB') and
                is_middle_edge_solved(state, 'LB') and
                is_middle_edge_solved(state, 'FL'))

def find_u_edge_candidate(state):
    for en in U_cycle:
        (f1,i1),(f2,i2) = edge_defs[en]
        c1, c2 = state[i1], state[i2]
        if c1 in ('O','W') or c2 in ('O','W'):
            continue
        return en
    return None

def find_u_edge_by_colors(state, colors_set):
    for en in U_cycle:
        (f1,i1),(f2,i2) = edge_defs[en]
        if set([state[i1], state[i2]]) == colors_set:
            return en
    return None

def insert_middle_from_U_above(front_face, direction):
    if direction == 'right':
        rf = right_of(front_face)
        do('U'); do(rf); do("U'"); do(rf+"'"); do("U'"); do(front_face+"'"); do('U'); do(front_face)
    else:
        lf = left_of(front_face)
        do("U'"); do(lf+"'"); do("U'"); do(lf); do('U'); do(front_face); do("U'"); do(front_face+"'")

def eject_incorrect_middle(front_face):
    insert_middle_from_U_above(front_face, 'right')

def solve_middle_layer():
    for _ in range(300):
        st = get_observation()
        if not any_middle_unsolved(st):
            break
        st = get_observation()
        ctr = centers_of(st)
        en = find_u_edge_candidate(st)
        if en is not None:
            side_map = {'UF':'F','UR':'R','UB':'B','UL':'L'}
            side = side_map[en]
            (f1,i1),(f2,i2) = edge_defs[en]
            if f1 == side:
                side_color = st[i1]
                up_color = st[i2]
            else:
                side_color = st[i2]
                up_color = st[i1]
            target_front = None
            for f in ['F','R','B','L']:
                if ctr[f] == side_color:
                    target_front = f
                    break
            if target_front is None:
                do('U'); continue
            st2 = get_observation()
            cur = find_u_edge_by_colors(st2, set([side_color, up_color]))
            if cur is None:
                do('U'); continue
            align_U_edge_to(st2, cur, UX_of[target_front])
            if ctr[right_of(target_front)] == up_color:
                insert_middle_from_U_above(target_front, 'right')
            elif ctr[left_of(target_front)] == up_color:
                insert_middle_from_U_above(target_front, 'left')
            else:
                do('U')
        else:
            st = get_observation()
            if not is_middle_edge_solved(st, 'FR'):
                eject_incorrect_middle('F')
            elif not is_middle_edge_solved(st, 'RB'):
                eject_incorrect_middle('R')
            elif not is_middle_edge_solved(st, 'LB'):
```

```
                    eject_incorrect_middle('L')
                elif not is_middle_edge_solved(st, 'FL'):
                    eject_incorrect_middle('F')

    # ========== Last Layer (OLL + PLL) ==========

    U_indices = [36,37,38,39,40,41,42,43,44]
    def u_face_all_orange(state):
        return all(state[i] == 'O' for i in U_indices)

    def u_edges(state):
        return {'top': state[37], 'left': state[39], 'right': state[41], 'bottom': state[43]}

    def u_cross_orange(state):
        e = u_edges(state)
        return e['top']=='O' and e['left']=='O' and e['right']=='O' and e['bottom']=='O'

    def solve_oll_edges():
        FRURUFp = ["F", "R", "U", "R'", "U'", "F'"]
        for _ in range(12):
            st = get_observation()
            if u_cross_orange(st):
                return
            e = u_edges(st)
            flags = {pos:(col=='O') for pos,col in e.items()}
            cnt = sum(flags.values())
            if cnt == 0:
                do_seq(FRURUFp)
            elif cnt == 2:
                if flags['left'] and flags['right']:
                    do_seq(FRURUFp)
                elif flags['top'] and flags['bottom']:
                    U_times(1); do_seq(FRURUFp)
                else:
                    for _ in range(4):
                        st2 = get_observation()
                        e2 = u_edges(st2)
                        if e2['top']=='O' and e2['left']=='O':
                            do_seq(FRURUFp); break
                        U_times(1)
            else:
                do_seq(FRURUFp)

    def solve_oll_corners():
        sune = ["R","U","R'","U","R","U2","R'"]
        antisune = ["R'","U'","R","U'","R'","U2","R"]
        for _ in range(36):
            st = get_observation()
            if u_face_all_orange(st):
                return
            for _ in range(4):
                st = get_observation()
                if st[2] == 'O':
                    do_seq(sune); break
                elif st[27] == 'O':
                    do_seq(antisune); break
                else:
                    do('U')

    edges_U_top = {
        'UF': (('U',43), ('F',1)),
        'UR': (('U',41), ('R',28)),
        'UB': (('U',37), ('B',10)),
        'UL': (('U',39), ('L',19)),
    }

    def edges_in_place_top(state):
        ctr = centers_of(state)
        for name, ((f1,i1),(f2,i2)) in edges_U_top.items():
            if state[i2] != ctr[f2]:
                return False
        return True

    def corner_side_colors_match(state, corner_name):
        faces = [f for f,_ in corners[corner_name]]
        idxs = [i for _,i in corners[corner_name]]
        ctr = centers_of(state)
        for f,i in zip(faces,idxs):
            if f == 'U': continue
            if state[i] != ctr[f]:
                return False
        return True
```

```python
def all_corners_in_place(state):
    return all(corner_side_colors_match(state, name) for name in ['UFR','URB','UBL','ULF'])

corner_cycle = ["U","R","U'","L","U","R'","U'","L"]
Ua = ["R","U'","R","U","R","U","R","U'","R","U'","R2"]
Ub = ["R2","U","R","U","R'","U'","R'","U'","R'","U","R'"]

def solve_pll_corners():
    for _ in range(12):
        st = get_observation()
        if all_corners_in_place(st):
            return
        # try to position a correct corner at UFR
        for _ in range(4):
            st = get_observation()
            if corner_side_colors_match(st, 'UFR'):
                break
            do('U')
        do_seq(corner_cycle)

def invert_move(m):
    if m.endswith("2"): return m
    if m.endswith("'"): return m[:-1]
    return m + "'"

def invert_seq(seq):
    return [invert_move(m) for m in reversed(seq)]

def try_edge_perm(alg):
    do_seq(alg)
    st = get_observation()
    if is_solved_state(st) or edges_in_place_top(st):
        return True
    do_seq(invert_seq(alg))
    return False

def solve_pll_edges():
    for _ in range(12):
        st = get_observation()
        if edges_in_place_top(st):
            return
        for _ in range(4):
            if try_edge_perm(Ua): return
            if try_edge_perm(Ub): return
            do('U')

# ========== Run the full pipeline ==========

st0 = get_observation()
if is_solved_state(st0):
    final_answer("")
else:
    # 1) White cross
    build_white_cross()
    if is_solved_state(get_observation()):
        final_answer("")
    # 2) White corners
    solve_white_corners()
    if is_solved_state(get_observation()):
        final_answer("")
    # 3) Middle layer
    solve_middle_layer()
    if is_solved_state(get_observation()):
        final_answer("")
    # 4) Last layer
    solve_oll_edges()
    solve_oll_corners()
    solve_pll_corners()
    solve_pll_edges()
    # Align U and check
    for _ in range(4):
        if is_solved_state(get_observation()):
            final_answer("")
        do('U')
    # If still not solved, print state for further iteration
    print(get_observation())

```

**Observation.**

```
RWRORRGBRGRGRGGWGORYBYBGBOBYGGOYYYYYBBYOOWOBOORWBWWWWW
```

## L.3 BASIC AGENT – BRUTE-FORCE ENUMERATION

| Model | Agent Type | Input | Reward | Depth |
|-------|-----------|-------|--------|-------|
| gpt-oss-120b | Basic | Full Symbolic | No reward | 12 |

This case demonstrates the behavior of a less capable agent in a no-reward setting. The agent executes up to 20,000 moves, selecting each move essentially at random, with only a minimal stochastic backtracking mechanism (5% chance of undoing the previous move). There is no systematic planning, pruning, or search applied, and the cube remains unsolved. Aside from the stochastic backtracking, the agent's behavior is equivalent to a random walk over the state space.

**Code.**

```python
import random

# Moves and their inverses
moves = ["F","F'","B","B'","L","L'","R","R'","U","U'","D","D'"]
inverse = {"F":"F'","F'":"F","B":"B'","B'":"B",
           "L":"L'","L'":"L","R":"R'","R'":"R",
           "U":"U'","U'":"U","D":"D'","D'":"D"}

def is_solved(state):
    # solved state order: FRONT=R, BACK=G, LEFT=B, RIGHT=Y, UP=O, DOWN=W
    solved = "R"*9 + "G"*9 + "B"*9 + "Y"*9 + "O"*9 + "W"*9
    return state == solved

# Simulated annealing / random walk with occasional acceptance of worse moves
max_steps = 20000   # allow many moves within a single tool call
state = get_observation()
if is_solved(state):
    print("Already solved")
else:
    for step in range(max_steps):
        if is_solved(state):
            print("Cube solved at step", step)
            break
        mv = random.choice(moves)
        make_move(mv)
        new_state = get_observation()
        # If the move improved (or equal) heuristic, keep it.
        # We don't have direct heuristic, so we just keep the move most of the time,
        # but occasionally revert a bad move to avoid dead-ends.
        # Simple rule: if new_state is same as previous (unlikely) revert, else keep.
        # To allow some backtracking, we revert with small probability.
        keep = True
        if random.random() < 0.05:   # 5% chance to undo
            make_move(inverse[mv])
            new_state = get_observation()
            keep = False
        state = new_state
    else:
        print("Reached max steps without solving. Final state:", state)

# Final verification
final_answer("")
```

*Not solved.*

## L.4 BASIC AGENT – FAILED PRUNING

| Model | Agent Type | Input | Reward | Depth |
|-------|-----------|-------|--------|-------|
| GPT-5 | Basic | Full Symbolic | No reward | 12 |

In this test case, the model implements a *meet-in-the-middle search* for Rubik's Cube solving. The procedure first builds a dictionary of states reachable from the solved cube (`build_goal_map`), and then performs a forward search from the scrambled cube (`forward_meet_in_middle`) to detect an intersection. This reflects the model's recall of classical bidirectional search techniques.

The implementation introduces several forms of pruning:

**Move-level pruning.** Immediate inverses are skipped. (`if m == inverse_move(last_move): continue`). These rules eliminate trivial local redundancies such as `R R'`, or `U U'`.

**State duplicate elimination.** Both functions maintain a `visited` set (`if new_state in visited: continue`) to prevent rediscovering identical states along different paths.

Despite these measures, the pruning remains insufficient. Move-level constraints only remove redundant short patterns, and the `visited` set curbs local repetition but cannot control the exponential growth of unique cube states. Indeed, even at depth 5 the algorithm reports over 9,000 states in the goal-side map. Without stronger heuristics (e.g., pattern databases or domain-specific evaluation functions), the search rapidly exhausts computational limits without finding a solution.

This case illustrates that while the model can recall canonical pruning strategies, their limited scope fails to achieve practical efficiency in large combinatorial spaces.

**Thought.**

```
The previous searches exceeded operation limits. I will switch to a meet-in-the-middle search:
- Build a dictionary of states reachable from the solved state within depth Dg=5 (using the learned
move permutations), storing the sequences from solved to each state.
- Then breadth-first search from the current state up to depth Ds=5, checking at each node if it
intersects the goal-side dictionary. If so, concatenate the forward path with the inverse of the
goal-side path to obtain a full solution.
- Apply the resulting sequence to the real cube using make_move, verify if solved, and if so call
final_answer("").

This approach keeps expansions manageable and avoids heavy recursion or long-running loops.
```

**Code.**

```python
import time

# We assume these exist from earlier:
# perm_by_move, apply_perm, inverse_move, invert_seq, solved, do_moves, get_observation

move_order = ['U',"U'",'R',"R'",'F',"F'",'L',"L'",'D',"D'",'B',"B'"]

def build_goal_map(depth_limit=5, time_limit=6.0):
    start_time = time.time()
    # Each node: (state, last_move, seq_tuple)
    start = solved
    visited = set([start])
    goal_map = {}   # state -> seq from solved to state (tuple of moves)
    frontier = [(start, None, ())]
    depth = 0
    while depth < depth_limit and frontier:
        if time.time() - start_time > time_limit:
            break
        next_frontier = []
        for state, last_move, seq in frontier:
            for m in move_order:
                if last_move is not None:
                    # prune: same face twice or immediate inverse
                    if m[0] == last_move[0] or m == inverse_move(last_move):
                        continue
                new_state = apply_perm(state, perm_by_move[m])
                if new_state in visited:
                    continue
                visited.add(new_state)
                new_seq = seq + (m,)
                goal_map[new_state] = new_seq
                next_frontier.append((new_state, m, new_seq))
        frontier = next_frontier
        depth += 1
```

```python
        print("Goal-map size (<= depth", depth_limit, "):", len(goal_map))
        return goal_map

def forward_meet_in_middle(start_state, goal_map, depth_limit=5, time_limit=6.0):
    start_time = time.time()
    if start_state in goal_map:
        # Directly within goal radius
        tail = invert_seq(list(goal_map[start_state]))
        return True, tail

    visited = set([start_state])
    frontier = [(start_state, None, ())]
    depth = 0
    while depth < depth_limit and frontier:
        if time.time() - start_time > time_limit:
            break
        next_frontier = []
        for state, last_move, seq in frontier:
            # Check meet
            if state in goal_map:
                tail = invert_seq(list(goal_map[state]))
                return True, list(seq) + tail
            for m in move_order:
                if last_move is not None:
                    if m[0] == last_move[0] or m == inverse_move(last_move):
                        continue
                new_state = apply_perm(state, perm_by_move[m])
                if new_state in visited:
                    continue
                visited.add(new_state)
                new_seq = seq + (m,)
                # Early meet check
                if new_state in goal_map:
                    tail = invert_seq(list(goal_map[new_state]))
                    return True, list(new_seq) + tail
                next_frontier.append((new_state, m, new_seq))
        frontier = next_frontier
        depth += 1
    return False, None

# Build meet-in-the-middle structures
start_state = get_observation()
print("Current state:", start_state)
goal_map = build_goal_map(depth_limit=5, time_limit=5.0)
ok, solution_seq = forward_meet_in_middle(start_state, goal_map, depth_limit=5, time_limit=5.0)

if not ok or solution_seq is None:
    print("First meet-in-the-middle attempt failed; trying slightly deeper on forward side...")
    ok, solution_seq = forward_meet_in_middle(start_state, goal_map, depth_limit=6, time_limit=6.0)

if not ok or solution_seq is None:
    print("Could not find a meet-in-the-middle solution within limits.")
    final_answer("Could not solve within limits.")
else:
    print("Found sequence length:", len(solution_seq))
    print("Sequence:", ' '.join(solution_seq))
    # Apply to real cube
    do_moves(solution_seq)
    final_state = get_observation()
    print("Solved?", final_state == solved)
    if final_state == solved:
        final_answer("")
    else:
        final_answer("Applied a sequence, but cube not solved.")
```

**Observation.**

```
Current state: RGGRRWRGWOWBGGBGBOYRYYBYRYYBRGYYGBBGWOYOOWWOOOWROWRBBW
Goal-map size (<= depth 5 ): 9732
First meet-in-the-middle attempt failed; trying slightly deeper on forward side...
<Killed by the Python Interpreter>
```

*Not solved.*

## L.5 REWARD – BYPASS VISUAL REASONING

| Model | Agent Type | Input | Reward | Depth |
|-------|-----------|-------|--------|-------|
| GPT-5 | Basic | Vertex View | Heuristic | 3 |

In this case (Step 1), the agent was required to explicitly call `get_observation()` to view the cube image. However, throughout its search process the agent did not make any use of the visual input; it only called the image observation once at the end, to display the final cube state. Instead, its entire search relied exclusively on the heuristic reward. The code shows a greedy lookahead strategy up to depth three, where move sequences were evaluated and selected solely on the basis of their reward values.

This behavior illustrates that when a simple external reward is available, even a capable agent may effectively bypass the provided visual input. Rather than engaging in visual reasoning to reconstruct the cube state, the agent defaulted to symbolic search in the move space, guided entirely by reward optimization.

*Step 1*

**Thought.**

```
Thought: I'll use greedy and limited lookahead search guided by the heuristic reward. At each iteration
I will:
- Try all 12 basic moves and pick the one with the largest positive reward; if found, apply it and
continue.
- If no single move improves, try pairs of moves (depth 2) to find a positive net improvement; if
found, apply the best pair.
- If still no improvement, try depth 3 sequences for a positive net improvement; if found, apply the
best triple.
- Repeat until no improvements are found up to depth 3 or a reasonable move limit is reached.
I will then show the current cube observation so I can visually assess if it's solved. If it is, I will
call final_answer("").
```

**Code.**

```python
# Greedy + lookahead solver using the heuristic reward

# Define moves and inverses
moves = ["F","F'","B","B'","L","L'","R","R'","U","U'","D","D'"]
inv = {m: (m[:-1] if m.endswith("'") else m + "'") for m in moves}

def try_best_sequence(max_depth=3):
    """
    Try to find the best improving sequence up to max_depth.
    Returns (best_reward, best_sequence) with best_reward > 0 if found, else (0, []).
    """
    best_reward = 0.0
    best_seq = []

    # Depth 1
    for m1 in moves:
        r1 = make_move(m1)
        # undo
        _ = make_move(inv[m1])
        if r1 > best_reward:
            best_reward = r1
            best_seq = [m1]

    if best_reward > 0 or max_depth == 1:
        return best_reward, best_seq

    # Depth 2
    for m1 in moves:
        r1 = make_move(m1)
        for m2 in moves:
            r2 = make_move(m2)
            net = r1 + r2
            # undo m2
            _ = make_move(inv[m2])
```

```python
            if net > best_reward:
                best_reward = net
                best_seq = [m1, m2]
        # undo m1
        _ = make_move(inv[m1])

    if best_reward > 0 or max_depth == 2:
        return best_reward, best_seq

    # Depth 3
    for m1 in moves:
        r1 = make_move(m1)
        for m2 in moves:
            r2 = make_move(m2)
            for m3 in moves:
                r3 = make_move(m3)
                net = r1 + r2 + r3
                # undo m3
                _ = make_move(inv[m3])
                if net > best_reward:
                    best_reward = net
                    best_seq = [m1, m2, m3]
            # undo m2
            _ = make_move(inv[m2])
        # undo m1
        _ = make_move(inv[m1])

    return best_reward, best_seq

# Main improvement loop
total_improvement = 0.0
applied_moves = []
iterations = 0
max_iterations = 200  # safety cap on number of applied moves

print("Starting heuristic-guided improvement...")
while iterations < max_iterations:
    iterations += 1
    best_r, best_seq = try_best_sequence(max_depth=3)
    if best_r <= 0 or not best_seq:
        print(f"No improving sequence found at iteration {iterations}. Stopping local search.")
        break
    # Apply the best found sequence
    seq_r_sum = 0.0
    for m in best_seq:
        r = make_move(m)
        seq_r_sum += r
        applied_moves.append(m)
    total_improvement += seq_r_sum
    print(f"Iteration {iterations}: Applied {best_seq} with net reward {seq_r_sum:.3f}. Total
    improvement: {total_improvement:.3f}")

print(f"Finished. Applied {len(applied_moves)} moves. Total heuristic improvement from start:
{total_improvement:.3f}")
print("Final observation for visual check:")
obs = get_observation()
print(obs)  # This will display the current cube in vertex view for visual verification.
```

**Observation.**

```
Starting heuristic-guided improvement...
Iteration 1: Applied ['D', "B'"] with net reward 4.000. Total improvement: 4.000
Iteration 2: Applied ["D'"] with net reward 2.000. Total improvement: 6.000
No improving sequence found at iteration 3. Stopping local search.
Finished. Applied 3 moves. Total heuristic improvement from start: 6.000
Final observation for visual check:
<PIL.Image.Image image mode=RGB size=84x84 at 0x75CD29633A50>
```

## L.6 REWARD – AS A MISLEADING SIGNAL

| Model | Agent Type | Input | Reward | Depth |
|---|---|---|---|---|
| GPT-5 | Basic | Full Symbolic | Heuristic | 3 |

This experiment shows how a simplistic external reward can misalign with an advanced agent's internal reasoning. In Steps 2–7, GPT-5 applied several non-trivial search strategies: greedy hill-climbing with depth-2 lookahead and random shaking, heuristic-guided DFS, constrained stochastic search, and macro-based methods incorporating standard Rubik's Cube algorithms (e.g., "Sexy Move" and "Sune"). However, all of these approaches were centered on maximizing the provided heuristic reward—a single numerical score intended to measure progress. The result was limited: the heuristic plateaued (no improvement in `cumulative_delta`), and the cube was never solved.

In Step 8, the agent explicitly reflected, *"The heuristic-based approaches failed to make progress... I will switch to a deterministic, constraint-driven search that directly targets subgoals (e.g., build the white cross)."* After abandoning the external reward and instead adopting a subgoal-based strategy, the agent made immediate progress (white cross success).

This case illustrates that even for highly capable agents, a simplistic reward signal may conflict with their internal problem-solving process, reducing performance until the agent reverts to its own structured reasoning.

Due to space constraints, we only include the observation outputs and code fragments of the key steps.

*Step 2*

**Thought.**

```
Thought: I will try to solve the cube using the available tools. Since we can get a heuristic reward
for each move, I'll implement a greedy hill-climbing approach guided by this reward: at each step,
evaluate all moves and choose the one that gives the highest positive improvement. If no single move
gives improvement, I'll try a depth-2 lookahead (pairs of moves). If still stuck, I'll use occasional
random \shake" moves to escape local minima. I will keep checking if the cube is solved by verifying
each face is uniform; once solved, I will call final_answer("") as instructed.
```

**Code.**

```python
from random import choice, randint, seed
seed(42)

# Helper functions
def is_solved(state):
    if not isinstance(state, str) or len(state) != 54:
        return False
    faces = [
        state[0:9],    # FRONT (R)
        state[9:18],   # BACK (G)
        state[18:27],  # LEFT (B)
        state[27:36],  # RIGHT (Y)
        state[36:45],  # UP (O)
        state[45:54],  # DOWN (W)
    ]
    return all(len(set(face)) == 1 for face in faces)

def inverse(move):
    return move[:-1] if move.endswith("'") else move + "'"

moves = ['F', "F'", 'B', "B'", 'L', "L'", 'R', "R'", 'U', "U'", 'D', "D'"]

def try_single_moves():
    # Evaluate all single moves and return the best positive move and its reward
    best_move = None
    best_reward = float('-inf')
    for m in moves:
        r = make_move(m)
        # revert
        make_move(inverse(m))
        if r > best_reward:
            best_reward = r
            best_move = m
    return best_move, best_reward

def try_depth2_moves():
    # Evaluate all pairs of moves and return the best positive pair and cumulative reward
    best_pair = None
```

```python
        best_reward = float('-inf')
        for m1 in moves:
            r1 = make_move(m1)
            for m2 in moves:
                r2 = make_move(m2)
                total = r1 + r2
                # revert m2 then m1
                make_move(inverse(m2))
                if total > best_reward:
                    best_reward = total
                    best_pair = (m1, m2)
            make_move(inverse(m1))
        return best_pair, best_reward

    def faces_solved_count(state):
        # number of fully solved faces
        faces = [
            state[0:9], state[9:18], state[18:27], state[27:36], state[36:45], state[45:54]
        ]
        return sum(1 for f in faces if len(set(f)) == 1)

    # Start from current observation
    state0 = get_observation()
    print("Initial state:", state0)
    print("Initially solved?", is_solved(state0))
    print("Initial faces solved:", faces_solved_count(state0))

    max_iterations = 600
    stagnation = 0
    moves_made = []
    cumulative_delta = 0.0

    for it in range(1, max_iterations + 1):
        # Check if solved
        st = get_observation()
        if is_solved(st):
            print(f"Solved at iteration {it} with {len(moves_made)} moves.")
            final_answer("")
            break

        # Try best single move
        m1, r1 = try_single_moves()
        if r1 > 0:
            rr = make_move(m1)
            cumulative_delta += rr
            moves_made.append(m1)
            stagnation = 0
        else:
            # Try depth-2
            pair, rpair = try_depth2_moves()
            if pair is not None and rpair > 0:
                rr1 = make_move(pair[0])
                rr2 = make_move(pair[1])
                cumulative_delta += (rr1 + rr2)
                moves_made.extend([pair[0], pair[1]])
                stagnation = 0
            else:
                # Random shake to escape local minima
                rand_moves = randint(1, 3)
                for _ in range(rand_moves):
                    m = choice(moves)
                    rr = make_move(m)
                    cumulative_delta += rr
                    moves_made.append(m)
                stagnation += 1

        # Periodic logging
        if it % 20 == 0:
            st = get_observation()
            print(f"Iter {it}: cumulative_delta={cumulative_delta:.2f}, moves={len(moves_made)},
            faces_solved={faces_solved_count(st)}")

    else:
        # If loop completes without break (not solved), print status
        st = get_observation()
        print("Not solved within iteration budget.")
        print("Faces solved:", faces_solved_count(st))
        print("Current state:", st)
```

**Observation.**

```
Initial state: BRRBRRWRRYYOGGGGGGBBWBBWGGYBYYBYYBRROOOOOOGWWROOWWYWWY
Initially solved? False
Initial faces solved: 0
Iter 20: cumulative_delta=0.00, moves=33, faces_solved=0
Iter 40: cumulative_delta=0.00, moves=71, faces_solved=0
Iter 60: cumulative_delta=1.00, moves=105, faces_solved=0
Iter 80: cumulative_delta=0.00, moves=147, faces_solved=0
Iter 100: cumulative_delta=0.00, moves=189, faces_solved=0
Iter 120: cumulative_delta=0.00, moves=230, faces_solved=0
Iter 140: cumulative_delta=0.00, moves=267, faces_solved=0
Iter 160: cumulative_delta=0.00, moves=312, faces_solved=0
Iter 180: cumulative_delta=0.00, moves=351, faces_solved=0
Iter 200: cumulative_delta=0.00, moves=387, faces_solved=0
Iter 220: cumulative_delta=0.00, moves=431, faces_solved=0
Iter 240: cumulative_delta=0.00, moves=472, faces_solved=0
Iter 260: cumulative_delta=0.00, moves=512, faces_solved=0
Iter 280: cumulative_delta=0.00, moves=555, faces_solved=0
Iter 300: cumulative_delta=0.00, moves=595, faces_solved=0
Iter 320: cumulative_delta=0.00, moves=638, faces_solved=0
Iter 340: cumulative_delta=0.00, moves=678, faces_solved=0
Iter 360: cumulative_delta=0.00, moves=718, faces_solved=0
Iter 380: cumulative_delta=0.00, moves=757, faces_solved=0
Iter 400: cumulative_delta=0.00, moves=796, faces_solved=0
Iter 420: cumulative_delta=0.00, moves=834, faces_solved=0
Iter 440: cumulative_delta=0.00, moves=877, faces_solved=0
Iter 460: cumulative_delta=0.00, moves=922, faces_solved=0
Iter 480: cumulative_delta=0.00, moves=965, faces_solved=0
Iter 500: cumulative_delta=0.00, moves=1000, faces_solved=0
Iter 520: cumulative_delta=0.00, moves=1043, faces_solved=0
Iter 540: cumulative_delta=0.00, moves=1082, faces_solved=0
Iter 560: cumulative_delta=0.00, moves=1124, faces_solved=0
Iter 580: cumulative_delta=0.00, moves=1167, faces_solved=0
Iter 600: cumulative_delta=0.00, moves=1210, faces_solved=0
```

*Step 3*

**Thought.**

```
The previous greedy random approach did not make reliable progress in the heuristic metric. I will
switch to a monotonic heuristic-guided depth-limited search: at each iteration, I will search for a
short sequence of moves (up to a given depth) whose cumulative heuristic reward is non-negative and
preferably positive. This avoids regressing in the staged heuristic and should help climb from 0 → 7.
I will implement a DFS that only accepts moves with non-negative immediate reward, avoids repeating
the same face consecutively, and finds the best cumulative gain sequence. I will apply sequences that
yield positive gain; if none found, I can apply a neutral (0 gain) sequence to reposition and try
again with deeper depth. I will keep checking for solved state, and call final_answer("") immediately
once solved.
```

*Stpe 4*

**Thought.**

```
The previous approach explored too many branches and hit the operation limit. I'll switch to a
constrained stochastic search that samples a limited number of short random sequences each iteration,
evaluates their net heuristic gain (sum of move rewards), and applies the best positive-gain sequence
if found. If none is found, I'll apply a small random \shake" to escape local minima. This keeps
computation bounded and avoids deep recursion. I'll regularly check if the cube is solved and call
final_answer("") immediately when it is.
```

*Step 5*

**Thought.**

```
The previous stochastic and exhaustive searches were too heavy and hit the operation limit without
improving the heuristic. I will switch to a lightweight, structured macro-search: try a small library
of well-known Rubik's Cube algorithms (sexy move, sledgehammer, sune, etc.). For each macro, I'll
evaluate its net heuristic gain by executing it (summing rewards). If net gain > 0, I'll keep it;
otherwise I'll undo it. This bounded, deterministic approach reduces operations while still seeking
monotonic heuristic improvements. I'll repeat this for a limited number of iterations, checking for
solved state frequently. If solved, I will immediately call final_answer("").
```

*Step 8*

**Thought.**

```
The heuristic-based approaches failed to make progress and hit operation limits. I will switch to a
deterministic, constraint-driven search that directly targets subgoals (e.g., build the white cross on
the Down face) by exploring short sequences using the environment as the cube model. The search:
- Applies moves via make_move and backtracks with inverse moves to keep the environment consistent.
- Prunes immediate inverses and same-face repetitions.
- Checks goal predicates by reading the state_string via get_observation().
I'll first try to complete the white cross (four Down edges in correct positions and orientations). If
I get the cube solved at any point, I'll immediately call final_answer("").
```

**Code.**

```python
import random

# Utilities
def is_solved(state):
    if not isinstance(state, str) or len(state) != 54:
        return False
    faces = [state[0:9], state[9:18], state[18:27], state[27:36], state[36:45], state[45:54]]
    return all(len(set(face)) == 1 for face in faces)

def faces_solved_count(state):
    faces = [state[0:9], state[9:18], state[18:27], state[27:36], state[36:45], state[45:54]]
    return sum(1 for f in faces if len(set(f)) == 1)

def inverse_move(m):
    return m[:-1] if m.endswith("'") else m + "'"

moves = ['F', "F'", 'B', "B'", 'L', "L'", 'R', "R'", 'U', "U'", 'D', "D'"]

# Target constraints helpers: (index -> expected color)
# Face centers for reference (fixed)
CENTER = { 'F': 4, 'B': 13, 'L': 22, 'R': 31, 'U': 40, 'D': 49 }
CENTER_COLOR = { 4:'R', 13:'G', 22:'B', 31:'Y', 40:'O', 49:'W' }

# For the white cross on Down (D), we require the four D edges to be placed and oriented correctly.
# Based on the provided net:
# Down face indices:
# 45 46 47
# 48 49 50
# 51 52 53
# Adjacent matching edges on side faces:
# DF: D[0][1]=46 with F[2][1]=7  -> colors W and R
# DR: D[1][2]=50 with R[2][1]=34 -> colors W and Y
# DB: D[2][1]=52 with Bk top-middle B[0][1]=10 -> colors W and G
# DL: D[1][0]=48 with L[2][1]=25 -> colors W and B

def cross_constraints():
    return {
        46: 'W', 7: 'R',
        50: 'W', 34:'Y',
        52: 'W', 10:'G',
        48: 'W', 25:'B',
    }

def edge_constraint_DF():
    return {46:'W', 7:'R'}

def edge_constraint_DR():
    return {50:'W', 34:'Y'}
```

```python
def edge_constraint_DB():
    return {52:'W', 10:'G'}

def edge_constraint_DL():
    return {48:'W', 25:'B'}

def constraints_satisfied(state, constraints):
    # constraints: dict index->color
    for idx, col in constraints.items():
        if state[idx] != col:
            return False
    return True

def get_state():
    return get_observation()

# Depth-limited DFS to find a sequence that satisfies a given constraints set at the end.
# Prunes immediate inverse and same-face repetitions.
def find_sequence_for_constraints(constraints, max_depth=7, max_expansions=120000):
    best = {'seq': None}
    counters = {'exp': 0}
    path = []

    def dfs(depth, last_move):
        if best['seq'] is not None:
            return
        if depth > max_depth or counters['exp'] >= max_expansions:
            return
        # Check immediate goal (depth can be 0 at start)
        st = get_state()
        if constraints_satisfied(st, constraints):
            best['seq'] = path.copy()
            return
        if depth == max_depth:
            return
        for m in moves:
            if last_move is not None:
                if m[0] == last_move[0]:      # avoid same face twice in a row
                    continue
                if m == inverse_move(last_move):  # avoid immediate inverse
                    continue
            r = make_move(m)
            counters['exp'] += 1
            path.append(m)
            dfs(depth + 1, m)
            if best['seq'] is not None:
                # leave state as is; sequence already applied during search? No, we applied the moves
                # during exploration.
                # We must backtrack to restore before returning control to caller
                make_move(inverse_move(m))
                path.pop()
                return
            # backtrack this move
            make_move(inverse_move(m))
            path.pop()
            if counters['exp'] >= max_expansions or best['seq'] is not None:
                return

    dfs(0, None)
    return best['seq']

# Function to attempt building the white cross iteratively
def build_white_cross():
    targets = [edge_constraint_DF(), edge_constraint_DR(), edge_constraint_DB(), edge_constraint_DL()]
    # We'll try each edge until all are solved
    attempts_per_edge = 6  # bounded attempts with small random perturbations
    for i, tgt in enumerate(targets):
        for attempt in range(attempts_per_edge):
            st = get_state()
            # If edge already solved, move to next
            if constraints_satisfied(st, tgt):
                break
            # Find a short sequence to solve this edge
            seq = find_sequence_for_constraints(tgt, max_depth=6, max_expansions=90000)
            if seq is not None:
                # Apply the sequence (it's already applied during the search? No, we backtracked; we
                # need to apply now)
                for m in seq:
                    make_move(m)
                # Verify
                st2 = get_state()
                if not constraints_satisfied(st2, tgt):
```

```
                        # If somehow not satisfied, undo and try a tiny shake
                        for m in reversed(seq):
                            make_move(inverse_move(m))
                else:
                    break
            else:
                # Small random shake to change positions
                k = random.randint(1, 2)
                last_face = None
                for _ in range(k):
                    cand = [m for m in moves if last_face is None or m[0] != last_face]
                    mv = random.choice(cand)
                    make_move(mv)
                    last_face = mv[0]
        else:
            # Failed to solve this edge within attempts
            return False
    # After solving all four individual edges, ensure full cross correctness (all 8 constraints)
    st = get_state()
    if constraints_satisfied(st, cross_constraints()):
        return True
    # If individual edges are placed but cross not fully consistent, try a short refinement to meet
    full constraints
    seq = find_sequence_for_constraints(cross_constraints(), max_depth=7, max_expansions=140000)
    if seq is not None:
        for m in seq:
            make_move(m)
        st2 = get_state()
        return constraints_satisfied(st2, cross_constraints())
    return False

# Driver: attempt to build white cross; if cube becomes solved, finalize.
random.seed(23)
state0 = get_state()
print("Start state:", state0)
print("Faces solved at start:", faces_solved_count(state0))
if is_solved(state0):
    final_answer("")

success_cross = build_white_cross()
st = get_state()
print("White cross success:", success_cross, "Faces solved after cross attempt:",
faces_solved_count(st), "State:", st)

# If by chance the cube is fully solved now, finalize.
if is_solved(st):
    final_answer("")
```

**Observation.**

```
Start state: GYOGROWRBOWWGGBORRROOWBBRGYYRWBYRWBGYYBGOOBOGRWGYWWBYY
Faces solved at start: 0
White cross success: True Faces solved after cross attempt: 0 State:
GYRGRRRRWGGWYGORBBYROGBBYBOWOWYYBYYRORGOOGBOYBWGWWWOWB
```

*Step 9*

**Thought.**

```
Thought: Great|building the white cross worked and increased the heuristic to 1. I'll continue with the
same constraint-based search approach to solve the four white corners and complete the first layer.
I'll encode the four down-layer corner constraints, then search short sequences that satisfy each
corner's constraints while pruning redundant branches. If at any time the cube becomes solved, I will
immediately call final_answer("").
```

## L.7 REWARD – AS A GUIDANCE

| Model | Agent Type | Input | Reward | Depth |
|---|---|---|---|---|
| Claude Sonnet 4 | Basic | Full Symbolic | Sticker | 2 |

In this example, the agent explicitly relies on the external reward signal to guide its search. At each step, candidate moves are evaluated by their immediate reward: negative rewards trigger an immediate undo (e.g., $U$ with reward $-12.0$), while positive rewards are retained (e.g., $D$ with reward $+12.0$). The agent's behavior shows that even a simple reward can provide effective guidance, shaping its local search trajectory and preventing it from devolving into purely random exploration.

*Step 7*

**Thought.**

```
Thought: Excellent! The L move gave us a +10 reward and we now have a better cube state. Let me check
the current state and continue looking for more positive moves.
```

**Code.**

```python
# Check current state after the L move
current_state = get_observation()
print(f"Current state after L: {current_state}")

# Count correct stickers
correct_stickers = sum(1 for i in range(54) if current_state[i] == solved_state[i])
print(f"Correct stickers now: {correct_stickers}/54")

# Continue trying more moves to see if we can improve further
remaining_moves = ["U", "D", "B"]
for move in remaining_moves:
    reward = make_move(move)
    print(f"Move {move}: reward = {reward}")

    if reward > 0:
        print(f"Keeping move {move} as it improved the cube")
        break
    else:
        # Undo the move
        opposite_move = move + "'"
        undo_reward = make_move(opposite_move)
        print(f"Undoing with {opposite_move}: reward = {undo_reward}")
```

**Observation.**

```
Current state after L: RRRRRRYYYBBBGGGGGGBBBBBBRRRYYYYYYGGGOOOOOOOOOWWWWWWWWW
Correct stickers now: 41/54
Move U: reward = -12.0
Undoing with U': reward = 12.0
Move D: reward = 12.0
Keeping move D as it improved the cube
```

*Step 8*

**Thought.**

```
Thought: Great progress! The D move gave us another +12 reward. Let me check the current state and see
how close we are to solving the cube.
```

## L.8 TRIAL-AND-ERROR TOOL LEARNING

| Model | Agent Type | Input | Reward | Depth |
|-------|-----------|-------|--------|-------|
| GPT-5 | Standard-Solver | Full Symbolic | No reward | 8 |

In this example, the agent initially attempted to convert the environment state into the solver's required format and call the planner (Step 2), but the planner returned errors due to undefined edges. In Steps 3–5, the agent experimented with alternative mappings, but all attempts failed to produce a valid planner input. To overcome this, the agent exhaustively explored all $4^6$ per-face rotation combinations to find a scheme compatible with the planner's input format, ultimately identifying a scheme compatible with the planner. Upon applying the planner-generated move sequence, the cube was successfully solved. This process exemplifies trial-and-error tool learning, where the agent iteratively experiments, identifies the constraints of the external tool, and adapts its strategy to reliably harness the planner for spatial reasoning tasks.

*Step 2*

**Thought.**

```
Thought: I have the current cube state string. Next, I will convert this environment state (color-coded
with faces in the order F,B,L,R,U,D) to the solver's required facelet format (URFDLB), using the center
colors to map colors to face letters: O->U, Y->R, R->F, W->D, B->L, G->B. I will start with the
simplest per-face orientation (top-left to bottom-right) and face order mapping: U(36{44), R(27{35),
F(0{8), D(45{53), L(18{26), B(9{17), concatenated in that order. I will call call_planner with this
string and print the solution (or error). If a solution is returned, in the following step I will parse
and apply the moves with make_move, expanding 1/2/3 quarter-turn counts into multiple calls.
```

**Code.**

```python
# Build mapping from environment color to solver face letter
color_to_face = {
    'O': 'U',   # Up
    'Y': 'R',   # Right
    'R': 'F',   # Front
    'W': 'D',   # Down
    'B': 'L',   # Left
    'G': 'B',   # Back
}

state = obs  # from previous step
# Define the index ranges for each face in the environment (face-on, row-major)
env_faces = {
    'F': list(range(0, 9)),
    'B': list(range(9, 18)),
    'L': list(range(18, 27)),
    'R': list(range(27, 36)),
    'U': list(range(36, 45)),
    'D': list(range(45, 54)),
}

# Solver requires URFDLB order
solver_face_order = ['U', 'R', 'F', 'D', 'L', 'B']

# For now, assume per-face orientation is identity (face-on row-major)
solver_indices = []
for face in solver_face_order:
    if face == 'U':
        solver_indices.extend(env_faces['U'])
    elif face == 'R':
        solver_indices.extend(env_faces['R'])
    elif face == 'F':
        solver_indices.extend(env_faces['F'])
    elif face == 'D':
        solver_indices.extend(env_faces['D'])
    elif face == 'L':
        solver_indices.extend(env_faces['L'])
    elif face == 'B':
        solver_indices.extend(env_faces['B'])

# Convert environment colors to solver face-letter string in the computed order
```

```python
try:
    solver_str = ''.join(color_to_face[state[i]] for i in solver_indices)
except KeyError as e:
    print("Encountered unknown color in state:", e)
    solver_str = None

print("Proposed solver string:", solver_str)

if solver_str is not None:
    plan = call_planner(tool_cube_representation=solver_str)
    print("Planner output:", plan)
```

**Observation.**

```
Execution logs:
Proposed solver string: DRFFUDFULBRRDRRDBBDFUDFBDFBBDRBDULLUFLLFLLFLLUURRBURBU
Planner output: Error: Some edges are undefined.
Last output from code snippet:
Error: Some edges are undefined.
```

*Step 6*

**Thought.**

```
My previous edge-validation used sorted two-letter strings but compared them to unsorted expectations,
which made all candidates fail.

I will fix the validation by comparing to the alphabetically sorted set of expected edge labels. Then
I'll brute-force face rotations, find a valid scheme, call the planner, and verify the cube is solved.
```

**Code.**

```python
# Fix edge validation and brute-force search for a valid per-face rotation scheme, then plan and apply
moves.

state = obs  # current cube state string

# Environment face indices (row-major as per problem)
env_faces = {
    'F': list(range(0, 9)),
    'B': list(range(9, 18)),
    'L': list(range(18, 27)),
    'R': list(range(27, 36)),
    'U': list(range(36, 45)),
    'D': list(range(45, 54)),
}

def rotate_indices(face_indices, rotation):
    # rotation: 0 (identity), 1 (90° CW), 2 (180°), 3 (270° CW)
    if rotation == 0:
        order = [0,1,2,3,4,5,6,7,8]
    elif rotation == 1:
        order = [6,3,0,7,4,1,8,5,2]
    elif rotation == 2:
        order = [8,7,6,5,4,3,2,1,0]
    elif rotation == 3:
        order = [2,5,8,1,4,7,0,3,6]
    else:
        raise ValueError("Invalid rotation")
    return [face_indices[i] for i in order]

# Build color_to_face mapping from centers
centers = {
    'F': state[4],
    'B': state[9+4],
    'L': state[18+4],
    'R': state[27+4],
```

```python
    'U': state[36+4],
    'D': state[45+4],
}
color_to_face = {
    centers['U']: 'U',
    centers['R']: 'R',
    centers['F']: 'F',
    centers['D']: 'D',
    centers['L']: 'L',
    centers['B']: 'B'
}
print("Centers:", centers)
print("Color->Face:", color_to_face)

# Precompute rotated indices for each face and rotation to speed up
pre_rot = {face: {r: rotate_indices(env_faces[face], r) for r in range(4)} for face in
['U','R','F','D','L','B']}

def build_solver_str_fast(scheme):
    idx = []
    idx += pre_rot['U'][scheme['U']]
    idx += pre_rot['R'][scheme['R']]
    idx += pre_rot['F'][scheme['F']]
    idx += pre_rot['D'][scheme['D']]
    idx += pre_rot['L'][scheme['L']]
    idx += pre_rot['B'][scheme['B']]
    return ''.join(color_to_face[state[i]] for i in idx)

def face_base(face_letter):
    base = {'U':0,'R':9,'F':18,'D':27,'L':36,'B':45}
    return base[face_letter]

def get_f(s, face, pos):
    return s[face_base(face)+pos]

# Edge definitions in Kociemba URFDLB indexing (face-local 0..8)
edges_def = [
    (('U',7), ('F',1)),   # UF
    (('U',5), ('R',1)),   # UR
    (('U',1), ('B',1)),   # UB
    (('U',3), ('L',1)),   # UL
    (('D',1), ('F',7)),   # DF
    (('D',5), ('R',7)),   # DR
    (('D',7), ('B',7)),   # DB
    (('D',3), ('L',7)),   # DL
    (('F',5), ('R',3)),   # FR
    (('F',3), ('L',5)),   # FL
    (('B',3), ('R',5)),   # BR
    (('B',5), ('L',3)),   # BL
]

# Expected edges as alphabetically sorted two-letter strings
expected_sorted_edges = set(['FU','RU','BU','LU','DF','DR','BD','DL','FR','FL','BR','BL'])

def edges_valid(s):
    # Build sorted two-letter strings for each edge pair
    counts = {}
    for (f1,p1),(f2,p2) in edges_def:
        a = get_f(s, f1, p1)
        b = get_f(s, f2, p2)
        pair = ''.join(sorted([a,b]))
        if pair not in expected_sorted_edges:
            return False
        counts[pair] = counts.get(pair, 0) + 1
    # Ensure each expected edge appears exactly once
    for e in expected_sorted_edges:
        if counts.get(e, 0) != 1:
            return False
    return True

found_scheme = None
found_solver_str = None

# Brute force all 4^6 rotation combinations (4096)
outer_break = False
for u in range(4):
    if outer_break: break
    for r in range(4):
        if outer_break: break
        for f in range(4):
            if outer_break: break
            for d in range(4):
                if outer_break: break
```

```
                    for l in range(4):
                        if outer_break: break
                        for b in range(4):
                            scheme = {'U':u,'R':r,'F':f,'D':d,'L':l,'B':b}
                            s = build_solver_str_fast(scheme)
                            if edges_valid(s):
                                found_scheme = scheme
                                found_solver_str = s
                                outer_break = True
                                break

print("Found scheme:", found_scheme)
print("Solver string preview:", found_solver_str[:30] if found_solver_str else None)

if found_solver_str:
    plan = call_planner(tool_cube_representation=found_solver_str)
    print("Planner output:", plan)
```

**Observation.**

```
Centers: {'F': 'R', 'B': 'G', 'L': 'B', 'R': 'Y', 'U': 'O', 'D': 'W'}
Color->Face: {'O': 'U', 'Y': 'R', 'R': 'F', 'W': 'D', 'B': 'L', 'G': 'B'}
Found scheme: {'U': 0, 'R': 0, 'F': 0, 'D': 0, 'L': 0, 'B': 2}
Solver string preview: DRFFUDFULBRRDRRDBBDFUDFBDFBBDR
Planner output: R2 L1 B2 R1 U1 R1 B1 U2 D2 F2 L2 F2 U1 D2 (14f)
```

*Then executed the solution sequence and solved at Step 7.*

## L.9 BYPASSING VISUAL TASKS VIA SYMBOLIC REASONING

We illustrate the performance gap between *Face View* and *Vertex View* inputs using two representative cases.

In the *Face View* setting, the cube is presented as regular $3 \times 3$ grids. The agent reframed the task as a symbolic parsing problem: mapping each face's colors into the URFDLB string required by the solver. This bypassed visual reasoning, yet worked reliably due to the input's highly ordered structure. The planner accepted the constructed string and returned a valid solution, which the agent executed successfully.

In the *Vertex View* setting, these distorted layouts cannot be parsed directly into solver input without geometric inference. The agent nevertheless attempted symbolic reconstruction via pixel sampling, palette classification, k-means clustering, PCA ordering, and multi-view accumulation, but the resulting cube string was incomplete.

These two cases highlight a key difference between the tasks. In *Face View*, the orderly layout enables a symbolic parsing shortcut that bypasses genuine visual reasoning. In *Vertex View*, this strategy is infeasible, and performance drops sharply. This suggests that even highly capable models tend to prefer symbolic over visual reasoning whenever possible, and that less regular task formats are informative for exposing their limitations in spatial understanding.

### L.9.1 FULL VIEW

| Model | Agent Type | Input | Reward | Depth |
|-------|-----------|-------|--------|-------|
| GPT-5 | Ideal-Solver | Full Visual State | no reward | 20 |

*Step 7*

**Thought.**

Our previous classification mistakenly excluded near-white pixels, undercounting W. I will recompute the sticker colors by averaging patches around each sticker center while only ignoring the gray background and black grid lines (not near-white), then classify to the nearest canonical color. I'll rebuild the 54-character state string, ensure it has 9 of each color, call the planner, expand and apply the moves, and finally submit final_answer("").

**Code.**

```python
# Recompute colors including white pixels properly (ignore only gray background and black grid).
img = get_observation()
w, h = img.size
BG = (50,50,50)
BLACK = (0,0,0)

def avg_color_around_include_white(p, r=6):
    cx, cy = p
    samples = []
    for dy in range(-r, r+1):
        y = cy + dy
        if y < 0 or y >= h:
            continue
        for dx in range(-r, r+1):
            x = cx + dx
            if x < 0 or x >= w:
                continue
            rgb = img.getpixel((x,y))
            if rgb == BG or rgb == BLACK:
                continue
            samples.append(rgb)
    if not samples:
        samples = [img.getpixel((cx, cy))]
    sr = sum(p[0] for p in samples)/len(samples)
    sg = sum(p[1] for p in samples)/len(samples)
    sb = sum(p[2] for p in samples)/len(samples)
    return (sr, sg, sb)

canon = {
    'R': (255, 0, 0),
    'G': (0, 255, 0),
    'B': (0, 0, 255),
    'Y': (255, 255, 0),
    'O': (255, 165, 0),
    'W': (255, 255, 255),
}

def classify_avg(rgbf):
    r,g,b = rgbf
    best=None; bd=1e18
    for k,v in canon.items():
        d=(r-v[0])**2+(g-v[1])**2+(b-v[2])**2
        if d<bd:
            bd=d; best=k
    return best

def colors_from_grid_patch_inc_white(grid_points):
    return [classify_avg(avg_color_around_include_white(p, r=6)) for p in grid_points]

# Use previously computed grids U_grid, F_grid, D_grid, B_grid, L_grid, R_grid
U_colors = colors_from_grid_patch_inc_white(U_grid)
F_colors = colors_from_grid_patch_inc_white(F_grid)
D_colors = colors_from_grid_patch_inc_white(D_grid)
B_colors = colors_from_grid_patch_inc_white(B_grid)
L_colors = colors_from_grid_patch_inc_white(L_grid)
R_colors = colors_from_grid_patch_inc_white(R_grid)

from collections import Counter
state = ''.join(F_colors + B_colors + L_colors + R_colors + U_colors + D_colors)
print("Color counts:", Counter(state), "length:", len(state))
print("State:", state)

# Call planner
plan = call_golden_planner(tool_cube_representation=state)
print("Planner:", plan)

# Parse and expand moves like 'L3 U1 B1 R2 ... (19f)'
import re
tokens = re.findall(r'([URFDLB])([123])', plan)
```

```
expanded = []
for face, num in tokens:
    n = int(num)
    if n == 1:
        expanded.append(face)
    elif n == 2:
        expanded.extend([face, face])
    elif n == 3:
        expanded.extend([face, face, face])
print("Expanded moves:", expanded)

# Apply moves
for mv in expanded:
    print(make_move(move=mv))

# Submit final check
final_answer("")
```

**Observation.**

```
Color counts: Counter({'W': 9, 'Y': 9, 'R': 9, 'B': 9, 'O': 9, 'G': 9}) length: 54
State: WYYYRBOWGYWRWGBRWGOOYGBRWOYOOORYOBGBBYBROGGGRGBWYWBRRW
Planner: U3 F2 R2 U2 F2 D2 F2 L1 B3 R1 U2 F3 D3 L3 B3 U2 L2 R1 D3 U1 (20f)
Expanded moves: ['U', 'U', 'U', 'F', 'F', 'R', 'R', 'U', 'U', 'F', 'F', 'D', 'D', 'F', 'F', 'L', 'B',
'B', 'B', 'R', 'U', 'U', 'F', 'F', 'D', 'D', 'D', 'L', 'L', 'L', 'B', 'B', 'B', 'U', 'U', 'L',
'L', 'R', 'D', 'D', 'D', 'U']
Applied move `U` on the cube
Applied move `U` on the cube
Applied move `U` on the cube
Applied move `F` on the cube
Applied move `F` on the cube
Applied move `R` on the cube
// ...
Applied move `R` on the cube
Applied move `D` on the cube
Applied move `D` on the cube
Applied move `D` on the cube
Applied move `U` on the cube
```

*Solved.*

### L.9.2   VERTEX VIEW

| Model | Agent Type | Input | Reward | Depth |
|-------|-----------|-------|--------|-------|
| GPT-5 | Standard-Solver | Vertex View | no reward | 20 |

*Step 7*

**Thought.**

```
The previous approach hit the operation limit due to heavy k-means and broad view sweeps, and also
misclassified colors. I'll switch to a lighter pipeline:
- Classify colors by nearest reference palette (O/W/R/G/B/Y) for robustness.
- Sample far fewer pixels (stride=5) and run k-means with fewer iterations (k=27, iters=8) to find
sticker centers in each view.
- Group the 27 clusters spatially into 3 faces, order each face's 3x3 via PCA, and use the center
sticker to name the face.
- Collect exactly 5 views (right, right, down, right, right) to cover all faces.
- Convert the six color grids into the URFDLB string (O,Y,R,W,B,G mapped to U,R,F,D,L,B), call the
planner, apply the moves, then check solved state.

If the planner errors (mapping inconsistent), I'll print the composed string to debug. Otherwise I'll
execute the plan and finish.
```

**Code.**

```python
from collections import defaultdict
from math import sqrt

# Nearest-palette color mapping for robustness
PALETTE = {
    'O': (255, 140, 0),    # orange (slightly darker than 165 to avoid yellow confusion)
    'W': (255, 255, 255),
    'R': (220, 0, 0),
    'G': (0, 200, 0),
    'B': (0, 0, 220),
    'Y': (255, 255, 0),
}

def rgb_to_color_letter(rgb):
    r,g,b = rgb
    # ignore mid-gray detection here; caller ensures colored pixel
    best = None
    bestd = 1e18
    for k,(pr,pg,pb) in PALETTE.items():
        d = (r-pr)**2 + (g-pg)**2 + (b-pb)**2
        if d < bestd:
            bestd = d; best = k
    return best

def is_colored_pixel(p):
    r,g,b = p
    m = max(r,g,b); n = min(r,g,b)
    # reject dark (grid/outline)
    if m < 60:
        return False
    # reject mid-gray background (low saturation, mid-high brightness)
    if (m-n) < 25 and 120 <= m <= 230 and 120 <= n <= 230:
        return False
    # accept bright white
    if r+g+b > 700:
        return True
    # accept sufficiently saturated colors
    if (m-n) >= 35:
        return True
    return False

def sample_colored_pixels(img, stride=5, max_samples=1500):
    px = img.load()
    w,h = img.size
    pts, cols = [], []
    cnt = 0
    for y in range(0,h,stride):
        for x in range(0,w,stride):
            p = px[x,y]
            if is_colored_pixel(p):
                pts.append((x,y))
                cols.append(p)
                cnt += 1
                if cnt >= max_samples:
                    return pts, cols
    return pts, cols

def kmeans_points(points, k, iters=8):
    if not points or len(points) < k:
        return None, None, None
    # init farthest-point
    centers = [points[0]]
    for _ in range(1,k):
        best = None; bestd = -1
        for p in points:
            dmin = min((p[0]-c[0])**2+(p[1]-c[1])**2 for c in centers)
            if dmin > bestd:
                bestd = dmin; best = p
        centers.append(best)
    assigns = [0]*len(points)
    for _ in range(iters):
        changed = False
        for i,p in enumerate(points):
            ci = min(range(k), key=lambda j: (p[0]-centers[j][0])**2+(p[1]-centers[j][1])**2)
            if assigns[i] != ci:
                assigns[i] = ci; changed = True
        if not changed:
            break
        newc = []
        for j in range(k):
            xs=[points[i][0] for i in range(len(points)) if assigns[i]==j]
            ys=[points[i][1] for i in range(len(points)) if assigns[i]==j]
```

```python
            if xs:
                newc.append((sum(xs)/len(xs), sum(ys)/len(ys)))
            else:
                newc.append(centers[j])
        centers = newc
    clusters = [[] for _ in range(k)]
    for i in range(len(points)):
        clusters[assigns[i]].append(i)
    return centers, assigns, clusters

def majority_color(color_list):
    counts = defaultdict(int)
    for rgb in color_list:
        c = rgb_to_color_letter(rgb)
        counts[c] += 1
    return max(counts.items(), key=lambda kv: kv[1])[0] if counts else 'W'

def extract_27_stickers(img):
    pts, cols = sample_colored_pixels(img, stride=5, max_samples=1500)
    print("Sampled colored:", len(pts))
    res = kmeans_points(pts, 27, iters=8)
    if res[0] is None:
        return []
    _, assigns27, clusters27 = res
    stickers = []
    for j in range(27):
        idxs = clusters27[j]
        if not idxs:
            continue
        cx = sum(pts[i][0] for i in idxs)/len(idxs)
        cy = sum(pts[i][1] for i in idxs)/len(idxs)
        col = majority_color([cols[i] for i in idxs])
        stickers.append(((cx,cy), col))
    return stickers

def pca_axes(points):
    if not points:
        return (1.0,0.0),(0.0,1.0)
    mx = sum(p[0] for p in points)/len(points)
    my = sum(p[1] for p in points)/len(points)
    sxx = sum((p[0]-mx)**2 for p in points)/len(points)
    syy = sum((p[1]-my)**2 for p in points)/len(points)
    sxy = sum((p[0]-mx)*(p[1]-my) for p in points)/len(points)
    tr = sxx+syy
    det = sxx*syy - sxy*sxy
    disc = max(tr*tr/4 - det, 0.0)
    lam1 = tr/2 + sqrt(disc)
    if abs(sxy) + abs(lam1 - sxx) < 1e-9:
        u=(1.0,0.0)
    else:
        u=(sxy, lam1 - sxx)
        n = sqrt(u[0]*u[0]+u[1]*u[1])
        u=(u[0]/n, u[1]/n)
    v=(-u[1], u[0])
    return u,v

def kmeans_scalar(values, k=3, iters=8):
    # 1D kmeans implemented via 2D wrapper to reuse
    pts = [(v,0.0) for v in values]
    _, assigns, clusters = kmeans_points(pts, k, iters)
    return assigns, clusters

def group_into_faces(stickers):
    centers=[s[0] for s in stickers]
    res = kmeans_points(centers, 3, iters=8)
    if res[0] is None:
        return []
    _, assigns3, clusters3 = res
    faces=[]
    for j in range(3):
        idxs = clusters3[j]
        group = [stickers[i] for i in idxs]
        if len(group) >= 7:
            faces.append(group)
    return faces

def order_face_grid(face_stickers):
    pts=[s[0] for s in face_stickers]
    u,v = pca_axes(pts)
    proj=[]
    for i,(pt,col) in enumerate(face_stickers):
        pu = pt[0]*u[0] + pt[1]*u[1]
        pv = pt[0]*v[0] + pt[1]*v[1]
```

```python
        proj.append((pu,pv,i))
    col_vals = [p[0] for p in proj]
    row_vals = [p[1] for p in proj]
    cols_assigns, _ = kmeans_scalar(col_vals, 3, iters=8)
    rows_assigns, _ = kmeans_scalar(row_vals, 3, iters=8)
    col_groups = defaultdict(list); row_groups = defaultdict(list)
    for idx,(pu,pv,i) in enumerate(proj):
        col_groups[cols_assigns[idx]].append(i)
        row_groups[rows_assigns[idx]].append(i)
    ordered_cols = sorted(col_groups.keys(), key=lambda c: sum(face_stickers[i][0][0] for i in
    col_groups[c])/len(col_groups[c]))
    ordered_rows = sorted(row_groups.keys(), key=lambda r: sum(face_stickers[i][0][1] for i in
    row_groups[r])/len(row_groups[r]))
    grid = [[None]*3 for _ in range(3)]
    for idx,(pu,pv,i) in enumerate(proj):
        rlab = rows_assigns[idx]; clab = cols_assigns[idx]
        r = ordered_rows.index(rlab); c = ordered_cols.index(clab)
        grid[r][c] = face_stickers[i][1]
    return grid

def faces_from_view(img):
    stickers = extract_27_stickers(img)
    print("Sticker clusters:", len(stickers))
    faces_groups = group_into_faces(stickers)
    print("Faces in this view:", len(faces_groups))
    faces = {}
    for group in faces_groups:
        grid = order_face_grid(group)
        center = grid[1][1]
        faces[center] = {"grid": grid}
    return faces

def accumulate_faces(acc, new_faces):
    for k,v in new_faces.items():
        if k not in acc:
            acc[k]=v

# Collect faces from a fixed small sequence of views
observed = {}
img = get_observation()
accumulate_faces(observed, faces_from_view(img))
apply_view_transformation("view_right")
img = get_observation()
accumulate_faces(observed, faces_from_view(img))
apply_view_transformation("view_right")
img = get_observation()
accumulate_faces(observed, faces_from_view(img))
apply_view_transformation("view_down")
img = get_observation()
accumulate_faces(observed, faces_from_view(img))
apply_view_transformation("view_right")
img = get_observation()
accumulate_faces(observed, faces_from_view(img))

print("Collected faces:", list(observed.keys()))
for k in observed:
    print(k, observed[k]["grid"])

# Map to solver's URFDLB letters
color_to_face_letter = {'O':'U','W':'D','R':'F','G':'B','B':'L','Y':'R'}

def grid_to_str(grid):
    return ''.join(''.join(grid[r][c] for c in range(3)) for r in range(3))

def compose_URFDLB(obs):
    # URFDLB color centers order: O (U), Y (R), R (F), W (D), B (L), G (B)
    order = ['O','Y','R','W','B','G']
    s=""
    for ck in order:
        if ck not in obs:
            print("Warning missing face:", ck)
            grid = [[ck]*3 for _ in range(3)]
        else:
            grid = obs[ck]["grid"]
        mapped = [[color_to_face_letter.get(ch,'U') for ch in row] for row in grid]
        s += grid_to_str(mapped)
    return s

solver_str = compose_URFDLB(observed)
print("Solver string:", solver_str)

plan = call_planner(tool_cube_representation=solver_str)
print("Planner output:", plan)
```

```python
def execute_plan(plan_str):
    toks = plan_str.strip().split()
    for t in toks:
        if '(' in t:  # end marker with "(xf)"
            break
        face = t[0]
        num = t[1] if len(t) > 1 else '1'
        if num == '1':
            print(make_move(face))
        elif num == '2':
            print(make_move(face))
            print(make_move(face))
        elif num == '3':
            print(make_move(face + "'"))

if plan and not plan.strip().startswith("Error"):
    execute_plan(plan)

# Final check
final_answer("")
```

**Observation.**

```
Sampled colored: 106
Sticker clusters: 27
Faces in this view: 3
Sampled colored: 106
Sticker clusters: 27
Faces in this view: 3
Sampled colored: 106
Sticker clusters: 27
Faces in this view: 3
Sampled colored: 103
Sticker clusters: 27
Faces in this view: 2
Sampled colored: 102
Sticker clusters: 27
Faces in this view: 2
Collected faces: ['B', 'R', None, 'O', 'G', 'W']
B [['Y', 'Y', 'O'], ['O', 'B', 'O'], [None, 'B', None]]
R [[None, 'Y', 'W'], ['R', 'R', 'Y'], [None, None, 'B']]
None [['R', None, 'G'], ['W', None, None], ['G', 'Y', 'Y']]
O [['R', 'R', 'O'], ['G', 'O', 'B'], [None, 'G', None]]
G [['B', 'G', 'W'], ['R', 'G', 'W'], ['R', 'O', None]]
W [['W', 'W', 'Y'], ['Y', 'W', 'W'], ['B', 'Y', None]]
Warning missing face: Y
Solver string: FFUBULUBURRRRRRRRRURDFFRUULDDRRDDLRURRUULUULULBDFBDFUU
Planner output: Error: Cube definition string FFUBULUBURRRRRRRRRURDFFRUULDDRRDDLRURRUULUULULBDFBDFUU
does not contain exactly 9 facelets of each color.
```

*Not Solved.*

