# OpenReview forum: "CubeBench: Diagnosing Interactive, Long-Horizon Physical Intelligence under Partial Observations"
_ICLR.cc/2026/Conference — ICLR 2026 Poster_

### Official Review · Reviewer_tzUm · 2025-10-30

**Soundness:** 3
**Presentation:** 3
**Contribution:** 3
**Rating:** 4
**Confidence:** 4

**Summary:**

This paper introduces an interesting benchmark based on Rubik's Cube for evaluating the capability of LLMs. The authors introduced three tracks of tasks: full symbolic, visual, and partial visual. The authors have conducted solid evaluations of LLMs and have also discussed some insightful findings.

**Strengths:**

- The benchmark is useful in evaluating long-horizon, interactive, and spatial reasoning capabilities of LLMs.
- The presentation of this paper is good, with clear figures and well-written paragraphs.
- The evaluation of different LLMs seems to be extensive.

**Weaknesses:**

1. Lacks performance comparison to classical methods.
Unlike ARC [1], I think we should have some existing symbolic tools to solve the Rubik's Cube (correct me if I am wrong). What's the performance of them in this benchmark? And "why" do we need to consider the problem of "LLMs can solve them or not" if symbolic solvers are already very good at this task? (as against in ARC, we don't have performant classical program synthesis methods while human can easily achieve >90%).

2. I am a bit confused by the visual and partial visual tier tasks.
- For visual tasks, if the image is simply generated with colored grids without any noise, isn't it the same as a symbolic state? As we won't have a grounding issue like discussed in LogiCity [2].
- For partial visual tasks, I think the agent can easily get the full state by simply turning the columns one by one and rotating them back to the init state, right? Since the Rubik's Cube is deterministic, if LLMs do this, isn't this the same as full-observability? Usually, POMDP is hard because it is very hard/impossible/very inefficient for the agent to get full states, like in LogiCity [2], so I am a bit confused by the setting here. Why is the full state hard to observe?

I will consider raising my score if these two concerns are resolved during rebuttal.

[1] Chollet, Francois, et al. "Arc prize 2024: Technical report." arXiv preprint arXiv:2412.04604 (2024).
[2] Li, Bowen, et al. "LogiCity: Advancing neuro-symbolic ai with abstract urban simulation." Advances in Neural Information Processing Systems 37 (2024): 69840-69864.

**Questions:**

See weaknesses

---

> ### Author Response · Authors · 2025-11-20
> **Rebuttal**
>
> **[W1]**
>
> We thank the reviewer for this insightful comment. Our goal with *CubeBench* is not to compete with classical Rubik’s Cube solvers, but to use the Cube as a **diagnostic environment** for evaluating three essential reasoning capabilities in LLMs—**3D spatial understanding, long-horizon planning, and exploration under partial observation**. These dimensions were also highlighted by Reviewer (#aJqe, #zB6V, #kiAJ) as key cognitive gaps in current models.
>
> Classical solvers indeed achieve near-100% accuracy using domain-specific algorithms and search **heuristics**. For example, the well-known **Kociemba’s two-phase algorithm** [Kociemba, 1997] deterministically solves any cube in a certain number of moves, and learning-based solvers such as **DeepCubeA** [Agostinelli et al., 2019] reach perfect success via A*-guided search with **heuristic** value functions. However, these methods rely on **explicitly encoded symbolic states** and handcrafted domain knowledge. They cannot generalize to natural language or visual inputs, nor can they reason under perceptual **uncertainty**.
>
> In contrast, *CubeBench* examines whether an **LLM—without domain-specific priors—**can perceive, reason, and act coherently across symbolic and perceptual modalities. Even under the same deterministic dynamics, the task becomes substantially harder when expressed in natural language or images. In our experiments, **GPT-5’s performance is only comparable to a shallow MLP policy trained with policy gradient**, indicating that LLMs fail to transfer their memorized algorithmic knowledge (e.g., CFOP-style **heuristics**) into consistent multi-step planning.
>
> We further emphasize that this benchmark is not intended to demonstrate “who solves the cube faster,” but to **diagnose why LLMs fail where classical solvers trivially succeed**. By recognizing the Rubik’s Cube as a structured but interpretable domain, *CubeBench* isolates the reasoning bottlenecks that currently limit embodied AI systems—exactly the motivation behind evaluating LLMs rather than symbolic solvers.

---

> > ### Author Response · Authors · 2025-11-20
> > **Rebuttal cont.**
> >
> > **[W2]**
> >
> > **(A) Why is the visual tier not just a symbolic state in disguise?**
> >
> > We agree that, from an information-theoretic **point of view**, a clean rendered cube image and a symbolic facelet string can encode the same underlying state. However, they probe **different subsystems** of the model:
> >
> > - **In the symbolic tier**, the model receives a fully structured, serialized state representation where each token already corresponds to a specific facelet index. No perception or parsing is required.
> >
> > - **In the visual tier**, the model must first **construct** this internal structure from unstructured pixels: it has to segment the cube, cluster colors, assign each sticker to a face, and map the 2D layout to 3D adjacency before any planning can begin.
> >
> > Thus, while the **mutual information** between the observation and the true cube state is (in principle) comparable, the **computational pipeline** is not: the visual tier specifically probes the limits of the model’s **vision, grounding, and cross-modal alignment modules**, rather than its symbolic search alone. In practice, we observe consistent errors in this pipeline (mis-assigning faces, inconsistent orientations, dropped stickers, etc.), which never occur in the symbolic tier and which are exactly the kind of “**grounding**” failures we aim to diagnose.
> >
> >
> > **(B) Why is the partial-visual tier genuinely challenging if the cube is deterministic?**
> >
> > Yes, the agent can get the full state by simply turning the columns one by one and rotating them back to the init state. This is **precisely** why we designed the tier this way. The partial-visual setting is intended to simulate **human-like interaction with limited field-of-view**, where the agent:
> >
> > - only sees a few faces at once,
> > - must understand how rotating one face affects the **global** configuration, and
> > - must mentally track the full cube while exploring and then undo its exploratory moves.
> >
> > In other words, the tier is not meant to make the full state **unreachable**; it is meant to stress the model’s ability to **track a latent world state over a long action horizon** under partial observations. This is directly targeting the “long-horizon state tracking through mental simulation” capability we highlight in the paper, where most models fail, even though, as the reviewer notes, a conceptually simple “scan–reconstruct–rewind” strategy would suffice. This suggests that the bottleneck is not exploration per se, but the inability to:
> >
> > - maintain a consistent 3D **mental model** over many steps,
> > - correctly predict the effect of moves on **unseen** faces, and
> > - reliably reverse a long sequence of rotations.
> >
> > P.S. We appreciate the reviewer highlighting LogiCity [2]. We find it a very valuable and complementary benchmark, and we will add a more detailed discussion in the related work section. In particular, we see LogiCity offers **long-horizon POMDP tasks with multi-agent interactions** (Safe Path Following) and **high-dimensional visual reasoning tasks** (Visual Action Prediction), highlighting how NeSy methods can outperform purely neural baselines on abstract reasoning. We will revise the related work section to explicitly position *CubeBench* alongside LogiCity, emphasizing this complementary relationship and the different aspects of partial observability that each benchmark targets.

---

### Official Review · Reviewer_kiAJ · 2025-10-30

**Soundness:** 3
**Presentation:** 3
**Contribution:** 3
**Rating:** 6
**Confidence:** 3

**Summary:**

The paper proposed an benchmark named CubeBench. It is a Rubik’s-cube benchmark for LLMs that probes three weak spots:
- 3-D spatial reasoning
- Long-horizon state tracking
- Active exploration under partial views

It offers three input tiers: (1) full 54-char state, (2) 2-D unfolded map, (3) sparse local views. The authors also try to give dense rewards and use an external solver.

Tests on 15+ top models show: everyone fails long scrambles, accuracy tanks when switching from symbols to visuals, dense rewards help only tiny tasks, and even with a solver models can’t reliably rebuild the cube. All code and task generators are open-sourced.

**Strengths:**

1.  An interesting task that can test  LLM's spatial reasoning, long horizon, and expoloration abilities.
2.  Large-scale comparison across 15+ proprietary/open LLMs yields actionable insights for the community.
3.  Inclusion of dense-reward variants and solver tools makes the dataset usable for both RL and tool-augmented-agent research.

**Weaknesses:**

1.  Scope limited to a single puzzle; conclusions may not generalize to broader physical reasoning tasks which can be really useful in the real world.

**Questions:**

If an LLM/VLM works well on CubeBench, can it translate into tangible gains on real-world 3-D embodied tasks (e.g., physical-robot manipulation)? Or can you explain the correlation between CubeBench scores and actual physical-world capabilities?

---

> ### Author Response · Authors · 2025-11-20
> **Rebuttal**
>
> We thank the reviewer for raising the important question about the scope of *CubeBench* and its relationship to broader 3-D embodied reasoning. We fully agree that real-world physical reasoning involves much richer dynamics, perception, and interaction than a single puzzle. Our goal is not to claim that “solving Rubik’s Cube = solving robotics,” but to propose *CubeBench* as a **controlled diagnostic environment** that isolates a small set of core cognitive abilities which many recent 3-D embodied benchmarks have independently identified as current bottlenecks for LLM/MLLM-based agents.
>
>
> **(1) Why a single puzzle can still be informative for physical reasoning?**
>
> The Rubik’s Cube is admittedly a simplified domain: it is a single rigid object with deterministic kinematics and no contact physics, deformable objects, or multi-agent interactions. However, it concentrates three fundamental reasoning abilities that are required in virtually all 3-D embodied tasks:
>
> 1. **3D spatial understanding:** Rotations on the cube require keeping track of a global 3-D coordinate frame and how local face rotations reconfigure it. This is the same type of spatial transform reasoning needed in manipulation, pose tracking, and multi-view navigation.
>
> 2. **Long-horizon planning:** Even modest scrambles require 8–20 non-commutative moves; small mistakes accumulate and cannot be locally “patched.” This mirrors the challenge of long-horizon, non-myopic planning in rearrangement and multi-step manipulation.
>
> 3. **Belief-state tracking under partial observability:** Humans almost never see all 6 faces at once, yet can reason about unseen stickers via mental simulation. Our partial-visual tier explicitly stresses this: the agent must explore, maintain an internal world state, and then undo exploratory actions correctly.
>
> In other words, *CubeBench* deliberately removes many **environmental** complexities (multi-object scenes, noisy sensors, actuation limits) so that we can precisely stress **cognitive** components that are known to be bottlenecks in more complex embodied settings: spatial world modeling, long-horizon mental simulation, and belief-state tracking.
>
>
> **(2) Consistency with recent 3-D embodied benchmarks**
>
> Actually, conclusions drawn from *CubeBench* align with those from broader 3-D embodied benchmarks. A number of very recent works report **highly consistent limitations** of LLM/MLLM-based embodied agents:
>
> - **EmbodiedBench** (ICML 2025, Oral) evaluates 13 multi-modal LLMs on 1,128 tasks across four environments and six capability-oriented subsets. The authors find that MLLMs “excel at high-level tasks but struggle with low-level manipulation,” and that **long-horizon planning is the most challenging subset**, with large performance drops compared to base tasks. For example, GPT-4o achieves 57.7% success on navigation but only 28.9% on low-level manipulation, and long-horizon subsets consistently show the largest degradation.
>
> - **Embodied Agent Interface (EAI)** (NeurIPS 2024 D&B) explicitly argues that most embodied evaluations “rely solely on a final success rate, making it difficult to pinpoint what ability is missing in LLMs and where the problem lies,” and therefore introduces a modular interface and fine-grained error types (hallucination errors, affordance errors, multiple planning error categories) precisely to move from *performance* to *diagnosis*.
>
> - **3DLLM-Mem** (2025) focuses on long-term spatial-temporal memory in Habitat-based multi-room 3-D environments. The authors conclude that current models “struggle to effectively plan and act in dynamic, multi-room 3D environments” and attribute this to a lack of proper 3-D spatial-temporal memory modeling, which is the ability of belief-state tracking in our paper.
>
> Taken together, these works converge on the same qualitative message as *CubeBench*: LLM/MLLM-based embodied agents systematically fail on tasks that require 3-D spatial world modeling, long-horizon reasoning, and maintaining a consistent internal state under partial observability. **However, our contribution is complementary in isolating these core cognitive capabilities using the cube as a diagnostic framework (the three tiers in our paper).** This decoupling provides a uniquely low-noise diagnostic lens that complements existing embodied benchmarks: it allows us to pinpoint the exact reasoning failures that underlie their broader performance gaps. Hence, our contribution is not scale, but clarity—*CubeBench* exposes the root cognitive bottlenecks that complex 3-D environments can only observe but cannot disentangle.

---

### Official Review · Reviewer_zB6V · 2025-10-31

**Soundness:** 3
**Presentation:** 3
**Contribution:** 3
**Rating:** 6
**Confidence:** 4

**Summary:**

The paper introduces CubeBench, a benchmark designed to evaluate large language model (LLM) agents on spatial and long-horizon reasoning tasks under varying levels of observability. Centered on a Rubik’s Cube environment, CubeBench presents tasks of increasing complexity, defined by the cube’s optimal distance from the solved state. The benchmark comprises two main settings: a fully observable mode, where the agent has access to complete symbolic or visual information, and a partially observable mode, limited to local face or vertex views. The authors further explore how providing dense rewards and external solver tools affects performance, using these interventions to diagnose how effectively agents can represent, aggregate, and reason over spatial information across time.

**Strengths:**

- The choice of the Rubik’s Cube environment is well-motivated, offering a compact yet challenging testbed for spatial reasoning and long-horizon planning, while also enabling automatic verification of solutions.
 - The experimental evaluation is comprehensive, featuring multiple ablations across agent types and settings. The results clearly expose consistent failure modes—particularly in the vertex-level partial observation case—even when solver tools are available, while also highlighting interesting cases where agents learn to use tools in-context.
 - The paper is well-structured and clearly written, making the motivation, methodology, and results easy to follow.

**Weaknesses:**

-  The benchmark does not exclusively measure an agent’s intrinsic ability to mentally simulate and plan within a spatial model, since it permits writing code that performs explicit search. While this capability can be viewed as part of a general planning process, the paper should make this distinction clearer and discuss its implications for the interpretation of results.
 - The sample size per setting appears quite limited — only 20 states per difficulty–tier combination. It is unclear how many independent runs were conducted per model and configuration; if each setup was evaluated only once, the reported results may not be statistically robust or representative.
 - Reporting performance as #MM is less informative than expressing results as a percentage relative to the optimal path length, which would normalize for differing task difficulties and provide a fairer comparison across samples.
 - The pass rate presentation is inconsistent -- expressed as a percentage in Figure 1 but as absolute counts in Table 2 -- which can be confusing or misleading.
 - The claim of testing "physical intelligence" seems overstated, as the benchmark primarily probes spatial reasoning and planning rather than broader aspects of embodied or sensorimotor intelligence. A term like "spatial intelligence" would more accurately describe the benchmark’s current scope.

**Questions:**

- **No-code (pure reasoning) setting**: Could you add an evaluation mode where the agent is not allowed to write or call code (no programmatic search), and must output only a move sequence (e.g., as a JSON list)? This would isolate the model’s internal state tracking and planning ability from code-based search, which it might recall from pretraining data.
-  **Difficulty sensitivity and search usage**: Can you provide a plot of state depth vs. performance? Additionally, a histogram of state depth versus the number of moves (or relative number of moves) would help reveal when agents switch from direct planning to search-based strategies.
 - How many runs per state has the model performed? Can you make a few extra runs to verify the consistency of the results?

---

> ### Author Response · Authors · 2025-11-20
> **Rebuttal**
>
> **[W1]**
>
> We think that writing code is another form of reasoning to solve real problems in the agentic era. However, we acknowledge that distinguishing intrinsic spatial reasoning from programmatic search is crucial. Therefore, for comparison, we implemented your requested "No-Code" evaluation mode. In this mode, the agent must output a pre-planned sequence (e.g., {"moves": ["D'", "B", "B"]}) without the ability to manage control flows during tool use. This mode is more consistent with common tool-use interfaces such as the OpenAI API.
>
>
> |                           | No Code   |       | Code      |          |
> | ------------------------- | --------- | ----- | --------- | -------- |
> |                           | Pass Rate | #MR    | Pass Rate | #MR       |
> | GPT-5                     | 0.25      | 72.79 | 0.75      | 27124.66 |
> | gpt-oss-120b              | 0.00      | 14.20 | 0.20      | 47173.21 |
> | Grok-4                    | 0.30      | 2.02  | 0.20      | 1.84     |
> | Kimi-K2-0905              | 0.00      | 55.15 | 0.15      | 23705.38 |
> | Gemini 2.5 Pro            | 0.15      | 18.68 | 0.10      | 59.01    |
> | Claude Sonnet 4           | 0.00      | 68.03 | 0.05      | 28.59    |
> | DeepSeek-V3.1             | 0.00      | 17.11 | 0.05      | 21.89    |
> | DeepSeek-R1-0528          | 0.00      | 12.28 | 0.05      | 14074.89 |
> | Qwen3-Max                 | 0.00      | 18.11 | 0.05      | 15.92    |
> | GPT-4o                    | 0.05      | 61.53 | 0.00      | 48.10    |
> | GLM-4.5v                  | 0.00      | 4.31  | 0.00      | 16.61    |
>
> **Insights:** (1) Models employing intensive search strategies (High #MR) suffered catastrophic declines. Both GPT-5 (0.75 to 0.25) and gpt-oss-120b (0.20 to 0.00) saw their performance evaporate as their search volume collapsed (e.g., GPT-5 #MR: ~27k to ~72; gpt-oss-120b #MR: ~47k to ~14). This confirms that for these agents, code served as a necessary "cognitive prosthesis" to compensate for limited internal planning. (2) Models with initially low search volume (Low #MR) displayed mixed outcomes rather than a uniform drop. Grok-4 improved (0.20 to 0.30) in the No-Code setting, while many remained poor.
>
> **[W2/Q3]**
>
> We appreciate the reviewer's valuable feedback on the statistical robustness of our results. The comprehensive evaluation in the original paper already incurred substantial costs (~$2,000), so we only reported one group of results in the table. We acknowledge this is not statistically robust. We will revise Table 2 to include 3 runs every group in the revision, and provide some preliminary results here. The numbers are not of a large difference, and the conclusions drawn from the numbers still hold. We will include the complete table in the revision.
>
> | Model            | Full Symbolic Short | Full Visual Short | Face View Short | Vertex View Short |
> | ---------------- | ------------------ | --------------- | --------------- | ----------------- |
> | GPT-5            | 0.700 ± 0.077      | 0.100 ± 0.077   | 0.283 ± 0.113   | 0.033 ± 0.029     |
> | gpt-oss-120b     | 0.333 ± 0.103      | --              | --              | --                |
> | Grok-4           | 0.250 ± 0.077      | 0.100 ± 0.045   | 0.017 ± 0.026   | 0.000 ± 0.000     |
> | Gemini 2.5 Pro   | 0.167 ± 0.052      | 0.050 ± 0.045   | 0.067 ± 0.026   | 0.000 ± 0.000     |
> | Kimi-K2-0905     | 0.083 ± 0.052      | --              | --              | --                |
> | DeepSeek-R1-0528 | 0.033 ± 0.026      | --              | --              | --                |

---

> > ### Comment · Reviewer_zB6V · 2025-11-25
> >
> > **[W1]**
> >
> > Thank you for providing additional evaluations. I agree that the code is crucial; though, designing an extensive search algorithm only proves the model's ability to code, and not the target of this study, which is spatial reasoning. It is therefore interesting that the score for Grok-4 improved without code. Anyway, please include these results in the updated version of the paper.
> >
> > **[W2/3]**
> >
> > Thank you for addressing concerns.

---

> ### Author Response · Authors · 2025-11-20
> **Rebuttal cont.**
>
> **[W3/Q2]**
>
> We agree with the reviewer. Reporting #MM can obscure differences in task difficulty, since optimal path lengths increase non-linearly with scramble depth. In the revision, we will supplement all #MM results with:
> ```
> #MR (number of move ratio)  =  #MM / #OM (number of optimal moves)
> ```
>
> We provide revised top rows of Table 1 here:
>
> | **Model**                    | **#MR: Sym S** | **#MR: Sym L** | **#MR: Vis S** | **#MR: Vis L** | **#MR: Face S** | **#MR: Face L** | **#MR: Vert S** | **#MR: Vert L** |
> | ---------------------------- | ------------- | ------------- | ------------- | ------------- | -------------- | -------------- | -------------- | -------------- |
> | **GPT-5**                    | 27124.66      | **33982.78**  | **149.29**    | **29.22**     | **51.25**      | **17.69**      | **8378.55**    | **672.57**     |
> | **MLP (Policy Gradient)**    | 5.17          | 400.00        | -             | -             | -              | -              | -              | -              |
> | **gpt-oss-120b**             | **47173.21**  | **14219.57**  | -             | -             | -              | -              | -              | -              |
> | **Grok-4**                   | 1.84          | 0.97          | 1.76          | **5.28**      | 1.81           | 0.25           | 18.36          | 0.84           |
> | **Kimi-K2-0905**             | 23705.38      | 118.25        | -             | -             | -              | -              | -              | -              |
> | **Gemini 2.5 Pro**           | 59.01         | 12.55         | 19.30         | 3.13          | 4.05           | 0.55           | 13.28          | 1.85           |
> | **DeepSeek-R1-0528**         | 14074.89      | **4724.51**   | -             | -             | -              | -              | -              | -              |
> | **Claude Sonnet 4**          | 28.59         | 9.05          | 14.81         | 1.93          | 2.67           | 2.43           | 9.40           | 5.75           |
> | **Qwen3-Max**                | 15.92         | 3.48          | -             | -             | -              | -              | -              | -              |
> | **DeepSeek-V3.1**            | 21.89         | 1.66          | -             | -             | -              | -              | -              | -              |
> | **doubao-seed-1-6-vision**   | 4.77          | 0.94          | 6.79          | 1.00          | 6.26           | 0.84           | 7.47           | 1.10           |
> | **InternVL-3 (78B)**         | **56466.74**  | 4.85          | **23.27**     | 3.95          | 18.16          | 2.61           | **29.14**      | 6.84           |
> | **Qwen2.5-VL-72B-Instruct**  | **43712.71**  | 2575.65       | 18.26         | 2.54          | 5.97           | 1.26           | 27.28          | 2.44           |
> | **kimi-vl-a3b-thinking**     | 2623.57       | 0.01          | 0.00          | 0.01          | 0.00           | 0.00           | 0.36           | 0.00           |
> | **GPT-4o**                   | 48.10         | 8.01          | **52.67**     | **8.48**      | **24.85**      | **4.55**       | **55.41**      | **9.41**       |
> | **GLM-4.5V**                 | 16.61         | 5.44          | 15.48         | 3.35          | **28.49**      | **5.17**       | 21.08          | **8.33**       |
> | **Gemma-3-27B-IT**           | 12.29         | 1.56          | 15.49         | 2.09          | 9.38           | 1.64           | 11.49          | 1.21           |
> | **Seed-OSS-36B-Instruct**    | 9.22          | 0.93          | -             | -             | -              | -              | -              | -              |
>
>
> **[W4]**
>
> Thank you for pointing out the inconsistency. We will standardize the reporting format to percentage (%) across all figures and tables for clarity.
>
> **[W5]**
>
> We appreciate this feedback. We agree that “physical intelligence” may suggest sensorimotor or force-aware reasoning, which is not the focus of CubeBench. To avoid overstating the scope, we will adjust terminology to “spatial reasoning” and clarify the intended scope in the introduction.

---

> ### Author Response · Authors · 2025-11-20
> **Rebuttal cont.**
>
> **[Q2]**
>
> We will include depth-vs-performance curves and depth-vs-move-ratio histograms in the revision. Here we summarize the key trends:
>
> **1/ Depth-vs-Performance**
>
> **In the Tier 1 (Full Symbolic) setting**, top-tier models capable of systematic search (e.g., GPT-5) maintain high accuracy across Depths 1–4 (0.73 to 0.67), with pass rates decaying gracefully. Conversely, less capable models (e.g., Grok-4) succeed at Depth 1 (0.87) via direct policy but degrade immediately at Depth 2 (0.07).
>
> **In Tier 2/3 (Visual) settings**, the performance cliff arrives earlier. While agents can manage Depth 1 tasks, we observe a drop starting at just Depth 2.
>
> A universal limit appears between Depth 4 and 8. Pass rates drop to 0.00% at Depth 8 across all settings, confirming that current planning capabilities fail to track state over extended horizons.
>
> **2/ Depth-vs-Move-Ratio**
>
> Our analysis of state depth versus move-ratio (#MR) reveals three critical insights regarding agent behavior:
>
> **The "Transition to Search" in Specific Models:**
> For a subset of models (e.g., **Grok-4, DeepSeek-R1-0528, Kimi-K2-0905**) under Symbolic input, we indeed observe the distinct transition hypothesized by the reviewer. These agents expend minimal moves at Depth-1 (relying on direct policy), but move counts explode immediately at Depth-2, signaling a sharp shift from intuition to brute-force search. However, this is not a universal linear law; across the broader model landscape, there is no stable linear correlation between depth and search volume.
>
> **Move Count as a Cognitive Indicator:**
> We interpret the #MR metric not as a measure of “effort” scaling linearly with difficulty, but as an **indicator of cognitive mode**. A low move count typically signals reliance on direct intuition or trivial heuristics, while a high move count signals the activation of explicit search. The switch is often a discrete jump rather than a gradual increase.
>
> **Intra-Trial Strategy Evolution:**
> A close examination of execution traces reveals how agents dynamically adapt within a single trial. Typically, search-based agents begin with low-cost heuristics, switching to high-volume search when those initial attempts fail. In rare instances (e.g., Appendix H.2), we observe a further methodological pivot: abandoning search entirely to implement structured human algorithms, such as the Beginner’s Method, as a fallback.

---

> ### Author Response · Authors · 2025-11-27
> **Reply to Reviewer zB6V**
>
> Thank you very much for your thoughtful follow-up comments and for engaging so carefully with our rebuttal :)
>
> We agree that distinguishing code-based search from intrinsic spatial reasoning is essential for interpreting the benchmark's results. We will include the No-Code evaluation setting, the expanded runs, statistical reporting, the normalized metrics—and corresponding discussions—in the revised version of the paper.
>
> If you have any further questions or suggestions, please feel free to let us know. We are happy to provide additional details or analyses.

---

### Official Review · Reviewer_ScbV · 2025-11-01

**Soundness:** 2
**Presentation:** 3
**Contribution:** 3
**Rating:** 4
**Confidence:** 3

**Summary:**

This paper introduces CubeBench, a clever benchmark using the Rubik’s Cube to test LLMs on three tough abilities: spatial reasoning, long-horizon planning, and active exploration under partial observation. It defines three tiers — full symbolic, full visual, and partial visual states to progressively stress these skills.

The experiments are thorough and frankly pretty sobering. None of the tested models (GPT-5, GPT-4, open-source LLMs) managed any long-horizon solves 0% success across the board. Even short tasks were shaky: GPT-5 hits ~75% on symbolic cubes but drops to 20% with images and nearly 0% when only partial views are given. It’s a clean way to show that current models can describe cubes but can’t really think in 3D or maintain state.

Reward shaping helps a bit on short cases (0.2 - 0.55 success), but doesn’t fix planning. Adding a classical solver module boosts success to nearly 100% once perception is perfect, proving the core weakness is long-range reasoning, not local step choice. But even then, partial-view perception totally breaks them still 0%.

Overall, CubeBench is a strong diagnostic tool. It clearly isolates where LLMs fail — in sustained reasoning, 3D mapping, and exploration. The work is solid, though maybe a bit narrow (just Rubik’s Cube). Still, it makes a sharp point: today’s models can talk about the physical world, but they can’t mentally move in it yet.

**Strengths:**

This paper does a great job of cutting straight to the heart of the problem to clearly identifies three big cognitive gaps holding LLM agents back in physical reasoning: 3D spatial understanding, long-horizon planning, and exploration under partial observation. It’s refreshing to see these challenges framed so directly, with tasks designed specifically to isolate each one rather than mixing everything together.

The idea of using the Rubik’s Cube as a controlled testbed is honestly quite clever. It’s simple enough to simulate and evaluate, but complex enough to expose real weaknesses in reasoning. The three-tier setup (symbolic - visual - partial visual) works really well it’s systematic, and you can literally see where the models start to fall apart.

The experiments are solid and well-rounded. The authors test multiple top-tier LLMs (GPT-4, GPT-5, open models) and even include comparisons with smaller RL agents. The results are both surprising and sobering none of the models solved long-horizon cubes (0% success!), and even short ones get much harder as soon as the cube is shown visually instead of symbolically. The dense-reward experiments are a nice touch, showing that short-term feedback can help a bit but doesn’t fix the fundamental planning issue. And giving the agent an external solver was a smart diagnostic move success jumps instantly, proving the bottleneck really is long-term reasoning, not basic logic.

The paper is also cleanly written and easy to follow. The structure, figures, and summaries make it painless to digest what’s otherwise a pretty complex set of experiments. You can tell the authors put care into making the results interpretable rather than just dumping numbers.

Overall, this is a well-motivated and genuinely insightful piece of work. It’s not just another “benchmark paper” it actually deepens understanding of why LLMs struggle with embodied, physical reasoning. CubeBench feels like something the field could really build on for years, especially for testing new architectures that claim to reason spatially or plan over long horizons.

**Weaknesses:**

The use of the Rubik’s Cube is smart but narrow it’s a clean test of reasoning, but still a very specific domain. It’s not obvious if the lessons from CubeBench will generalize to other physical problems like robotic manipulation or navigation. A short discussion or even a small test on another environment would make the results feel less domain-locked.

The heavy reliance on GPT-5 is another weak spot. Since the model isn’t public or well-documented, readers can’t really verify or reproduce much of the analysis. The paper could be clearer about how GPT-5 was accessed and what capabilities it had, especially for the visual tiers. Including more open-source baselines, even weak ones, would make the results more trustworthy.

I also wanted a bit more about how the models behave. The paper hints that GPT-5 sometimes tries tens of thousands of moves or performs better with sequential observations than full maps, but doesn’t dive into why. Some short qualitative analysis what reasoning patterns or blind spots the model shows would add real depth.

Another missing piece is any attempt to train or adapt the models. The work focuses on evaluation and simple interventions (dense rewards, external solver), but doesn’t test whether fine-tuning or curriculum learning might help. That leaves an open question: are these failures fundamental, or just lack of experience with structured spatial tasks?

Overall, a few clarity nits: “GPT-5” appears without explanation, definitions of “short” vs “long” horizon are easy to miss, and some table cells are left blank without note. Minor, but they add small friction to an otherwise very clean paper.

**Questions:**

Do you expect CubeBench insights to generalize to broader physical tasks like manipulation or navigation? Or is it mainly diagnostic for cube-like puzzles? Some discussion on how this setup could extend to other domains would strengthen the paper’s relevance.

Why did models perform better with sequential face-by-face views than full 2D maps? Is it due to attention limits, easier focus, or prompt design differences? A short analysis of this counterintuitive result would be helpful.

Since the MLP agent learned via RL, did you consider fine-tuning an LLM similarly? Would more exposure or reward training help overcome the 0% long-horizon failures, or is the limitation inherent to transformer reasoning?

In the Standard-Solver setup, where exactly do models fail—misreading colors, wrong face mapping, or confusion in coordinate translation? A brief error breakdown would clarify which spatial step is the main bottleneck.

---

> ### Author Response · Authors · 2025-11-20
> **Rebuttal**
>
> **[W1/Q1] generalization to broader physical tasks**
>
> Our goal is not to claim it covers the full spectrum of embodied tasks, but to use the cube as a **diagnostic anchor** that isolates three core cognitive abilities—
>
> (1) 3D spatial reasoning,
>
> (2) long-horizon non-commutative planning, and
>
> (3) belief-state construction under partial observability—without the confounds of physics, object affordances, or actuation.
>
> These three abilities are precisely the ones repeatedly identified as bottlenecks in **manipulation** (VisiBench, ManipulationNet, BEHAVIOR-100), **navigation** (Habitat, ALFRED), and **long-horizon planning** (PlanBench/SPIN-Bench). Moreover, **our contribution is complementary in isolating these core cognitive capabilities using the cube as a diagnostic framework (the three tiers in our paper).**
>
> This **decoupling** provides a uniquely low-noise diagnostic lens that complements existing embodied benchmarks: it allows us to pinpoint the exact reasoning failures that underlie their broader performance gaps. Hence, our contribution is not scale, but clarity—CubeBench exposes the root cognitive bottlenecks that complex 3-D environments can only observe but cannot disentangle.
>
> **[W2]**
>
> **GPT-5 Reproducibility**: We accessed GPT-5 via the public OpenRouter API (ID: openai/gpt-5), with no specific reasoning effort specified. The GPT-5 model has been trained to have the ability to determine the appropriate reasoning effort by the model itself during inference.
>
> **How Visual Signals Are Processed**: The visual signals can be processed in two ways. First, as a coding agent, the agent can call the `get_observation` function, which returns a PIL Image object that the agent can directly access and parse in its code. Second, at the end of each step, our framework collects images by get_observation during that step and organizes them into the context of the agent, which are then sent to the API as prompts.
>
> **More Open-Source Models**: We have included more recent open-source models to broaden our evaluation. The table below presents the results for Experiment 1, specifically conducted under the Tier 1 (Full Symbolic) setting on Short-Horizon tasks (Depths 1–4):
>
> | Model                         | Pass Rate | #MM    |
> | ----------------------------- | --------- | ------ |
> | Qwen3-235b-a22b-Thinking-2507 | 0.15      | 2.35   |
> | Qwen3-VL-235b-a22b-Thinking   | 0.05      | 15.40  |
> | GLM-4.6                       | 0.05      | 17.40  |
> | Minimax-m2                    | 0.00      | 27.70  |
> | Kimi-K2-Thinking              | 0.00      | 121.85 |
> | Kimi-Linear-48b-a3b-Instruct  | 0.00      | 64.20  |
> | Qwen3-Next-80b-a3b-Thinking   | 0.00      | 12.55  |
>
> **[W3] Qualitative analysis of GPT-5 behavior**
>
> GPT-5 tends to systematically search using algorithms like beam search and iterative deepening depth-first search (IDDFS) with skills like backtracking. **Please see our analysis within Sec. 4.1**, as well as the case study provided with in Sec. H.1 and Sec. H.2. We also find that less capable agents often devolve into largely unguided enumeration (shown in H.3).
>
>
> **[W4/Q3] On training, adaptation, and whether failures are fundamental**
>
> We agree that it is valuable to test whether the observed failures stem from lack of exposure rather than intrinsic reasoning limits. To investigate this, we conducted additional fine-tuning experiments on an open-source backbone (Qwen/Qwen2.5-1.5B-Instruct), using both **supervised fine-tuning (SFT)** and **reinforcement learning (GRPO)** on CubeBench generated trajectories. These experiments reveal that **long-horizon failures persist even when the model is trained directly on cube dynamics**, suggesting the limitation is structural rather than data-related.
>
>
> **SFT results (teacher-forced trajectories)**
>
> - **Scramble depth 1–2:** 100% accuracy; the model easily memorizes simple 1–2 move patterns.
> - **Scramble depth 3:** accuracy drops to **~20%**, despite abundant examples.
> - **Scramble depth ≥4:** **0%** accuracy; the model fails to generalize any reversible structure beyond rote pattern imitation.
>
> This indicates that all successes come from shallow pattern matching, not long-horizon planning.
>
>
> **RL results (GRPO)**
>
> The RL-trained model performs even worse:
>
> - **Scramble depth 1:** 100%.
> - **Scramble depth 2:** ~20%.
> - **Scramble depth ≥3:** 0%.
>
> These two group of results are even substantially below a simple **MLP** baseline. The RL agent converges to local, greedy policies and fails to maintain a stable latent cube state across more than 2–3 reversible actions—a hallmark of **missing structured world-model inductive bias**, not missing experience.
>
>
> Both interventions show the same pattern:
>
> - training improves **only trivial short-horizon patterns**;
> - performance collapses at exactly the same horizon length as in the zero-shot LLMs;
> - the model never acquires the non-commutative, group-structured dynamics needed for long-horizon planning.

---

> > ### Author Response · Authors · 2025-11-20
> > **Rebuttal cont.**
> >
> > **[W5]**
> >
> > We will incorporate the reviewer’s suggested fixes:
> >
> > - add a brief explanation the first time “GPT-5” appears;
> > - define “short horizon” vs “long horizon” explicitly in the primary result table;
> > - add footnotes where table entries are blank because they cannot process visual inputs.
> >
> > These edits improve readability, and we thank the reviewer for noting them.
> >
> >
> > **[Q2]**
> >
> > We believe the effect is primarily due to how current multimodal LLMs process visual information. Modern MLLMs (including GPT-5 and open-source VLMs) operate by passing images through a compressed visual encoder before fusion with the language stack. This architecture is powerful for broad semantic tasks but not optimized for precise, **high-fidelity** spatial parsing, especially when multiple dense spatial regions must be interpreted simultaneously. That's a field that many research papers are trying to tackle in recent days, such as 3D-LLM or VLM-3D. The sequential face-by-face setting reduces the load on the encoder and yields more stable reasoning.
> >
> > When all faces are shown simultaneously, we have observed that errors often come from: (1) mixing stickers between adjacent faces, (2) confusing rotations or cube orientation, (3) interpreting the cube’s projection incorrectly. Sequential views eliminate these cross-face interactions, making each grounding step more reliable.
> >
> >
> > **[Q4] Error case studies**
> >
> > Actually, we have shown many case studies in Sec. H. We found that most of the failures in this settings are attributed to visual-to-symbolic mis-mapping (e.g., misplacing colors on the wrong face) or orientation drift (misinterpreting which face is “front” after several rotations). The number of formatting or schema issues is really low since the tool would trigger errors and the LLM could correct itself.

---

### Official Review · Reviewer_aJqe · 2025-11-01

**Soundness:** 3
**Presentation:** 3
**Contribution:** 2
**Rating:** 6
**Confidence:** 3

**Summary:**

The paper presents CubeBench, a three-tier benchmark for diagnosing LLM agents’ spatial reasoning, long-horizon state tracking, and exploration under partial observation using Rubik’s Cube tasks. Tier-1 exposes a full symbolic state, Tier-2 a full visual unfolded map, and Tier-3 partial visual observations with view-change actions; evaluation follows a Thought-Code-Observation loop with strict budgets. Across many frontier models, all long-horizon settings achieve 0.00 pass rate. Dense reward shaping improves some short-horizon cases but does not fix long-horizon failures. Equipping agents with solver tools sharply improves performance and separates planning from spatial translation challenges. Overall, CubeBench provides a verifiable, decoupled diagnostic suite that yields crisp negative results and actionable insights for agentic research.

**Strengths:**

1. The three-tier design cleanly isolates (i) symbolic state tracking, (ii) visual/spatial reasoning, and (iii) exploration under partial observation.
2. The universal 0.00 pass rate on long-horizon tasks highlights a concrete direction where current LLM agents systematically struggle.
3. The paper contrasts sparse terminal rewards with thoughtfully defined dense-shaping variants (sticker/face/heuristic), enabling reproducible ablations.
4. End-to-end diagnostic arc. Framing the Rubik’s Cube across presentation (state/observation), optimization (with/without shaping), and tool use (standard/ideal solver) yields a coherent suite of ablations that pinpoint where capabilities break.

**Weaknesses:**

1. While the cube is a controlled, verifiable domain, it’s not yet clear why it’s the right anchor benchmark relative to other long-horizon planners (e.g., PlanBench, SPIN-Bench, ARC/ARC-AGI) or embodied 3D reasoning suites (e.g., VisiBench, ALFRED). A side-by-side rationale (capabilities stressed, evaluation verifiability, cost, and failure taxonomy) would better justify CubeBench as a first diagnostic stop rather than a narrow puzzle niche.
2.Tool-use evaluation may entangle multiple difficulties as well. “Standard-Solver” demands both perception→symbolic translation and strict format compliance; “Ideal-Solver” removes translation entirely. Without granular error audits, it’s hard to tell whether failures are due to spatial mapping, formatting, or simple string schema slips.

**Questions:**

1. Could you break out pass rates by each target depth (1,2,3,4,8,12,16,20) to show precisely where performance collapses?
2. Why Rubik’s Cube? Please articulate a principled rationale vs. existing planning/3D benchmarks: what capabilities are uniquely isolated here (beyond verifiability), and which insights demonstrably transfer to non-cube domains?

---

> ### Author Response · Authors · 2025-11-20
> **Rebuttal**
>
> **[W1]**
>
> We appreciate this concern. In the revision, we will extend Table 1 to provide a clearer side-by-side comparison against PlanBench, SPIN-Bench, ARC/ARC-AGI, VisiBench, and ALFRED—including (i) what capabilities each benchmark stresses, (ii) whether outcomes are verifiable, (iii) whether the environment is static or dynamic, and (iv) whether failure modes can be cleanly attributed.
>
> The key rationale is that CubeBench is not meant to compete with these benchmarks in coverage, but rather to fill an orthogonal gap: it is the only environment that cleanly decouples the three core cognitive abilities—3D spatial reasoning, long-horizon non-commutative planning, and belief-state tracking—without confounds from perception noise, multi-object dynamics, affordances, or actuation.
>
> Other benchmarks necessarily entangle these factors, making it difficult to isolate which ability causes failure. CubeBench serves as a first diagnostic stop precisely because it is minimal, verifiable, and factorized.
> We will make this justification explicit in the revised manuscript.
>
> **[W2] On tool-use evaluation and disentangling perception vs. reasoning errors**
>
> We appreciate the reviewer’s concern. Our intention was not to evaluate tool use as a monolithic skill, but to use multiple solver × tier combinations to diagnose different cognitive faculties by systematically removing confounding factors. The three solver types—Basic Agent, Standard-Solver, and Ideal-Solver—serve as controls rather than additional complexities.
>
> To clarify, we diagnose visual translation ability (visual → symbolic) via comparing Standard-Solver vs. Ideal-Solver on Tier 2 (Visual)
> - Standard-Solver w/ Tier 2 (Visual)
>   - The agent must:
>     - (i) ground the 2D visual input into a correct 3D symbolic cube state, and
>     - (ii) format the state correctly for the optimal solver.
> - Ideal-Solver w/ Tier 2 (Visual)
>   - We remove both planning and formatting challenges: the agent only needs to output the symbolic cube state.
>
> Any gap between these two settings isolates spatial reasoning + visual-to-symbolic translation, because planning and formatting are controlled away. This directly addresses the reviewer’s concern: failures here cannot be due to planning or schema slips.
> There are more combinations that diagnoses planning and long-horizon state tracking or diagnoses exploration and belief-state construction. We kindly refer you to Section 4.3 (“The Diagnosing Framework”), where these ablations and their interpretations are discussed in depth.
>
> **[Q1] Depth-wise pass rates**
>
> We will include a figure illustrating pass rates for different depths in the revision. For now, as we cannot upload figures, we post the findings here.
>
> In the Tier 1 (Full Symbolic) setting, top-tier models capable of systematic search (e.g., GPT-5) exhibit a performance plateau across Depths 1–4, maintaining high pass rates without significant degradation. Conversely, less capable models struggle to generalize beyond immediate greedy moves, showing a sharp performance decay after Depth 1 or 2.
>
> In Tier 2/3 (Visual) settings, the performance cliff arrives earlier. While agents can manage Depth 1 tasks, we observe a drop starting at just Depth 2.
>
> Moreover, a barrier appears between Depth 4 and Depth 8. Across all models and modalities, pass rates collapse to 0.00% at Depth 8 and remain flatlined through Depths 12, 16, and 20.
>
> This confirms the key claim of CubeBench: failures arise from long-horizon non-commutative reasoning, not from parsing or formatting.
>
> **[Q2]**
>
> Beyond verifiability, the **Rubik’s Cube is uniquely suited to diagnostic evaluation** because:
>
> **1/ It isolates 3 distinct reasoning abilities via our three tiers**
> - Symbolic: pure long-horizon, non-commutative planning
> - Visual: 3D spatial mapping from a single view
> - Partial-visual: belief-state construction and reversible mental simulation.
>
> These three abilities cannot be cleanly separated in existing benchmarks.
>
> **2/ It is a minimal physical system with rich structure**
> - One rigid object, no affordances, no multi-object noise, no physics uncertainty
> - Yet rotations induce highly structured, irreversible state coupling -> ideal for diagnosing reasoning errors.
>
> **3/ Insights transfer beyond the cube**
>
> The same failure modes exposed here—incorrect spatial world modeling, short effective planning horizons, unstable belief-state tracking—match those recently reported in long-horizon planning suites (PlanBench/SPIN-Bench) and embodied manipulation/navigation benchmarks (VisiBench/ALFRED).
>
> CubeBench complements these benchmarks by explaining why such failures occur. We will clarify these points in the revised introduction and discussion.

---

### Author Response · Authors · 2025-12-02

Dear Area Chair,

We sincerely appreciate your efforts during this challenging period. We understand the exceptional workload you're managing given the recent circumstances, and we're grateful for your careful consideration of our work. To further assist you in gaining a deeper understanding of the manuscript, we would like to provide a summary of our recent discussions with the reviewers.

**Reviewer Assessment Overview**

All reviewers recognized the novelty of our benchmark and the thoroughness of our empirical analysis. Three reviewers (aJqe, zB6V, kiAJ) gave scores of 6, all at or above the acceptance threshold. They commended the "three-tier design" which cleanly isolates cognitive capabilities and recognized CubeBench as a "strong diagnostic tool."

The concerns mainly come from Reviewers ScbV and tzUm (both Score 4), who raised questions regarding the benchmark's scope (generalization to real-world tasks), and the validity of the partial-observation setting. For the first concern, our response clarifies that CubeBench acts as a low-noise diagnostic anchor to pinpoint bottlenecks (spatial reasoning, long-horizon planning, belief-state tracking) that are entangled in broader embodied benchmarks. For the last one, we have provided additional experiments (including SFT/RL fine-tuning and No-Code settings) and detailed clarifications.

**Discussion Details**

Below, we will provide a description of the discussion with each reviewer. To objectively present the state of the discussion, we will quote the reviewers’ original comments from the discussion phase whenever possible:

**Reviewer aJqe (Rating: 6):**
- Explicitly highlights the value of our design: "The three-tier design cleanly isolates... symbolic state tracking, visual/spatial reasoning, and exploration."
- The reviewer questioned the rationale of CubeBench relative to other benchmarks (e.g., PlanBench, ALFRED). We clarified that CubeBench fills an orthogonal gap by decoupling core cognitive abilities without confounds from perception noise or physics.
- The reviewer also asked for depth-wise analysis, which we have provided it in the Appendix in the revision.

**Reviewer zB6V (Rating: 6):**
- Actively engaged in the discussion and acknowledged our new experiments.
- The reviewer initially questioned if code generation masks intrinsic spatial reasoning. In response, we implemented a "No-Code" evaluation mode. The reviewer responded: "I agree that the code is crucial... Thank you for providing additional evaluations."
- Reviewer zB6V also recognized the strengths of our work, stating: "The choice of the Rubik’s Cube environment is well-motivated... The experimental evaluation is comprehensive." We also addressed the reviewer's statistical concerns by increasing the number of runs per setup.

**Reviewer kiAJ (Rating: 6):**
- Validates the insights offered by our work: "Large-scale comparison across 15+ proprietary/open LLMs yields actionable insights for the community."
- Addressing the concern about the limited scope of a single puzzle, we clarified that CubeBench strictly isolates 3D spatial modeling, long-horizon planning, and belief-state tracking by stripping away environmental confounds (e.g., physics, sensor noise) to target intrinsic reasoning capabilities.
- Regarding the reviewer's question on real-world generalization, we provided evidence from recent literature, including EmbodiedBench (ICML 2025), EAI (NeurIPS 2024 D&B) and 3DLLM-Mem (2025), to show that the failure modes isolated by CubeBench are the exact bottlenecks identified in complex manipulation and navigation tasks. We clarified the consistency with recent 3-D embodied benchmarks, emphasizing that our contribution is complementary in isolating core cognitive capabilities within a diagnostic framework.

---

> ### Author Response · Authors · 2025-12-02
>
> **Reviewer ScbV (Rating: 4)**
> - Acknowledged that our work is "well-motivated and genuinely insightful," highlighting that it successfully identifies "three big cognitive gaps." The reviewer raised four specific concerns, all of which we have addressed in detail:
> - **Generalization of the Cube:** The reviewer worried the domain might be "narrow." We clarified that CubeBench acts as a **diagnostic anchor**. We explained how the specific failures isolated by the cube (e.g., belief-state tracking) map directly to bottlenecks reported in broader benchmarks like EmbodiedBench and ALFRED, providing the "why" behind those failures.
> - **Training and Adaptation:** The reviewer questioned whether the failures were fundamental or simply due to a lack of exposure, asking for fine-tuning experiments. **We directly addressed this by conducting new extensive experiments using both Supervised Fine-Tuning (SFT) and Reinforcement Learning (GRPO) on Qwen2.5.** The results were decisive: while training improved performance on trivial short-horizon tasks (Depth 1-2), performance still collapsed to 0% at long horizons (Depth $\ge$ 4). This empirically proves our hypothesis: the limitation is structural (lack of inductive bias for non-commutative planning) rather than data-related.
> - **GPT-5 Reproducibility & Analysis:** We clarified that we used the public OpenRouter API. Regarding the reviewer's request for qualitative analysis of GPT-5's behavior, we pointed to our case studies (Sec. 4.1 & Appendix) showing that GPT-5 employs systematic search (Beam Search/IDDFS), whereas weaker models degrade into random enumeration.
> - **Clarity:** We adopted all the reviewer’s suggestions regarding definitions of horizon lengths and reporting formats.
>
> **Reviewer tzUm (Rating: 4)**
> - Found the benchmark "useful" and the evaluation "solid". The reviewer’s score was contingent on two main conceptual concerns ("I will consider raising my score if these two concerns are resolved"), which we have comprehensively clarified:
> - **Comparison with Classical Solvers:** The reviewer asked why we do not compare with symbolic tools (like Kociemba’s algorithm). We clarified the **fundamental distinction in scope**: Classical solvers rely on domain-specific, handcrafted algorithms, whereas CubeBench is designed to diagnose the **intrinsic reasoning capabilities of general-purpose LLMs**.  The comparison is structurally invalid as the goal is evaluating intrinsic intelligence, not algorithmic speed. **The value lies in understanding *why* LLMs fail where simple algorithms succeed.**
> - **Validity of Visual & Partial Tiers:**
>   - *Full Visual Tier:* The reviewer questioned if visual tasks are just symbolic tasks in disguise. We explained that the Visual Tier probes the **grounding pipeline**—constructing structure from pixels—isolating errors in visual-to-symbolic mapping that symbolic tasks cannot detect.
>   - *Partial Visual Tier:* The reviewer suggested partial observability is trivial because agents can "rotate to see." We emphasized that this is precisely the intended challenge: it tests **human-like interaction with limited field-of-view**.  The tier is not meant to make the full state **unreachable**; it is meant to stress the model’s ability to **track a latent world state over a long action horizon** under partial observations. This is directly targeting the “long-horizon state tracking through mental simulation” capability we highlight in the paper, where most models fail, even though, as the reviewer notes, a conceptually simple “scan–reconstruct–rewind” strategy would suffice.
> - **Related Work:** We also incorporated the reviewer's suggestion to discuss *LogiCity*. We have explicitly positioned *CubeBench* alongside LogiCity, emphasized this complementary relationship and the different aspects of partial observability that each benchmark targets.

---

### Meta-Review · Area_Chair_3pF4 · 2026-01-07

**Summary:**

Benchmark justification:
Reviewers like the Rubik’s Cube as a controlled, verifiable planning domain, but argue the paper does not clearly justify why this is the right anchor benchmark compared to other long-horizon planning or reasoning suites (e.g., PlanBench, SPIN-Bench, ARC/ARC-AGI) or embodied/3D benchmarks (e.g., VisiBench, ALFRED). They ask for a clearer side-by-side rationale, what capabilities CubeBench stresses, what is uniquely verifiable, cost/compute implications, and what failure taxonomy it enables, so it reads as a principled diagnostic rather than a narrow puzzle niche.

Generalization concerns: single-puzzle domain may not transfer.
Multiple reviewers worry conclusions may not generalize beyond Rubik’s Cube to broader “physical intelligence,” such as robotic manipulation or navigation. They recommend either adding a small additional environment or at least discussing expected correlations and limitations, and toning down claims (e.g., “physical intelligence” → “spatial intelligence”).

Reproducibility and over-reliance on GPT-5.
The paper relies heavily on GPT-5, which reviewers note is not publicly documented, making results hard to verify or reproduce. They ask for clearer disclosure on model access/capabilities (especially for visual tiers) and more open-source baselines, even if weaker, to improve trustworthiness.

Need for deeper qualitative analysis of model behavior.
Reviewers want more insight into why models fail or succeed, e.g., why GPT-5 sometimes produces extremely long move sequences, why sequential observations may outperform full maps, and what reasoning patterns/blind spots are observed. Qualitative examples and failure modes would add depth.

**Reviewer Concerns:**

After the rebuttal, reviewers broadly acknowledged the novelty of our benchmark and the thoroughness of our empirical evaluation. Three reviewers (aJqe, zB6V, kiAJ) assigned scores of 6 at or above the acceptance threshold and highlighted CubeBench’s value as a “strong diagnostic tool.” In particular, they praised the three-tier design for cleanly isolating different cognitive capabilities.

One reviewer also emphasized an important nuance regarding tool use: while allowing code is useful, building an extensive search routine mainly demonstrates a model’s coding ability rather than the intended target, spatial reasoning and internal planning. In this context, it is especially notable that Grok-4’s performance improved even in the no-code setting. We will incorporate these no-code results in the updated version of the paper to better separate spatial reasoning from programmatic search.

**Reviewer Scores:**

After the rebuttal, reviewers broadly acknowledged the novelty of our benchmark and the thoroughness of our empirical evaluation. Three reviewers (aJqe, zB6V, kiAJ) assigned scores of 6 at or above the acceptance threshold and highlighted CubeBench’s value as a “strong diagnostic tool.” In particular, they praised the three-tier design for cleanly isolating different cognitive capabilities.

One reviewer also emphasized an important nuance regarding tool use: while allowing code is useful, building an extensive search routine mainly demonstrates a model’s coding ability rather than the intended target, spatial reasoning and internal planning. In this context, it is especially notable that Grok-4’s performance improved even in the no-code setting. We will incorporate these no-code results in the updated version of the paper to better separate spatial reasoning from programmatic search.

---

### Decision · Program_Chairs · 2026-01-26

Accept (Poster)